# Temporal Graph Thumbnail: Robust Representation Learning with Global Evolutionary Skeleton

**Weining Shi**[1], **Zhisen Wen**[1], **Chentao Zhang**[1], **Qinggang Zhang**[2], **Zhihong Zhang**[1*]

[1] School of Informatics, Xiamen University, Xiamen, Fujian, China
[2] The Hong Kong Polytechnic University, Hong Kong, China

```
{weining, wenzhisen, 30920241154582}@stu.xmu.edu.cn,
zhihong@xmu.edu.cn,
qinggangg.zhang@connect.polyu.hk
```

## Abstract

Temporal graphs are commonly employed as conceptual models for capturing time-evolving interactions in real-world systems. Representation learning on such non-Euclidean data typically depends on aggregating information from neighbors, and the presence of temporal dynamics further complicates this process. However, neighbors often contain noisy information in practice, making the unreliable propagation of knowledge and may even lead to the model failure. Although existing methods employ adaptive spatiotemporal neighbor sampling strategies or temporal dependency modeling frameworks to enhance model robustness, their constrained sampling scope limits handling of severe noise and long-term dependencies. This limitation can be attributed to a fundamental cause: neglecting global evolution inherently overlooks the temporal regularities encoded in continuous dynamics. To address this, we propose the **T**emporal **G**raph **T**humbnail (**TGT**), encapsulating a temporal graph's global evolutionary skeleton as a `thumbnail` to characterize temporal regularities and enhance model robustness. Specifically, we model the `thumbnail` by leveraging von Neumann graph entropy and node mutual information to extract essential evolutionary skeleton from the raw temporal graph, and subsequently use it to guide optimization for model learning. In addition to rigorous theoretical derivation, extensive experiments demonstrate that TGT achieves superior capability and robustness compared to baselines, particularly in rapidly evolving and noisy environments. The code is available at `https://github.com/shiwning/TGT`.

## 1 Introduction

Temporal graphs arise naturally in a wide range of real-world domains, including transportation systems (Zhao et al., 2019; Yu et al., 2018; Li et al., 2018; Guo et al., 2019), recommendation systems (Xiang et al., 2010; Fan et al., 2021), and social networks (Deng et al., 2019; Tang et al., 2009). Representation learning on non-Euclidean structures is already difficult, and the temporal variation of node features and graph topology makes it even more complex (Gravina & Bacciu, 2024). Most studies on temporal graph representation extend the message aggregation and passing mechanisms (Gilmer et al., 2017) from static graph learning to dynamic settings. Some studies address temporal dynamics by slicing graphs into discrete snapshots and analyzing them as time series (Seo et al., 2018; Li et al., 2018; Bai et al., 2021), while others model temporal changes in a continuous, online fashion (Micheli & Tortorella, 2022; da Xu et al., 2020; Rossi et al., 2020).

However, real-world data are often noisy, posing substantial challenges to model robustness and generalization. For illustration, consider a Q&A forum where user posts are modeled as nodes

---

*Corresponding author.

and user interactions are edges. There are various sources of noise in this social network temporal graph, including off-topic information in an answer, incorrect associations between answers and unrelated questions, and message disorder caused by network latency. Since context for representation is captured by aggregating spatiotemporal neighbor information along the topology, redundant or erroneous node features, spurious or obsolete edges, and incorrectly recorded timestamps can significantly degrade context quality, thus weakening the representation quality (Hou et al., 2020).

Several robust representation approaches have been developed to combat noise in temporal graph learning. These studies can be broadly categorized into two main directions: *a)* **neighbor sampling adaptation.** Adjusting spatiotemporal neighbor sampling improves the reliability of aggregated messages by reducing the influence of noisy or unreliable neighbors. For instance, Li et al. (2024a) conceptualizes the neighbor update as a sequence decision problem and employs reinforcement learning to address it effectively. Chen et al. (2018) models the propagation of noise as a Markov chain and proposes a greedy algorithm to rewire edges for neighbor sampling. *b)* **historical information assistance.** By capturing consistent patterns in historical interactions, temporal modeling reduces sensitivity to transient noise or irregular events, thereby improving model robustness. Zhang et al. (2022) mitigates spatiotemporal distribution shifts by making full use of invariant historical patterns observed across sequences, while Yuan et al. (2024) derives consensual conditions for temporal information and devise information bottleneck to capture temporal correlations.

However, these approaches remain limited by their constrained sampling scope, restricting their ability to handle denser noises and capture long-term temporal dependencies. Specifically, Lee et al. (2024) observed experimentally that models relying on neighbor or historical information struggled with long-term dependencies, leading to larger errors. Sankar et al. (2020) shows that when both neighbor and historical information become unreliable, the quality of the learned embeddings can significantly deteriorate. We attribute these limitations to two main factors. *a)* **lack of global evolution modeling.** Existing methods adopt a narrow scope of neighbor sampling, overlooking the useful information from global evolution. When local information is insufficient to resist the noise interference, model performance degrades significantly. *b)* **absence of effective constraints for denoising.** Excessive denoising can distort critical information, while insufficient processing leaves residual noises. In the absence of constraints, striking a balance between the two becomes difficult.

To this end, we introduce the **T**emporal **G**raph **T**humbnail (**TGT**) framework, which characterizes the global evolutionary skeleton of temporal graphs as a **thumbnail** and employs it as an optimization constraint to guide effective compression and denoising of representations, improving both the capability and robustness of the model. Similar to a video cover, which uses a single image to summarize the video's content, we extract a static graph, akin to a snapshot, to encapsulate the evolutionary information of the temporal graph into its `thumbnail`. Specifically, TGT addresses the aforementioned challenges by addressing two key research questions: *(i)* How to model a `thumbnail` from the temporal graph that effectively serves as its skeleton with the global evolution features, and *(ii)* how to design effective constraints based on the `thumbnail` to guide robust learning process.

In our TGT framework, we design a thumbnail modeling approach that derives the conditional likelihood estimation of the `thumbnail` under the raw temporal graph. It characterizes structural evolution grounded in von Neumann graph entropy and captures node feature evolution using sequential node mutual information through the Donsker-Varadhan representation. Based on the `thumbnail`, we further derive effective constraints for representation learning that approximates the mutual information bounds between the data, tasks, and `thumbnail`, effectively balancing robustness and expressiveness. Our contributions can be summarized as follows:

• We **model the thumbnail** for temporal graphs that captures their global evolution features through online computation of von Neumann entropy and alignment with node feature mutual information.

• We **formulate optimization constraints** based on the evolutionary information embedded in the thumbnail, guiding the learning process to enhance model capability and robustness.

• We **conduct comprehensive empirical evaluations** across multiple benchmark datasets. The results demonstrate that TGT achieves statistically significant improvements in performance compared to baselines. The results highlight the effectiveness of our method characterized by substantial evolution constraint, as well as its robustness in handling systematic noise.

## 2 PROBLEM STATEMENT

### 2.1 THUMBNAIL OF GRAPH SEQUENCE

We aim to mine a specific graph that captures the evolutionary skeleton of a temporal graph, encapsulating the global evolution features of the graph sequence. Considering a temporal graph $\mathcal{G} = \{G^i\}_{i=1}^T$ characterized by a historical sequence of snapshots (notably, TGT focuses primarily on discrete-time dynamic graphs), our method models a static `thumbnail` $\mathcal{G}_T = \{V_{\mathcal{G}_T}, E_{\mathcal{G}_T}\}$, which is designed to distill and encapsulate the essential evolutionary information of the entire temporal process within $\mathcal{G}$. The corresponding adjacency matrix $\mathbf{M}$ is composed of $M_{\alpha\beta} = 1$ if $(\alpha, \beta) \in E_{\mathcal{G}T}$, and $M_{\alpha\beta} = 0$ otherwise. Based on sufficient temporal correlations, we construct a set of assignment matrices, denoted as $\mathcal{S} = \{S^1, S^2, \ldots, S^T\}$, which map $V^i$ in $G^i$ to $V_{\mathcal{G}_T}$ via the function $\mathcal{F}$. $s_{a\alpha}^i$ in $S^i$ denotes the mapping from node $a$ in $G^i$ to node $\alpha$ in $\mathcal{G}_T$ at timestamp $i$.

$$s_{a\alpha}^i = \begin{cases} 1 & \text{if } \mathcal{F}(v_a^i) = \alpha, \\ 0 & \text{otherwise.} \end{cases} \tag{1}$$

To estimate the evolution of the graph sequence $\mathcal{G}$, the posterior probability $P(\mathcal{G} \mid \mathcal{G}_T, \mathcal{S})$ can be formulated as a probability distribution function in aggregate (Han et al., 2015). Due to the temporal relationship between snapshots, the joint distribution cannot be simply computed by multiplying independent distributions. Instead, we model the joint probability as a weighted conditional distribution over each snapshot and thumbnail, parameterized by learnable coefficients $B_a^i$.

$$P(\mathcal{G} \mid \mathcal{G}_T, \mathcal{S}) = \prod_{G^i \in \mathcal{G}} \left[ P\left(G^i \mid \mathcal{G}_T, S^i\right) \right] = \prod_{G^i \in \mathcal{G}} \left[ \prod_{a \in V^i} \sum_{\alpha \in V_{\mathcal{G}_T}} K_a^i \exp\left(\mu \sum_{b \in V^i} \sum_{\beta \in V_{\mathcal{G}_T}} A_{ab}^i M_{\alpha\beta} S_{b\beta}^i\right) \right],$$

$$\text{where} \quad \mu = \ln\frac{1 - P_e}{P_e}, \quad K_a^i = P_e^{|V^i| \cdot |V_{\mathcal{G}_T}|} B_a^i. \tag{2}$$

$P_e$ is the probability of a matching error between observed graph vertices and the corresponding vertices in `thumbnail`, while $B_a^i$ is the sampled probability computed at time step $i$ based on the feature $x_a^i$ of vertex $a$ (Han et al., 2015) (more details for Eq. 2 in appendix A.2). The aforementioned conditional likelihood estimation is applicable to both directed and undirected graphs and provides an effective representation of the temporal graph $\mathcal{G}$, assuming the thumbnail is accurate.

### 2.2 REPRESENTATION LEARNING WITH THUMBNAIL

The problem we examine involves a temporal graph $\mathcal{G}$, where the graph stream comprises snapshots $G^t$ at each timestamp $t$, up to a maximum timestamp $T$. Each snapshot $G^t = \{V^t, E^t\}$ consists of vertices $V^t = \{v_1^t, \cdots, v_n^t\}$ of size $n$ and edges $E^t \subseteq V^t \times V^t$ of size $m$ at timestamp $t$. The vertices $V^t$ possess $k$-dimensional features represented by $\mathbf{X}^t \in \mathbb{R}^{n \times k}$. The adjacency matrix $\mathbf{A}^t = \{a_{ij}\}^{n \times n}$ is defined such that $a_{ij}^t = 1$ if $(i, j) \in E^t$ and $a_{ij}^t = 0$ otherwise. We aim to input the temporal graph $\mathcal{G}$ and utilize the node features $\mathbf{X}^{1:T}$ along with the topological structure $\mathbf{A}^{1:T}$ to derive the node embedding $\mathbf{Z}^{T+1} \in \mathbb{R}^{n \times k}$ for the subsequent time step, thereby facilitating downstream tasks. We aim to develop the robust representation $\mathbf{Z}^{T+1}$, particularly under conditions where node and structural features are sparse, unreliable, or perturbed. To meet these requirements, the representation should adhere to the principle of *minimal sufficiency* (Tishby & Zaslavsky, 2015):

$$\mathbf{IB_Z} = \arg\min_{\mathbf{Z}}\{-I(\mathbf{Y}; \mathbf{Z}) + \beta I(\mathcal{G}; \mathbf{Z})\}, \tag{3}$$

where $\beta$ is the Lagrange multiplier to balance the two mutual information $I$. During optimization, the mutual information between $\mathbf{Z}$ and the target label $\mathbf{Y}$ is maximized to enhance the representation capability of the model, while the mutual information between $\mathbf{Z}$ and the input data $\mathcal{G}$ is constrained to improve the model's robustness. By instantiating the intermediate constraint $Z$ through the `thumbnail` (as shown in Eq. 8), we effectively capture the evolution of topological dependencies in historical graph sequences, thereby improving representation quality and robustness for downstream tasks such as node classification and link prediction.

## 3 TEMPORAL GRAPH THUMBNAIL

In this section, we introduce our Temporal Graph Thumbnail in detail. Our framework consists of two components: *(i)* an approach for modeling the `thumbnail` that effectively captures global evolutionary features, and *(ii)* the optimization constraints for representation learning guided by the `thumbnail`. Finally, we instantiate training and inference processes to illustrate the full pipeline.

### 3.1 THUMBNAIL MODELING FROM TEMPORAL GRAPH

Existing methods fail to capture global evolutionary features, primarily due to the absence of an explicit model that characterizes the core evolutionary backbone underlying temporal graph sequences (Liu et al., 2023). This oversight weakens the temporal coherence between individual snapshots and the sequence as a whole, rendering such methods less robust to real-world perturbations. To this end, network entropy has become a widely adopted tool to characterize the structural dynamics of complex systems (Anand et al., 2014). Among various forms of network entropy, von Neumann graph entropy (VNGE) exhibits distinctive mathematical properties that make it well-suited for modeling the `thumbnail` (The reasons for choosing VNGE are detailed in the appendix A.1.3).

Inspired by VNGE's ability to coherently track temporal topological changes (Huang et al., 2023a) and encode evolving structural features (Liu et al., 2022; Alstott et al., 2015), we characterize the structural evolution by analyzing the mutual information between the `thumbnail` $\mathcal{G}_T$ and the temporal graph $\mathcal{G} = \{G^1, \ldots, G^T\}$ from the perspective of VNGE. We start by preparing the approximation $H_{VN}$, adapted from Ye et al. (2014), to characterize structural evolution.

$$H_{VN} = 1 - \frac{1}{|V_{\mathcal{G}_T}|} - \frac{1}{2|V_{\mathcal{G}_T}|^2} \left\{ -\sum_{(\alpha,\beta) \in E_{\mathcal{G}_{T_1}}} \frac{1}{d_\beta^{\text{out}} d_\alpha^{\text{out}}} + \sum_{(\alpha,\beta) \in E_{\mathcal{G}_T}} \left( \frac{d_\alpha^{\text{in}}}{d_\beta^{\text{in}} d_\alpha^{\text{out2}}} + \frac{1}{d_\beta^{\text{out}} d_\alpha^{\text{out}}} \right) \right\},$$

$$\text{where} \quad d_\alpha^{\text{in}} = \sum_{\gamma \in V_{\mathcal{G}_T}} M_{\gamma\alpha}, \quad d_\alpha^{\text{out}} = \sum_{\gamma \in V_{\mathcal{G}_T}} M_{\alpha\gamma}. \tag{4}$$

The set $E_{\mathcal{G}_{T_1}}$ represents the asymmetric directed edges of $\mathcal{G}_T$, where $E_{\mathcal{G}_{T_1}} = \{(u,v) \mid (u,v) \in E_{\mathcal{G}_T} \text{ and } (v,u) \notin E_{\mathcal{G}_T}\}$. Given the intrinsic interdependence in the evolutionary process, snapshots are not independent. Thus, $\mathbf{B}^i$ is a learnable probability weight that quantifies the probabilistic contribution of the snapshot $G^i$, corresponding to the parameter $\mathbf{K}^i$ and $\mathbf{B}^i$ as defined in Eq. 2 (proof in appendix A.2 and A.3.1).

$$I_s(\mathcal{G}_T; \mathcal{G}) = H_{VN}(\mathcal{G}_T) - \sum_{G^i \in \mathcal{G}} \mathbf{B}^i H_{VN}(\mathcal{G}_T | G^i). \tag{5}$$

Then, we use $\mathcal{F}$ to convert the non-Euclidean graph sequence $\mathcal{G}$ into vertex embeddings of `thumbnail`. We employ Donsker-Varadhan representation to derive an estimator for mutual information $I_{DV}$ between $\mathcal{G}$ and $\mathcal{G}_T$ (Belghazi et al., 2018) (proof in appendix A.3.2).

$$I_{DV}(\mathcal{G}_T; \mathcal{G}) = \sup_{f \in \mathcal{F}} \left( \mathbb{E}_P[f(V_\mathcal{G})] - \log \mathbb{E}_Q \left[ e^{f(V_\mathcal{G})} \right] \right). \tag{6}$$

We integrate $I_s(\mathcal{G}_T; \mathcal{G})$, which characterizes the evolution of the **topological structure**, with $I_{DV}(\mathcal{G}_T; \mathcal{G})$, which reflects the evolution of **node features**, to maximize the mutual information between the `thumbnail` and the raw temporal graph. This objective guides the learning of the mapping $\mathcal{F}$ and the coupling weights $\mathbf{B}^i$, as part of definition of the `thumbnail` in Eq. 1 and 2.

### 3.2 THUMBNAIL-GUIDED REPRESENTATION CONSTRAINTS

Existing representation methods are limited in their ability to model the global evolution inherent in temporal graphs, and they neglect the memory patterns and logical dependencies embedded within global temporal changes. With the guidance of the `thumbnail`, we formulate optimization constraints that leverage global evolutionary information to enhance representation learning.

Based on local dependency assumption for static graphs, we extend it to temporal graphs.

**Assumption 1** *Given the relevant data from neighbors within a certain distance range ($k$-hop) and time range ($\Delta t$) for vertex $v$ in temporal graph $\mathcal{G}$, $v$ depends only on these spatiotemporal neighbors $\mathcal{N}_{k,\Delta t}(v)$ and is independent of data from the rest of graph:*

$$P(x_v^t | \mathcal{G}^{1:t}) = P(x_v^t | \mathcal{G}_{\mathcal{N}_{k,\Delta t}(v)}^{t-\Delta t:t}). \tag{7}$$

Under this premise, we derive the variational bound of the mutual information. Constrained by the `thumbnail` $\mathcal{G}_T$, we refine the temporal graph $\mathcal{G}$ the principle of *minimal sufficiency* (Tishby & Zaslavsky, 2015):

$$\mathcal{G}_{TIB} = \arg\min_{\mathcal{G}_T}\{-I(\mathbf{Y};\mathcal{G}_T) + \beta I(\mathcal{G};\mathcal{G}_T)\}. \tag{8}$$

Our objective is to employ the `thumbnail` $\mathcal{G}_T$ as a bottleneck constraint to minimize redundancy in the original data while preserving sufficient evolutionary information to prevent distortion from excessive compression. Minimality is achieved by minimizing the upper bound of mutual information $I(\mathcal{G};\mathcal{G}_T)$ to compress the redundancy of $\mathcal{G}$. Sufficiency is achieved by maximizing the lower bound of mutual information $I(\mathbf{Y};\mathcal{G}_T)$ to capture essential features of $\mathcal{G}$.

**Upper Bound of $I(\mathcal{G};\mathcal{G}_T)$**   Inspired by GIB (Wu et al., 2020), we extend the variational approximation to the mutual information between the `thumbnail` and the original data, termed the Thumbnail Bottleneck (TB) constraint. Specifically, $S_{\mathcal{G}_{TX}}, S_{\mathcal{G}_{TA}} \subset [L]$ is the index sequence of the temporal graph snapshot, $L$ is the temporal graph time span. To enable tractable optimization, we introduce a variational distribution $\mathbb{Q}(\cdot)$ to approximate the true posterior $\mathbb{P}(\cdot)$.

$$I(\mathcal{G};\mathcal{G}_T) \leq I(\mathcal{G};\{Z_{\mathcal{G}_{TX}}^{(l)}\}_{l \in S_{\mathcal{G}_{TX}}} \cup \{Z_{\mathcal{G}_{TA}}^{(l)}\}_{l \in S_{\mathcal{G}_{TA}}}) \leq \sum_{l \in S_{\mathcal{G}_{TX}}} \text{TB}_{\mathcal{G}_{TX}}^{(l)} + \sum_{l \in S_{\mathcal{G}_{TA}}} \text{TB}_{\mathcal{G}_{TA}}^{(l)},$$

$$\text{where} \quad \text{TB}_{\mathcal{G}_{TA}}^{(l)} = \mathbb{E}\left[\log \frac{\mathbb{P}(Z_{\mathcal{G}_{TA}}^{(l)} | \mathbf{A}, Z_{\mathcal{G}_{TA}}^{(l-1)})}{\mathbb{Q}(Z_{\mathcal{G}_{TA}}^{(l)})}\right], \text{TB}_{\mathcal{G}_{TX}}^{(l)} = \mathbb{E}\left[\log \frac{\mathbb{P}(Z_{\mathcal{G}_{TX}}^{(l)} | Z_{\mathcal{G}_{TX}}^{(l-1)}, Z_{\mathcal{G}_{TA}}^{(l)})}{\mathbb{Q}(Z_{\mathcal{G}_{TX}}^{(l)})}\right], \tag{9}$$

$Z_{\mathcal{G}_{TX}}^{(l)}$ and $Z_{\mathcal{G}_{TA}}^{(l)}$ are $\mathcal{G}_T$'s embeddings that capture temporal evolution information. This helps resist perturbation attacks that violate inherent evolutionary patterns (proof in appendix A.3.3).

**Lower Bound of $I(\mathbf{Y};\mathcal{G}_T)$**   The variational lower bound of $I(\mathbf{X};\mathcal{G}_T)$ follows the approximation:

$$I(\mathbf{Y};\mathcal{G}_T) \geq 1 + \mathbb{E}_{P(\mathbf{Y},\mathcal{G}_T)}[\log \frac{P\left(\mathbf{Y} | (Z_{\mathcal{G}_{TX}}^{(l)}, Z_{\mathcal{G}_{TX}}^{(l)})\right)}{Q(\mathbf{Y})}] - \mathbb{E}_{P(\mathbf{Y})}[\frac{\mathbb{E}_{P(\mathcal{G}_T)} P\left(\mathbf{Y} | (Z_{\mathcal{G}_{TX}}^{(l)}, Z_{\mathcal{G}_{TX}}^{(l)})\right)}{Q(\mathbf{Y})}]. \tag{10}$$

$Z_{\mathcal{G}_{TX}}^{(l)}$ and $Z_{\mathcal{G}_{TA}}^{(l)}$ are determined in Eq. 1. In practice, reparameterizing the probability distribution function $P$ proves beneficial by exploiting prior data on the conditional probability distribution. This approach eliminates learning a mapping between $\mathcal{G}_T$ and $\mathbf{Y}$, which requires instead the learning of an approximate probability distribution $Q(\mathbf{Y})$ in a typically lower-dimensional vector space (proof in appendix A.3.4).

## 3.3 Instantiation of Training and Inference

We instantiate the previously introduced probability distributions and elaborate on the detailed pipeline of the framework. We illustrate the process using link prediction as an example, with the full pipeline presented as pseudocode in the appendix A.1.4.

**Thumbnail Modeling**   For the thumbnail modeling, we adopt the mutual information alignment method to design the loss function as follows:

$$\mathcal{L}_{evolution} = -\left(I_s(\mathcal{G}_T;\mathcal{G}) + I_{DV}(\mathcal{G}_T;\mathcal{G})\right). \tag{11}$$

The two terms in the formula are composed of Eq. 5 and Eq. 6 respectively. We define $G^i$ in Eq. 5 as a graph snapshot within the temporal neighborhood $[t - \Delta t, t]$, with the neighborhood size configurable as a hyperparameter. Furthermore, $\mathcal{F}$ in Eq. 6 is specified as neural network layers composed of learnable parameters.

**Instantiation of $I(\mathcal{G}; \mathcal{G}_T)$** To instantiate the node feature term $\text{TB}^{(l)}_{\mathcal{G}_{TX}}$ as defined in Eq. 9, we set $\mathbb{Q}(Z^{(l)}_{\mathcal{G}_{TX}})$ to a mixture of Gaussians, expressed as $\sum_{k=1}^{m} \pi_k \Phi(\mu_{q,k}, \sigma^2_{q,k})$. Here, $\pi_k$, $\mu_{q,k}$ and $\sigma_{q,k}$ are learnable parameters. We set $\mathbb{P}(Z^{(l)}_{\mathcal{G}_{TX}} | Z^{(l-1)}_{\mathcal{G}_{TX}}, Z^{(l)}_{\mathcal{G}_{TA}})$ as $\Phi(Z^{(l)}_{\mathcal{G}_{TX}}; \mu_p, \sigma^2_p)$, and then:

$$\text{TB}^{(t)}_{\mathcal{G}_{TX}} \doteq \sum_{v \in V^t} \left( \Phi(Z^{(t)}_{\mathcal{G}_{TX}}; \mu_p, \sigma^2_p) - \sum_{k=1}^{m} \Phi(\mu_{q,k}, \sigma^2_{q,k}) \right). \tag{12}$$

For structural term $\text{TB}^{(l)}_{\mathcal{G}_{TA}}$ as defined in Eq. 9, we similarly set $\mathbb{Q}(Z^{(l)}_{\mathcal{G}_{TA}})$ as Bernoulli distribution, $\text{Bernoulli}(\phi)$, where $\phi$ is a hyperparameter. Neighbor information is then aggregated by sampling from this Bernoulli distribution. The generative distribution $\mathbb{P}(Z^{(l)}_{\mathcal{G}_{TA}} | \mathbf{A}, Z^{(l-1)}_{\mathcal{G}_{TA}})$ is similarly Bernoulli, with sampling probability $\alpha_p$ computed from the node embeddings.

$$\text{TB}^{(t)}_{\mathcal{G}_{TA}} \doteq D_{KL}(\text{Bernoulli}(\alpha^t_p) \,\|\, \text{Bernoulli}(\phi)) = (1 - \alpha^t_p) \log \frac{1 - \alpha^t_p}{1 - \phi} + \alpha_p \log \frac{\alpha^t_p}{\phi}. \tag{13}$$

**Instantiation of $I(\mathbf{Y}; \mathcal{G}_T)$** Instantiating Eq. 10 depends on the downstream task. For instance, in link prediction the likelihood term $P\left(\mathbf{Y} \mid Z^{(L)}_{\mathcal{G}_{TX}}, Z^{(L)}_{\mathcal{G}_{TA}}\right)$ is modeled as a categorical distribution, where $Q(\mathbf{Y})$ denotes the empirical distribution of ground truth links. During optimization, the final expected joint likelihood term converges toward unity, effectively canceling the constant offset.

$$I(\mathbf{Y}; \mathcal{G}_T) \doteq \frac{1}{N} \sum_{i=1}^{N} \log \left[ \text{Categorical}(\mathbf{Y} | Z^{(l)}_{\mathcal{G}_{TX}}) \right] =: -\mathcal{L}_{CE}(\mathbf{w}_{out} \cdot \mathbf{Z}_{\mathcal{G}_{TX}}, \mathbf{Y}), \tag{14}$$

$\mathbf{w}_{out}$ denotes the weights of the downstream classifier. During optimization, this component refers to the calculation of cross-entropy loss, where $N$ represents the number of training samples.

In summary, the mutual information loss function for bottleneck constraints imposed by the `thumbnail` is as follows, where $S_A, S_X$ are the index sets satisfying Assumption 1.

$$\mathcal{L}_B = -I(\mathbf{Y}; \mathcal{G}_T) + \beta I(\mathcal{G}; \mathcal{G}_T) \doteq -I(\mathbf{Y}; \mathcal{G}_T) + \beta \left[ \sum_{t \in S_A} \text{TB}^{(t)}_{\mathcal{G}_{TA}} + \sum_{t \in S_X} \text{TB}^{(t)}_{\mathcal{G}_{TX}} \right]. \tag{15}$$

The overall training objectives of the proposed model can be reformulated as follows:

$$\mathcal{L} = \mathcal{L}_B + \lambda \cdot \mathcal{L}_{evolution}, \tag{16}$$

where $\lambda$ is the hyperparameter of the Lagrange multiplier. We adopt the Graph Attention Network (Veličković et al., 2018) as the backbone framework, meaning both $\mathcal{F}$ in Eq. 6 and $P_e$ in Eq. 2 are GAT-based encoders. The probability $B^t_a$ in Eq. 2 is derived by transforming node $a$'s encoding at time $t$ through a feedforward layer with a softmax. Therefore, according to Eq. 2, we can predict global evolution using the `thumbnail` $\mathcal{G}_T$, which is compressed by graph sequence $\mathcal{G}$.

## 4 EXPERIMENTS

We employ link prediction as a downstream task to assess the capability and robustness of TGT compared to established baselines. Specifically, we aim to address the following research questions: **Q1 (Fundamental Capability)**: How does TGT compare with SOTA methods in terms of fundamental capability? Ans 1. **Q2 (Robustness)**: How robust is TGT under various types of perturbations or noise against other baselines? Ans 2.1, 2.2, 2.3. **Q3 (Effectiveness of the Thumbnail)**: How effective is the `thumbnail` in modeling evolution? Ans 3.1. Why is von Neumann Graph Entropy (VNGE) chosen for modeling `thumbnail`? Ans 3.2. **Q4 (Contribution of Constraints)**: How do the constraints guided by the `thumbnail` contribute to enhancing the model's robustness? Ans 4..

### 4.1 EXPERIMENTAL SETUPS

Table 2: Inductive link predication performance on the Bitcoin, MathOverflow and MOOC datasets under clean settings.

| Model | | Bitcoin | | MathOverflow | | MOOC | |
|---|---|---|---|---|---|---|---|
| | | AUC | AP | AUC | AP | AUC | AP |
| *Dynamic GNNs (DGNNs)* | EvolveGCN | 67.59±0.3 | 63.38±0.2 | 75.59±0.2 | 72.73±0.4 | 72.35±0.3 | 73.59±0.2 |
| | JODIE | 74.47±0.3 | 75.50±0.4 | 67.06±1.2 | 66.32±0.6 | 73.19±0.7 | 71.78±0.9 |
| | DyREP | 70.43±0.5 | 69.79±0.4 | 63.50±0.5 | 63.37±0.6 | 81.36±0.1 | 78.35±0.3 |
| | TGN | 69.36±1.1 | 72.09±0.7 | 64.50±0.6 | 65.88±0.6 | 79.36±1.0 | 78.96±0.5 |
| *Robust Gener-alized DGNNs* | DIDA | 73.57±0.3 | 71.27±0.4 | 74.37±0.4 | 75.24±0.3 | 89.84±0.5 | 88.49±0.4 |
| | GIB+LSTM | 70.79±0.5 | 69.73±0.4 | 77.52±0.3 | 75.03±0.7 | 92.34±0.3 | 93.29±0.5 |
| | DGIB | 72.99±1.3 | 73.24±0.6 | 80.29±0.3 | 79.99±0.5 | 93.06±0.1 | 93.11±0.3 |
| | **TGT(ours)** | **91.41±0.2** | **91.01±0.3** | **82.38±0.6** | **81.17±0.4** | **95.42±0.4** | **94.56±0.7** |

**Datasets**  We conduct experiments on the link predic-tion task on three widely-used datasets. MathOverflow is a forum question-answer relationship data (Paranjape et al., 2017); Bitcoin dataset is a user transaction relation-ship data consisting of transaction records (Kumar et al., 2016; 2018); the MOOC dataset represents the actions taken by users on a popular MOOC platform (Kumar et al., 2019). More detailed information is shown in Ta-ble 1. *Evolution Frequency* refers to how often the graph evolves per day. Avg. Snap.nodes/edges represents the

Table 1: Details of datasets for experi-ments.

| Dataset | Bitcoin | MathOverflow | MOOC |
|---|---|---|---|
| Nodes | 9,664 | 24,818 | 7,047 |
| Edges | 59,778 | 506,550 | 411,749 |
| Timespan | 1903 days | 2350 days | 30 days |
| Link Type | Homogeneous | 3 | 5 |
| *Evolution Frequency* | 18.7 | 45.78 | 13724.97 |
| Avg. Snap. nodes | 7,034 | 21,683 | 7,047 |
| Avg. Snap. edges | 51,363 | 207,581 | 81,749 |
| Snap. span | 60 days | 20 days | 12 hours |

average number of nodes/edges in each snapshot, where the snap time span refers to the temporal scale at which the graph is chronologically split. Datasets of varying scales and evolution frequen-cies covering most real-world tasks. Detailed descriptions of datasets are provided in appendix A.4.

**Baselines**  We selected baselines from dual perspectives to demonstrate the effectiveness of TGT. Among temporal graph representation methods, we selected four superior methods. TGN (Rossi et al., 2020) is a framework for representation learning on streaming temporal graphs; JODIE (Ku-mar et al., 2019) learns node interaction relationships using RNNs; DyREP (Trivedi et al., 2019) en-codes nodes by modeling temporal point processes at dual scales; EvolveGCN (Pareja et al., 2020) captures graph sequence dynamics by evolving GCN parameters. Among robustness and bottleneck constraint, we selected three methods. DIDA (Zhang et al., 2022) leverages robust and generalized prediction patterns on temporal graph representation; GIB (Wu et al., 2020) computes the informa-tion bottleneck of graph snapshots, with LSTM adapted for dynamic scenes; DGIB (Yuan et al., 2024) imposes history constraints on the information bottleneck to obtain representations.

**Data Perturbation**  To verify TGT's robustness , we adopted adversarial attacks on training data in multiple aspects. *a)* **Feature Interference**: To assess the robustness of TGT under untargeted attacks, we introduce random Gaussian noise to node features and apply varying perturbation in-tensities (controlled by noise amplitude). *b)* **Structural Interference**: For topological attacks, we employ the **Nettack** (Zügner et al., 2018) to perform adversarial edge perturbations (e.g. deletion or negative sampling), specifically targeting substructures that maximize prediction loss in the proxy model. *c)* **Temporal Interference**: To evaluate TGT's stability and robust representation capabili-ties under temporal disruptions, we disrupt the chronological order of graph snapshots by randomly permuting them, simulating interference with the underlying temporal evolution dynamics (For more detailed information of attack settings, please refer to the appendix A.6).

## 4.2 EFFECTIVENESS EXPERIMENT

To address **Q1**, we conducted comprehensive experimental analyses on three distinct datasets, and subsequently compared the results with established baselines to rigorously demonstrate the superior performance of TGT. Implementation details of TGT are provided in appendix A.5. In the effec-tiveness experiment, we employed a clean data setting without any data interference and utilized inductive link prediction as the downstream task for the evaluation of representations. Drawing upon the original data, we conducted negative sampling by selecting 10% of the total edges to facil-itate the training of the link prediction task. After repeating the experiments three times, the average and range of the results are presented in the Table 2.

Table 3: Robustness results (**AUC**) on Bitcoin, MathOverflow and MOOC datasets with data perturbation at different levels.

| Dataset | Model | Clean | Feature Interference | | | Structure Interference | | | Temporal Interference | | |
|---|---|---|---|---|---|---|---|---|---|---|---|
| | | | 10% | 20% | 50% | 5% | 10% | 20% | $n=1$ | $n=2$ | $n=5$ |
| Bitcoin | EvolveGCN | 67.59±0.3 | 62.74±0.2 | 56.77±0.2 | 54.24±0.2 | 64.01±0.3 | 60.37±0.4 | 55.89±0.3 | 65.47±0.3 | 63.12±0.3 | 54.37±0.4 |
| | JODIE | 74.47±0.3 | 69.28±0.1 | 61.10±0.3 | 53.32±0.4 | 70.33±0.2 | 66.82±0.3 | 60.57±0.3 | 71.54±0.8 | 69.79±0.5 | 58.28±0.7 |
| | DyREP | 70.43±0.5 | 64.91±0.3 | 61.25±0.3 | 56.63±0.2 | 69.98±0.8 | 63.79±0.7 | 58.60±0.5 | 69.43±0.8 | 65.84±0.5 | 57.33±0.7 |
| | TGN | 69.36±1.1 | 67.34±0.6 | 62.31±0.4 | 59.06±1.3 | 66.61±0.5 | 62.18±0.6 | 58.73±0.5 | 68.46±0.8 | 66.92±0.7 | 61.74±0.8 |
| | DIDA | 73.57±0.3 | 71.05±0.2 | 68.43±0.3 | 64.20±0.2 | 70.93±0.2 | 68.71±0.3 | 65.69±0.4 | 71.23±0.2 | 69.65±0.3 | 64.29±0.4 |
| | GIB+LSTM | 70.79±0.3 | 69.26±0.3 | 63.73±0.5 | 58.37±0.3 | 68.61±0.5 | 65.09±0.3 | 62.93±0.2 | 69.15±0.3 | 67.17±0.6 | 63.41±0.8 |
| | DGIB | 72.99±1.3 | 69.92±0.6 | 63.63±0.5 | 60.78±0.7 | 70.13±0.3 | 65.84±0.6 | 59.13±0.4 | 71.27±0.5 | 68.44±0.7 | 62.53±0.6 |
| | **TGT** | **91.41±0.2** | **89.43±0.5** | **86.23±0.4** | **80.62±0.4** | **89.83±0.6** | **85.00±0.5** | **80.95±0.7** | **90.78±0.5** | **88.14±0.6** | **85.64±0.3** |
| MathOverflow | EvolveGCN | 75.59±0.2 | 67.22±0.5 | 61.14±0.3 | 54.79±0.2 | 66.68±0.5 | 63.11±0.4 | 55.23±0.3 | 69.39±0.3 | 67.90±0.3 | 56.63±0.5 |
| | JODIE | 67.06±1.2 | 63.23±0.3 | 59.56±0.3 | 51.92±0.2 | 64.71±0.4 | 59.13±0.3 | 53.19±0.3 | 66.34±0.3 | 63.93±0.2 | 55.37±0.3 |
| | DyREP | 63.50±0.5 | 59.13±0.3 | 53.32±0.6 | 54.17±0.7 | 60.19±0.2 | 56.59±0.2 | 53.26±0.3 | 61.71±0.4 | 60.07±0.2 | 53.33±0.5 |
| | TGN | 64.50±0.6 | 61.22±0.3 | 58.96±0.3 | 55.14±0.4 | 59.49±0.2 | 56.23±0.3 | 53.97±0.3 | 61.40±0.3 | 60.36±0.5 | 55.23±0.7 |
| | DIDA | 74.37±0.4 | 70.95±0.1 | 68.63±0.1 | 63.98±0.1 | 73.48±0.2 | 70.47±0.3 | 67.03±0.4 | 73.57±0.2 | 69.09±0.1 | 65.44±0.5 |
| | GIB+LSTM | 77.52±0.3 | 73.24±0.1 | 69.38±0.6 | 63.07±0.7 | 75.83±0.3 | 68.37±0.6 | 63.21±0.8 | 75.73±0.8 | 71.14±0.5 | 63.33±0.8 |
| | DGIB | 80.29±0.3 | 78.54±0.2 | 73.63±0.5 | 68.98±0.3 | 77.87±0.2 | 74.47±0.3 | 70.43±0.3 | 79.66±0.2 | 77.75±0.3 | 70.24±0.5 |
| | **TGT** | **82.38±0.6** | **79.22±0.3** | **74.37±0.3** | **71.42±0.2** | **80.01±0.3** | **77.74±0.3** | **75.93±0.7** | **81.57±0.2** | **80.71±0.5** | **76.37±0.4** |
| MOOC | EvolveGCN | 72.35±0.3 | 66.23±0.4 | 62.37±0.2 | 54.72±0.2 | 68.83±0.1 | 59.29±0.1 | 52.31±0.2 | 66.98±0.3 | 62.57±0.2 | 55.93±0.2 |
| | JODIE | 73.19±0.7 | 63.15±0.2 | 55.36±0.2 | 53.71±0.4 | 69.15±0.4 | 61.42±0.2 | 57.34±0.4 | 70.69±0.2 | 63.25±0.3 | 58.59±0.4 |
| | DyREP | 81.36±0.1 | 74.60±0.2 | 66.77±0.2 | 60.52±0.3 | 76.43±0.2 | 68.31±0.3 | 59.74±0.3 | 78.56±0.6 | 65.39±0.5 | 59.37±0.5 |
| | TGN | 79.36±1.0 | 73.63±0.4 | 68.41±0.3 | 60.03±0.3 | 75.31±0.2 | 69.27±0.2 | 61.33±0.3 | 78.29±0.7 | 72.68±0.8 | 63.53±1.1 |
| | DIDA | 89.84±0.5 | 79.73±0.1 | 73.62±0.3 | 64.01±0.3 | 84.48±0.2 | 71.47±0.3 | 61.03±0.4 | 86.53±0.2 | 82.79±0.3 | 73.66±0.2 |
| | GIB+LSTM | 92.34±0.3 | 73.21±0.3 | 65.25±0.3 | 63.67±0.2 | 83.05±0.2 | 82.64±0.4 | 74.37±0.3 | 89.95±0.7 | 85.76±0.4 | 69.68±0.7 |
| | DGIB | 93.06±0.1 | 84.35±0.1 | 75.24±0.2 | 63.32±0.3 | 87.75±0.1 | 84.27±0.7 | 79.69±0.3 | 91.53±0.2 | 85.36±0.3 | 70.32±0.6 |
| | **TGT** | **95.42±0.4** | **88.73±0.2** | **80.79±0.1** | **71.68±0.3** | **90.43±0.3** | **86.16±0.5** | **81.82±0.3** | **92.61±0.2** | **89.53±0.7** | **84.01±0.6** |

**Ans 1.** TGT consistently outperforms the baselines in fundamental representation capability across all three datasets, with its advantages particularly pronounced in scenarios with sparse neighbor information. By analyzing the datasets and the experimental results, it is evident that TGT exhibits significant advantages in the Bitcoin dataset. Specifically, its AUC and AP values have improved by 16. 94% and 15. 51%, respectively, compared to the runner-up. This improvement suggests that TGT can capture data features more effectively and demonstrate superior generalization capabilities. On MathOverflow, where social network interaction noise is prevalent, *Robust Generalized DGNNs* significantly outperform *DGNNs*. Furthermore, on the MOOC dataset, characterized by frequent evolutionary changes, TGT displays a clear performance edge. These results demonstrate the effectiveness of TGT's design, which leverages von Neumann entropy to constrain the information bottleneck associated with global structural evolution.

## 4.3 ROBUSTNESS EXPERIMENT

To address **Q2**, we evaluate the robustness of the model through data perturbation. Following the experimental settings outlined in the preceding subsection, we intentionally injected anomalies into the training dataset and conducted a series of experiments to evaluate the model's robustness along three distinct dimensions: node features, topological structures, and temporal correlations. After repeating three times, the average and range of the results are shown in Table 3.

**Feature Interference   Ans 2.1.**   TGT can effectively resist feature interference with different noise levels. Gaussian noise perturbation of node features significantly diminishes the performance of ordinary DGNNs. Comparatively, robust generalized DGNNs exhibited superior performance. Given limited node features in Bitcoin and MathOverflow datasets, DIDA, GIB+LSTM, and DGIB maintained efficacy, yet their robustness lagged significantly behind TGT on the MOOC dataset. TGT employs `thumbnail` to establish feature's constraint (Eq. 12) and thoroughly filters noise, thereby enhancing the model's robustness during optimization.

**Structure Interference   Ans 2.2.**   TGT demonstrates strong robustness against structural-level noise, maintaining stable performance under structural perturbations. Structural perturbations impact all baselines evenly, and structural noise is equally applied to neighboring samples across methods. Under interference, continuous DGNNs (e.g., TGN, DyREP) excel at online processing of abnormal edges, demonstrating enhanced robustness. TGT employs `thumbnail` to establish structure constraint (Eq. 13), effectively mitigating the interference of abnormal edges in crucial topological relationships between modeling nodes. As defined in Eq. 4, the von Neumann entropy captures richer topological information by inherently encoding structural dependencies, which provides TGT with a distinct advantage over existing SOTA robustness methods.

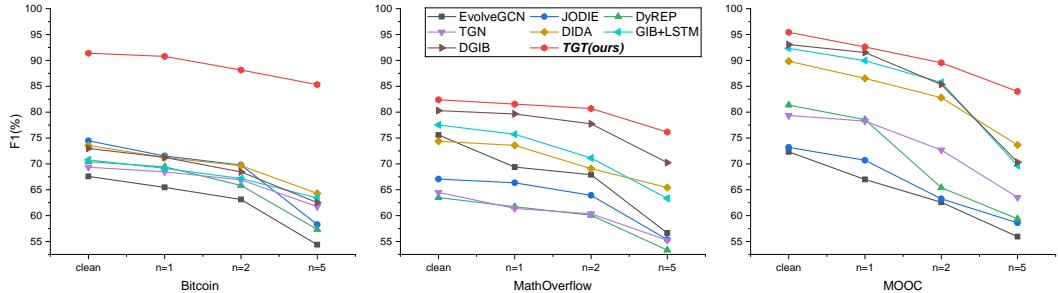

Figure 1: The accuracy curves of the algorithms after being subjected to different degrees of temporal interference on three datasets. By constraining the evolution process using von Neumann entropy, TGT is more resistant to temporal interference.

**Temporal Interference**   **Ans 2.3.** TGT exhibits strong robustness against temporal interference, with consistently superior performance compared to baselines. Temporal perturbations significantly impact evolving patterns in temporal graphs. The robustness evaluation results across different datasets are presented in Fig. 1. On the slowly evolving Bitcoin dataset, temporal interference minimally affects each method, with accuracy decline primarily attributed to feature and topology redundancy stemming from nodes and edges. In the MathOverflow dataset with extensive evolution, redundant historical information impedes model accuracy improvement. In the rapidly evolving MOOC dataset, temporal noise presents a more formidable challenge to the robustness of the model. EvolveGCN, for evolution modeling, is highly sensitive to temporal interference. DIDA, designed for data distribution shifts, excels in anti-interference. TGT demonstrates competitive performance across three datasets. TGT utilizes von Neumann entropy to model graph evolution trajectory(Eq. 5) enables effective representation generation despite severe temporal correlation disturbances.

## 4.4   ABLATION STUDY

To address **Q3**, we choose to use normalized node degree instead of von Neumann entropy in Eq. 11 as the baseline of ablation as *W/O VNGE*.We also added an ablation control without `thumbnail` *W/O T*, where the term $I_s(\mathcal{G}_T; \mathcal{G})$ in Eq. 11 is set to zero entirely. **Ans 3.1.** The `thumbnail` is essential for guiding the denoising process, removing this component leads to substantial degradation in the model's fundamental accuracy and robustness. As shown in Fig. 2, the dataset shows obvious differentiation of ablation effects, which is helpful to analyze the effectiveness of `thumbnail`.

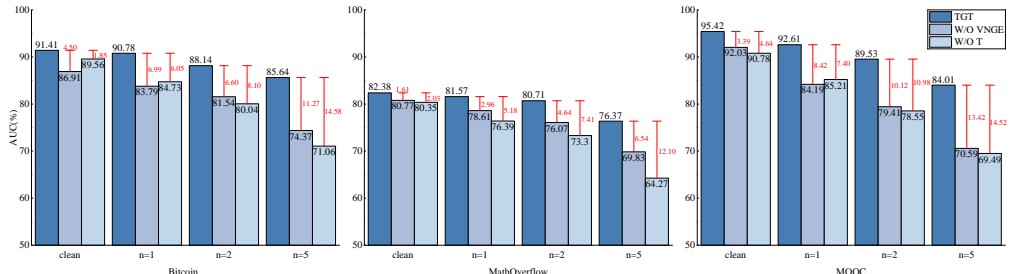

Figure 2: Robust performance of TGT on temporal interference on various datasets under different temporal evolution constraint ablations

With `thumbnail`'s guidance, the model outperforms the setting that ignores evolution, whether using basic node degree information or the proposed VNGE. **Ans 3.2.** The VNGE-based evolution modeling significantly outperforms the one using other methods, demonstrating the effectiveness of our core innovation. It is confirmed by the detailed ablation in the appendix A.7. To address **Q4**, we conduct extensive ablation experiments to verify the roles of derived constraints (Eqs. 12 and 13). **Ans 4.** Each Thumbnail-based constraint contributes to the fundamental capabilities and robustness of the model, confirmed in appendix A.8. Furthermore, we conduct hyperparameter sensitivity experiments on the Lagrangian coefficients. The results are shown in the appendix A.9.

## 5 CONCLUSION

In this paper, we introduce a novel framework called **T**emporal **G**raph **T**humbnail (**TGT**). By modeling the `thumbnail` from raw graph sequence based on von Neumann graph entropy and the mutual information of node features, we characterize the evolution skeleton of the temporal graphs to capture the global evolutionary information. Moreover, the `thumbnail` serves as an intermediary, establishing a bottleneck constraint between the original data and the target task to enhance the model's representation over critical information. Experiments validate TGT's superior robustness and generalization compared to other methods. **Limitations and future work**: Although TGT demonstrates stability on datasets with up to one million nodes, thumbnail modeling and real-time von Neumann entropy computation incur additional computational overhead, restricting its scalability to extremely large graphs. To address this, our future work will investigate pre-training strategies for thumbnail modeling to amortize processing time and hardware costs, while exploring the use of thumbnails as prompts in pre-trained temporal graph neural networks to improve model scalability.

### ETHICS STATEMENT

We have reviewed the ICLR Code of Ethics and have ensured full compliance with its guidelines, particularly in terms of data privacy, transparency, and responsible computing practices. The datasets used in the experiments include those in A.4 and A.10, which are widely used.

### REPRODUCIBILITY STATEMENT

To facilitate replication and extension of our work, we provide comprehensive resources: the code is given in Appendix A.1.1, dataset and preprocessing details in Appendix A.4, model hyperparameter configurations in Appendix A.5, and complete derivations and theoretical proofs in Appendix A.3. With the complete set of code, data, preprocessing details, hyperparameter settings, and theoretical derivations provided, the reproducibility of our work is fully ensured.

### ACKNOWLEDGMENT

The authors wish to thank the anonymous reviewers for their helpful comments. This work was partially funded by Xiamen Major Science and Technology Project (No. 3502Z20241028).

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

## A  APPENDIX

### A.1  FREQUENTLY ASKED QUESTIONS (FAQS)

#### A.1.1  CODE AVAILABILITY

To promote transparency and reproducibility, we have included the source code of TGT, along with training and evaluation scripts, in an anonymous repository [1]. We have also included a copy of this repository in the Supplementary Material. The archive also contains the running logs and the hyperparamters for TGT (detailed TGT implementation in appendix A.5). The noise generation methods we adopt are based on DeepRobust (Li et al., 2020) [2], with detailed attacks implementation provided in appendix A.6. We warmly welcome other researchers to reproduce and extend our work.

#### A.1.2  WHAT ARE ADVANTAGES OF TGT?

TGT introduces three key advantages over existing methods for temporal graph representation, specifically addressing their fundamental limitations through a tailored and principled design.

**Superior representation capability with thumbnail.**  Table 2 demonstrates that TGT consistently achieves superior representation capability across various data scenarios. Compared with other baselines, TGT benefits from the additional global evolution information provided by the `thumbnail`, leading to significantly improved representations. Notably, the advantage becomes even more pronounced in scenarios characterized by high randomness and frequent evolution.

---

[1] **Our TGT**: https://github.com/shiwning/TGT
[2] DeepRobust: https://github.com/DSE-MSU/DeepRobust

**Enhanced robustness across various noise conditions.** Table 3 provides strong evidence that TGT exhibits robust performance against various types of noise. With the guidance of the `thumbnail` constraints, TGT consistently outperforms baselines under different noise settings. In particular, TGT demonstrates significantly more robust performance under temporal perturbations.

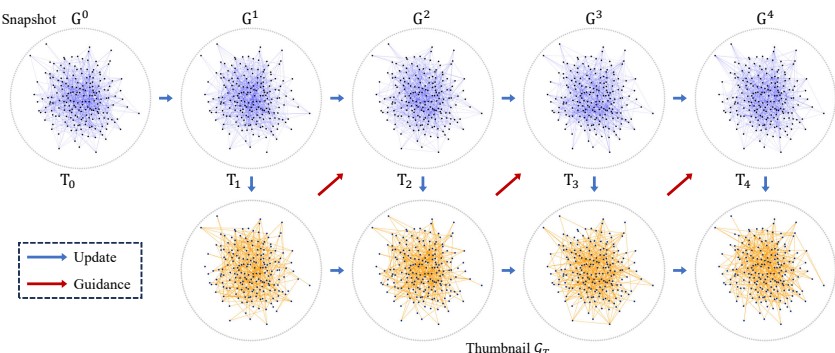

Figure 3: Visualization of the thumbnail modeling.

**Solid and explainable theoretical foundations.** We model the `thumbnail` by computing the von Neumann entropy and mutual information of sequential node features, followed by a principled variational derivation of the representation optimization constraints. Logically complete and rigorous derivations are provided in appendix to validate the theoretical soundness of the TGT. The relevant lemmas and assumptions are well-established and broadly accepted in prior research.

To further illustrate thumbnail clearly, we visualized the modeling process as shown in the Fig. 3. The visualization clearly shows that the thumbnail preserves the persistent and high-frequency structural patterns across time, while filtering out transient or noisy structures. Specially:

• Node connectivity patterns that recur across snapshots are retained with higher edge weights in the thumbnail, indicating that $\mathcal{G}_T$ successfully summarizes the stable evolutionary backbone.

• Ephemeral or low-consistency edges, which appear sporadically in individual snapshots, are largely suppressed, demonstrating the denoising effect induced by the VNGE-based constraint.

As shown in Fig. 3, the thumbnail graph becomes progressively more coherent over iterations, matching the theoretical goal of modeling the global evolution trajectory rather than any single temporal slice. This visualization also confirms, in a human-interpretable way, that the thumbnail effectively captures temporal regularities and structural evolution trends in the datasets, aligning with the conceptual description in Method and Algorithm 1.

### A.1.3 WHY VON NEUMANN ENTROPY? HOW ABOUT OTHER GRAPH ENTROPIES TO MODEL THUMBNAIL?

The von Neumann entropy (VNGE) plays a crucial role in understanding the structural and topological complexity of temporal graphs, as it effectively captures the information content in a manner that aligns with the continuous evolution of time-varying networks (Passerini & Severini, 2009). While other entropy measures, such as Gibbs entropy and Shannon entropy, can also capture structural evolution to some extent (Bianconi, 2009), they are not suitable for continuous computation. These methods typically rely on discrete snapshots, resulting in systemic information loss. More critically, these graph entropy measures do not satisfy the *subadditivity property* (De Domenico & Biamonte, 2016), which poses a challenge when designing local mutual information coupling (as seen in the second term of Eq. 5) because direct weighted accumulation for aggregation is not feasible. This necessitates additional assumptions or more complex modeling approaches. In contrast, VNGE can be computed online and, except in special cases (such as chain graphs or trivial graphs), satisfies the *subadditivity property* (De Domenico & Biamonte, 2016), significantly simplifying the process of capturing structural evolution. Given these considerations, we select VNGE over other graph entropies as a more concise approach for capturing reliable structural evolution.

The robustness of TGT to temporal order disruption stems from the VNGE-based constraint $I_s(\mathcal{G}_T; \mathcal{G})$ (Eq. 5), which fundamentally captures the global structural distribution of snapshots rather than their temporal order.

Specifically, the VNGE is defined over the normalized Laplacian spectrum(Appendix A.3.1), which depends solely on the eigenvalue distribution of the graph. Thus, in constructing the temporal thumbnail $\mathcal{G}_T$, maximizing the mutual information between $\mathcal{G}_T$ and the input sequence $\mathcal{G}$ promotes alignment of their spectral characteristics rather than merely aligning the temporal order of snapshots.

Since spectral quantities are permutation-invariant with respect to node or snapshot order, the learned thumbnail inherently focuses on statistical patterns of structural evolution that remain stable under temporal sequence shuffling (Braunstein et al., 2006). This means the model learns to retain information about how structural complexity evolves (e.g., entropy growth, connectivity changes) rather than the specific temporal sequence of local variations. Therefore, the VNGE-based mutual information regularization provides a temporal-order-agnostic bottleneck, enabling TGT to maintain robustness when snapshot order is permuted or partially corrupted.

Furthermore, as supported by the results in Table 5, TGT maintains consistent performance under temporal perturbation attacks, demonstrating that the VNGE constraint guides the model toward learning a temporal invariant summary that mitigates the impact of sequence tampering.

### A.1.4 COMPUTATIONAL COMPLEXITY ANALYSIS

---

Algorithm 1: Overall pipeline of **TGT** for link predication

---

**Input:** temporal graph $\mathcal{G} = \{G^t\}_{t=1}^T$; node features $\mathbf{X}^{1:T+1}$; number of neighbors to be sampled k; element-wise nonlinear rectifier $\tau$; Hyperparameters $\lambda, \beta, \Delta t$;

**Initialize:** initialize all weights; set relative time encoding for each timestamp $t$; $Z_{\mathcal{G}_{TX}}^{t}{}^{(0)} \leftarrow \mathbf{X}$ for each timestamp $t$;

**Output:** representation $\mathbf{Z}_X^{T+1}$; Predicted label $\hat{\mathbf{Y}}^{T+1}$ of next timestamp link occurrences.

1: Sample $\mathcal{N}_{k,\Delta t}(v)$ k-top closest spatiotemporal neighborhoods.
2: **for** layers $l = 1, 2, \ldots, L$ **and** $v \in \mathcal{V}^{1:T+1}$ **do**
3:     **for** timestamp $t$ in range[T+1] **do**
4:         $I_s(\mathcal{G}_T; \mathcal{G}) \leftarrow$ Eq. 5
5:         $\hat{\mathbf{Z}}_{\mathcal{G}_{TX}}^{t,(l-1)} \leftarrow \tau(\mathbf{Z}_{\mathcal{G}_{TX}}^{t,(l-1)})\mathbf{W}^{(l)}$
6:         $\phi_{v,k}^{t,(l)} \leftarrow \sigma\{(\hat{\mathbf{Z}}_v^{t,(l-1)} \| \hat{\mathbf{Z}}_u^{\{t-\Delta t:t\},(l-1)})\mathbf{W}_{attention}^{\top}\}_{u \in \mathcal{N}_{k,\Delta t}(v)}$;
7:         $\hat{\mathbf{Z}}_{\mathbf{A}}^{t,(l)} \leftarrow \cup_{v \in \mathcal{V}^t}\{u \in \mathcal{N}_{k,\Delta t}(v) | u \sim \text{Bernoulli}(\phi_{v,k}^{t,(l)})\}$;
8:         $\mathbf{Z}_{\mathcal{G}_{TX}}^{t,(l)} \leftarrow \Sigma_{(u,v) \in \hat{\mathbf{A}}^{t,(l)}}\{\hat{\mathbf{Z}}_{\mathcal{G}_{TX},v}^{t,(l-1)}\}_{v \in \mathcal{V}^t}$;
9:         $I_{DV}(\mathcal{G}_T; \mathcal{G}) = \mathbf{W}_{\mathcal{F}} \cdot \mathbf{Z}_{\mathcal{G}_{TX}}^{t,(l)} - \log \exp(\mathbf{W}_{\mathcal{F}} \cdot \hat{\mathbf{Z}}_u^{\{t-\Delta t:t\},(l-1)})$     # Eq. 6
10:         $l_{evolution}^{(t)} = I_s(\mathcal{G}_T; \mathcal{G}) - I_{DV}(\mathcal{G}_T; \mathcal{G})$     # Eq. 11
11:     **end for**
12:     $\mathcal{G}_T \leftarrow \sigma(\mathbf{Z}_{\mathbf{X}}^{(l)} \| \mathbf{Z}_{\mathbf{A}}^{(l)})\mathbf{W}_{\mathcal{F}}$
13:     $\hat{\mathbf{Y}}^{T+1} = \mathbf{Link\_predictor}(\mathcal{G}_T)$     # Eq. 2
14:     $\mathcal{L}_{IB} \leftarrow$ Eq. 15, $\mathcal{L}_{evolution} \leftarrow \sum_{t=1}^T(l_{evolution}^{(t)})$
15:     $\mathcal{L} \leftarrow \mathcal{L}_{IB} + \lambda \cdot \mathcal{L}_{evolution}$
16:     Update parameters by minimizing $\mathcal{L}$ and back-propagation.
17: **end for**

---

This section conducts a **computational complexity analysis** and gives future work directions for improvement and optimization. The complete pipeline is summarized in Algorithm 1, with input feature dimension $d$ and hidden dimension $d'$. The time complexity can be decomposed as following:

• **line 4:** Computing the VNGE per Eq. 4 costs $O(|E_{\mathcal{G}_T}|)$. During $\Delta t$, obtaining $I_s$ requires:

$$O(\Delta t \cdot |E_{\mathcal{G}_T}|).$$

- **line 5-8:** Sampling $k$ spatiotemporal neighbors and applying GAT feature-projection layers incurs

$$O\big(\Delta t \cdot k \cdot (|V_{\mathcal{G}_T}| \, d \, d' + |E_{\mathcal{G}_T}| \, d)\big).$$

- **line 9:** The final projection step costs $O\big(|V_{\mathcal{G}_T}| \, d \, d' + |V_{\mathcal{G}}^{\{\Delta t\}}| \, d \, d'\big)$, where $|V_{\mathcal{G}}^{\{\Delta t\}}|$ is the number of nodes in the previous $\Delta t$ time of this temporal graph. Since the thumbnail's node set is no smaller than that of any single time window, we conservatively approximate the complexity as $O(2 \, |V_{\mathcal{G}_T}| \, d \, d')$. Putting these together for an $L$-layer architecture yields the time complexity of

$$O\Big(L \cdot \Delta t \big(|E_{\mathcal{G}_T}| + k \cdot (|V_{\mathcal{G}_T}| \, d \, d' + |E_{\mathcal{G}_T}| \, d)\big) + |V_{\mathcal{G}_T}| \, d \, d'\Big).$$

While TGT incurs extra cost for online VNGE computation, its total complexity is comparable to leading temporal GNN baselines. Furthermore, because the `thumbnail` contains no more nodes than the entire temporal graph and is typically much smaller, the effective constant in TGT's representation-learning complexity is reduced.

Table 4: Comparison of training time per epoch (s) with state-of-the-art baselines in robust temporal graph learning across multiple datasets.

| Method | Bitcoin | MathOverflow | MOOC | tgbn-reddit | tgbn-genre |
|--------|---------|--------------|------|-------------|------------|
| DIDA | 23.46 | 37.77 | 8.91 | 27.65 | 17.80 |
| DGIB | 1.37 | 5.53 | 2.56 | 1.57 | 1.71 |
| TGN | 0.78 | 0.86 | 0.74 | 0.68 | 0.71 |
| TGT | 2.26 | 4.42 | 3.34 | 2.73 | 1.68 |

As shown in Table 4, TGT can efficiently handle datasets with node and edge counts on the order of hundreds of thousands. We conducted additional experiments measuring the per epoch training time of TGT compared with several baselines across five datasets (Bitcoin, MathOverflow, MOOC, tgbn-genre, and tgbn-reddit (Huang et al., 2023b)). From the Table 4, we get:

**Time overhead**  TGT is only moderately slower than lightweight temporal GNNs (e.g., TGN). On all datasets, TGT requires around 2–4 seconds per epoch, representing only a small overhead compared to TGN (0.7–0.9s) despite the additional global evolutionary modeling. This overhead mainly comes from computing the approximated trace term $\mathrm{Tr}(\tilde{L}^2)$, which is efficiently computed using sparse matrix and the low-rank Laplacian estimator described in Appendix A.3.1.

Although TGT incurs moderate additional cost due to the computation of VNGE (which depends on node degrees in Eq. 4), the runtime remains close to lightweight TGNNs. MathOverflow appears slower primarily because of its larger per-snapshot node count, while tgbn-reddit, despite larger average snapshots, benefits from a lower-dimensional preprocessed degree matrix.

Nevertheless, TGT is significantly more efficient than robustness-oriented baselines (e.g., DIDA) and still remains competitive with IB methods (e.g., DGIB). This indicates that our thumbnail-guided constraints provide robustness without incurring excessive computational costs.

**Memory footprint**  TGT remains comparable to baselines. Since TGT only stores:

- a static thumbnail graph $G_T$ (small number of nodes)

- low-rank Laplacian terms for VNGE approximation, the memory overhead remains negligible.

- node features for each snapshot.

These components together introduce negligible GPU memory overhead. In addition, since snapshot construction is entirely flexible with respect to temporal granularity, practitioners may adjust the snapshot time scale according to available GPU memory, ensuring efficient resource utilization on different hardware configurations.

In all our experiments, including the large-scale TGB datasets, TGT fits comfortably within **24GB GPU memory** (per RTX 3090 GPU), demonstrating that the method is scalable in practice.

Although VNGE introduces additional computation compared with purely local aggregation models, the overhead is modest in practice, and TGT achieves a favorable balance between robustness, global-evolution modeling, and computational efficiency. The additional results confirm that TGT is computationally practical even on large-scale datasets. For example, MathOverflow contains over 500,000 edges, and MOOC exceeds 400,000 (shown in Table 1). Since the theoretical upper bound of TGT's computational complexity scales linearly with the number of edges across all snapshots, scaling to larger datasets (e.g., datasets with millions of nodes) can be prohibitively time-consuming.

In our future work, we plan to explore pretraining-based approaches for thumbnail modeling, allowing the computational overhead to be offloaded to the data preprocessing stage and thereby improving the scalability of the framework.

## A.2 DETAILS FOR THUMBNAIL DEFINITION

The goal of the `thumbnail` is to summarize holistic evolutionary process features from discrete snapshots in a sequence. To this end, we define the structure of the `thumbnail` and the node correspondence between each snapshot in the sequence and the `thumbnail`, thereby obtaining the posterior probability of the next-timestamp temporal graph state. Our derivation of the `thumbnail` builds on widely recognized works (Luo & Hancock, 2001; Han et al., 2015).

Specifically, consider a temporal graph $\mathcal{G}$ represented as a sequence of length $T$: $\mathcal{G} = \{G^1, \ldots, G^T\}$, where the $i$-th snapshot is $G^i = \{V^i, E^i\}$ with node set $V^i$ and edge set $E^i$. We aim to determine a `thumbnail` $\mathcal{G}_T = \{V_{\mathcal{G}_T}, E_{\mathcal{G}_T}\}$, where $V_{\mathcal{G}_T}$ is the node set, $E_{\mathcal{G}_T}$ is the edge set, and the adjacency matrix $M$ satisfies:

$$M_{\alpha\beta} = \begin{cases} 1 & \text{if } (\alpha, \beta) \in E_{\mathcal{G}_T}, \\ 0 & \text{otherwise.} \end{cases} \tag{17}$$

We first consider a single snapshot $G^i$. For the adjacency matrix $A^i$ of this snapshot:

$$A^i_{ab} = \begin{cases} 1 & \text{if } (a, b) \in E^i, \\ 0 & \text{otherwise.} \end{cases} \tag{18}$$

where $a, b \in V^i$ and $\alpha, \beta \in V_{\mathcal{G}_T}$. For time step $i$, we define an **assignment matrix** $S^i$ with elements $s^i_{a\alpha}$, which denotes the correspondence between node $a$ in snapshot $G^i$ and node $\alpha$ in `thumbnail`:

$$s^i_{a\alpha} = \begin{cases} 1 & \text{if } \mathcal{F}(v^i_a) = \alpha, \\ 0 & \text{otherwise.} \end{cases} \tag{19}$$

where $\mathcal{F}$ denotes the mapping from snapshot $G^i$ to the `thumbnail`. In this paper, we instantiate $\mathcal{F}$ as a GAT network layer with trainable parameters, which is trained using a mutual information alignment loss (see in Eq. 11). In practice, the assignment matrix we seek should maximize the conditional likelihood of the observed snapshots given the available `thumbnail`:

$$S^i = \arg\max_{S^i} P\left(G^i \mid \mathcal{G}_T, S^i\right). \tag{20}$$

This is primarily based on the core idea that "*the correspondence matches assigned to the nodes of the data graph are hidden variables which have arisen through a noisy observation process*" (Luo & Hancock, 2001). Following standard methods for constructing likelihood functions of mixture distributions, we decompose the likelihood of the snapshot and sum over the corresponding nodes in the `thumbnail`:

$$P\left(G^i \mid \mathcal{G}_T, S^i\right) = \prod_{a \in V^i} \sum_{\alpha \in V_{\mathcal{G}_T}} p\left(x_a \mid y_\alpha, S^i\right), \tag{21}$$

$x_a$ denotes the embedding of node $a$ in the snapshot $G^i$, while $y_\alpha$ denotes the embedding of node $\alpha$ in the `thumbnail` $\mathcal{G}_T$. $P\left(x_a \mid y_\alpha, S^i\right)$ denotes the probability of these two under the assignment matrix $S^i$. This assumes that snapshot nodes are conditionally independent given the nodes of $\mathcal{G}_T$. We define the model by specifying $P\left(x_a \mid y_\alpha, S^i\right)$ through conditional probability:

$$P\left(x_a \mid y_\alpha, S^i\right) = \frac{P\left(x_a, y_\alpha, S^i\right)}{P\left(y_\alpha, S^i\right)}. \tag{22}$$

After applying the definitions and properties of conditional probability and performing some algebraic rearrangements (Luo & Hancock, 2001):

$$P\left(x_a \mid y_\alpha, S^i\right) = \frac{\left\{\prod_{b\in V^i}\prod_{\beta\in V_{\mathcal{G}_T}} \frac{P\left(x_a\mid y_\alpha, s^i_{b\beta}\right)P\left(y_\alpha\mid s^i_{b\beta}\right)P\left(s^i_{b\beta}\right)}{P\left(x_\alpha, y_\alpha\right)}\right\} P\left(x_a, y_\alpha\right)}{\left\{\prod_{b\in V^i}\prod_{\beta\in V_{\mathcal{G}_T}} \frac{P\left(y_\alpha\mid s^i_{b\beta}\right)P\left(s^i_{b\beta}\right)}{P(y_\alpha)}\right\} P\left(y_\alpha\right)}$$

$$= \left[\frac{1}{P\left(x_a\mid y_\alpha\right)}\right]^{|V^i|\times|V_{\mathcal{G}_T}|-1} \prod_{b\in V^i}\prod_{\beta\in V_{\mathcal{G}_T}} P\left(x_a\mid y_\alpha, s^i_{b\beta}\right). \tag{23}$$

Given that the snapshot node $x_a$ is conditionally independent of the thumbnail node $y_\alpha$ unless a correspondence $s^i_{a\alpha}$ exists, we get $P(x_a\mid y_\alpha)=P(x_a)$. we can simplify the above equation to:

$$P\left(x_a\mid y_\alpha, S^i\right) = B^i_a \prod_{b\in V^i}\prod_{\beta\in V_{\mathcal{G}_T}} P\left(x_a\mid y_\alpha, s_{b\beta}\right), \quad \text{where } B^i_a = \left[\frac{1}{P\left(x_a\right)}\right]^{|V^i|\times|V_{\mathcal{G}_T}|-1}. \tag{24}$$

Modeling a stable thumbnail requires more stable node information, and nodes that appear briefly may contribute less to the thumbnail modeling. To address this, our TGT introduces a weighting coefficient inversely proportional to the node frequency across snapshots within the time window, enabling tailored handling of transient or non-persistent nodes throughout the sequence.

Specifically, we implement the design described in Eq. 24. Taking newly added nodes as an example, if $x_a$ is a new node, the probability $P(x_a)$ in the denominator of $B^i_a$ is small (as newly added nodes have a low frequency of occurrence in the input snapshots). Its impact on $P(x_a|y_\alpha, s_{b\beta})$ (the thumbnail mapping function) is amplified due to the multiplicative relationship in Eq. 24, thereby enhancing the feature capture of transient or non-persistent nodes by the thumbnail.

To test for edge-consistency, we make use of an indicator that verifies if snapshot nodes $a, b \in V^i$ correspond to valid edges in $\mathcal{G}_T$. Formally:

$$A^i_{ab}M_{\alpha\beta}s^i_{b\beta} = \begin{cases} 1 & \text{if } (a,b)\in E^i \text{ and } (\alpha,\beta)\in E_{\mathcal{G}_T} \text{ and } \mathcal{F}(b)=\beta \\ 0 & \text{otherwise.} \end{cases} \tag{25}$$

When the edge $(a,b)\in E^i$ can be matched to a `thumbnail` edge $(\alpha,\beta)\in E_{\mathcal{G}_T}$ via the node correspondence $\mathcal{F}(a)=\alpha$ and $\mathcal{F}(b)=\beta$, the indicator function takes the value 1; otherwise, it is 0. To model the probabilistic relationship between snapshots and the `thumbnail` using a Bernoulli distribution with error probability $P_e$, we define:

$$P\left(x_a\mid y_\alpha, s^i_{b\beta}\right) = (1-P_e)^{A^i_{ab}M_{\alpha\beta}s^i_{b\beta}} P_e^{1-A^i_{ab}M_{\alpha\beta}s^i_{b\beta}}. \tag{26}$$

Substituting Eq. 26 into Eq. 24, and then substituting the resulting expression into Eq. 21, we derive the likelihood function for a single snapshot:

$$P\left(G^i\mid \mathcal{G}_H, S^i\right) = \prod_{a\in V^i}\sum_{\alpha\in V_{\mathcal{G}_T}} K^i_a \exp\left[\mu\sum_{b\in V^i}\sum_{\beta\in V_{\mathcal{G}_T}} A^i_{ab}M_{\alpha\beta}s^i_{b\beta}\right],$$

$$where \quad \mu = \ln\frac{1-P_e}{P_e}, \; K^i_a = P_e^{|V^i|\times|V_{\mathcal{G}_T}|}B^i_a. \tag{27}$$

For the entire graph sequence $\mathcal{G} = \{G^1,\dots,G^T\}$, we aggregate the likelihood functions of individual snapshots following the approach of Han et al. (2015), where $\mathcal{S} = \{S^1,\dots,S^T\}$ denotes the sequence of assignment matrices and then get Eq. 2:

$$P(\mathcal{G}\mid \mathcal{G}_T, \mathcal{S}) = \prod_{G^i\in\mathcal{G}}\left[P\left(G^i\mid \mathcal{G}_T, S^i\right)\right] = \prod_{G^i\in\mathcal{G}}\left[\prod_{a\in V^i}\sum_{\alpha\in V_{\mathcal{G}_T}} K^i_a \exp\left(\mu\sum_{b\in V^i}\sum_{\beta\in V_{\mathcal{G}_T}} A^i_{ab}M_{\alpha\beta}S^i_{b\beta}\right)\right],$$

$$where \quad \mu = \ln\frac{1-P_e}{P_e}, \quad K^i_a = P_e^{|V^i|\cdot|V_{\mathcal{G}_T}|}B^i_a.$$

## A.3 PROOF FOR EQUATIONS

### A.3.1 APPROXIMATION OF $H_{VN}$ (EQ. 4)

We initiate by providing a improved version of the approximation method for the von Neumann entropy of directed graphs from Ye et al. (2014). Initially, we define the von Neumann entropy, which can be derived from the normalized Laplacian spectrum, as outlined here:

$$H_{VN}(G) = -\text{Tr}(P \log P) = -\sum_{i=1}^{|V|} \frac{\lambda_i}{|V|} \log \frac{\lambda_i}{|V|}, \tag{28}$$

where $\lambda_1, \ldots, \lambda_{|V|}$ denote the eigenvalues of the combinatorial Laplacian matrix, $P$ represents the transition matrix with elements $P_{uv} = \frac{1}{d_u^{\text{out}}}$ if $(u,v)$ is in the edge set $E$ of graph $G$, and $P_{uv} = 0$ otherwise. By scaling the normalized Laplacian matrix by the reciprocal of its trace, one obtains a density matrix given by $\frac{\hat{L}}{|V|}$. The eigenvalues of this density matrix are $\left(\frac{\hat{\lambda}_1}{|V|}, \frac{\hat{\lambda}_2}{|V|}, \ldots, \frac{\hat{\lambda}_{|V|}}{|V|}\right)$. Consequently, the von Neumann entropy of the density matrix associated with the normalized Laplacian matrix of the graph is defined as follows:

$$H_{VN}(G) = -\sum_{j=1}^{|V|} \frac{\hat{\lambda}_j}{|V|} \log \frac{\hat{\lambda}_j}{|V|}. \tag{29}$$

The von Neumann entropy mentioned above depends on the computation of the spectrum of the normalized Laplacian, and hence, its computational complexity scales cubically with the number of nodes. The Taylor expansion for the expression $\ln\left(\frac{\lambda_j}{|V|}\right)$ is given by:

$$\ln\left(\frac{\lambda_j}{|V|}\right) = \left(\frac{\hat{\lambda}_j}{|V|} - 1\right) - \frac{1}{2}\left(\frac{\hat{\lambda}_j}{|V|} - 1\right)^2 + \frac{1}{3}\left(\frac{\hat{\lambda}_j}{|V|} - 1\right)^3 - \frac{1}{4}\left(\frac{\hat{\lambda}_j}{|V|} - 1\right)^4 + \cdots. \tag{30}$$

If we retain only the first term of the expansion and neglect the subsequent terms which contribute insignificantly, $\ln\left(\frac{\hat{\lambda}_j}{|V|}\right)$ can be approximated by $\left(\frac{\hat{\lambda}_j}{|V|} - 1\right)$. Subsequently, the von Neumann entropy $S_{VN}(G)$ can be approximated by the quadratic entropy $\sum_j \frac{\hat{\lambda}_j}{|V|}\left(1 - \frac{\hat{\lambda}_j}{|V|}\right)$, yielding:

$$H_{VN}(G) = -\sum_j \frac{\hat{\lambda}_j}{|V|} \ln \frac{\hat{\lambda}_j}{|V|} \simeq \sum_j \frac{\hat{\lambda}_j}{|V|}\left(1 - \frac{\hat{\lambda}_j}{|V|}\right) = \frac{1}{|V|}\sum_j \lambda_j - \frac{1}{|V|^2}\sum_j \lambda_j^2. \tag{31}$$

Using the fact that $\text{Tr}[\hat{L}^k] = \sum_j \hat{\lambda}_j^k$, the quadratic entropy can be rewritten as:

$$H_{VN}(G) = \frac{\text{Tr}[\hat{L}]}{|V|} - \frac{\text{Tr}\left[\hat{L}^2\right]}{|V|^2}. \tag{32}$$

The normalized Laplacian matrix $\hat{L}$ has unit diagonal elements. For the trace of the normalized Laplacian matrix we have:

$$\text{Tr}[\hat{L}] = |V|. \tag{33}$$

Similarly, for the trace of the square of the normalized Laplacian, we have:

$$\begin{aligned}
\text{Tr}\left[\hat{L}^2\right] &= \text{Tr}\left[I^2 - \left(\Phi^{1/2}P\Phi^{-1/2} + \Phi^{-1/2}P^T\Phi^{1/2}\right) + \frac{1}{4}\left(\Phi^{1/2}P\Phi^{-1/2}\Phi^{1/2}P\Phi^{-1/2}\right.\right. \\
&\quad \left.\left. + \Phi^{1/2}P\Phi^{-1/2}\Phi^{-1/2}P^T\Phi^{1/2} + \Phi^{-1/2}P^T\Phi^{1/2}\Phi^{1/2}P\Phi^{-1/2} + \Phi^{-1/2}P^T\Phi^{1/2}\Phi^{-1/2}P^T\Phi^{1/2}\right)\right] \\
&= \text{Tr}\left[I^2\right] - \text{Tr}[P] - \text{Tr}\left[P^T\right] + \frac{1}{4}\left(\text{Tr}\left[P^2\right] + \text{Tr}\left[P\Phi^{-1}P^T\Phi\right] + \text{Tr}\left[P^T\Phi P\Phi^{-1}\right] + \text{Tr}\left[P^{T^2}\right]\right) \\
&= |V| + \frac{1}{2}\left(\text{Tr}\left[P^2\right] + \text{Tr}\left[P\Phi^{-1}P^T\Phi\right]\right),
\end{aligned} \tag{34}$$

where $\Phi = diag(\phi(1), \phi(2), \dots)$ , $\phi$ is the unique left eigenvector of P, and then: (details in Ye et al. (2014))

$$\frac{\phi(u)}{\phi(v)} = \frac{d_u^{in}}{d_v^{in}}. \tag{35}$$

To continue the development we first partition the edgeset $E$ into two disjoint subsets $E_1$ and $E_2$, where $E_1 = \{(u,v)|(u,v) \in E \text{ and } (v,u) \notin E\}$, $E_2 = \{(u,v)|(u,v) \in E \text{ and } (v,u) \in E\}$. Then according to the definition of the transition matrix, we find:

$$\text{Tr}[P^2] = \sum_{u \in V} \sum_{v \in V} P_{uv} P_{vu} = \sum_{(u,v) \in E_2} \frac{1}{d_u^{out} d_v^{out}},$$

$$\text{Tr}[P\Phi^{-1}P^T\Phi] = \sum_{u \in V} \sum_{v \in V} P_{uv}^2 \frac{\phi(u)}{\phi(v)} = \sum_{(u,v) \in E} \frac{\phi(u)}{\phi(v) d_u^{out2}}. \tag{36}$$

Substitute Eq. 36 into Eq. 34, then Substitute Eq. 33 and Eq. 34 into Eq. 32, we get Eq. 4:

$$H_{VN} = 1 - \frac{1}{|V_{\mathcal{G}_T}|} - \frac{1}{2|V_{\mathcal{G}_T}|^2} \left\{ -\sum_{(\alpha,\beta) \in E_{\mathcal{G}_{T_1}}} \frac{1}{d_\beta^{\text{out}} d_\alpha^{\text{out}}} + \sum_{(\alpha,\beta) \in E_{\mathcal{G}_T}} \left( \frac{d_\alpha^{\text{in}}}{d_\beta^{\text{in}} d_\alpha^{\text{out2}}} + \frac{1}{d_\beta^{\text{out}} d_\alpha^{\text{out}}} \right) \right\},$$

$$\text{where} \quad d_\alpha^{\text{in}} = \sum_{\gamma \in V_{\mathcal{G}_T}} M_{\gamma\alpha}, \quad d_\alpha^{\text{out}} = \sum_{\gamma \in V_{\mathcal{G}_T}} M_{\alpha\gamma}.$$

Since the normalized Laplacian $\tilde{L}$ is sparse for most real-world graphs, $\text{Tr}(\tilde{L}^2)$ can be computed without eigenvalue decomposition by exploiting sparsity (Eq. 34-36):

$$\text{Tr}(\tilde{L}^2) = \sum_{i,j} \tilde{L}_{ij}^2 = \sum_{(u,v) \in E_{G_T}} w_{uv}^2 + \sum_{u \in V_{G_T}} d_u^2. \tag{37}$$

Both terms depend linearly on the number of edges, leading to $O(|E_{G_T}|)$ complexity per snapshot(Ye et al., 2014). During temporal aggregation, we process $\Delta t$ snapshots, thus the total complexity of the evolutionary term becomes $O(\Delta t \cdot |E_{G_T}|)$, which matches the asymptotic bound stated in the paper. No strong assumptions are required, only the standard sparsity condition $|E_{G_T}| \ll |V_{G_T}|^2$, which holds for most real-world dynamic graphs.

Furthermore, in practice, we pre-normalize edge weights and reuse cached degree matrices across consecutive snapshots. This amortizes the per-snapshot cost and further reduces the effective runtime, as validated by our empirical runtime profiling in Table 4.

### A.3.2 DONSKER-VARADHAN ESTIMATOR FOR MUTUAL INFORMATION (EQ. 6)

To prove Eq. 6, we need to show:

$$I_s(\mathcal{G}_T; \mathcal{G}) = \sup_{f \in \mathcal{F}} \left( \mathbb{E}_P[f(V_{\mathcal{G}})] - \log \mathbb{E}_Q \left[ e^{f(V_{\mathcal{G}})} \right] \right).$$

which is equivalent to proving:

$$D_{KL}(P \| Q) \geq \mathbb{E}_P[T] - \log(\mathbb{E}_Q[e^T]) = \sum_i p_i t_i - \log \sum_i q_i e^{t_i}. \tag{38}$$

To find the extremum of Eq. 38, we take the derivative with respect to $t_j$ and set it equal to zero:

$$\frac{\partial \left[ \sum_i p_i t_i - \log \sum_i q_i e^{t_i} \right]}{\partial t_j} = 0$$

$$\rightarrow p_j - \frac{q_j e^{t_j}}{\sum_i q_i e^{t_i}} = 0$$

$$\rightarrow p_j \sum_i q_i e^{t_i} = q_j e^{t_j} \tag{39}$$

$$\rightarrow t_j = \log \frac{p_j}{q_j} + \log \sum_i q_i e^{t_i}.$$

Let $\alpha = \log \sum_i q_i e^{t_i}$, and substitute $t_j$ into $t_i$:

$$
\begin{aligned}
\sum_i p_i t_i - \log \sum_i q_i e^{t_i} &= \sum_i p_i t_i - \log \sum_i q_i e^{t_i} \\
&= \sum_i p_i \left( \log \frac{p_j}{q_j} + \alpha \right) - \log \sum_i q_i e^{\log\left(\frac{p_j}{q_j} + \alpha\right)} \\
&= \sum_i p_i \left( \log \frac{p_j}{q_j} + \alpha \right) - \log \sum_i e^\alpha q_i \frac{p_j}{q_j} \\
&= \sum_i \left( p_i \log \frac{p_j}{q_j} \right) + \alpha - \alpha - \log \sum_i q_i \frac{p_j}{q_j} \\
&= \sum_i \left( p_i \log \frac{p_j}{q_j} \right) + \alpha - \alpha - \log 1 \\
&= \sum_i p_i \log \frac{p_j}{q_j} = D_{KL}(p\|q).
\end{aligned}
\tag{40}
$$

Thus, the inequality Eq. 6 becomes an equality at the extreme value. When substituting $f(V_\mathcal{G})$ for $T$ in Eq. 38, the equality holds.

### A.3.3 Upper bound of $I(\mathcal{G}; \mathcal{G}_T)$ (Eq. 9)

For any groups of indices $L, S_{\mathcal{G}_{TX}}, S_{\mathcal{G}_{TA}} \subset [L]$ such that $\mathcal{G} \perp Z_{\mathcal{G}_{TX}}^{(L)} | \{Z_{\mathcal{G}_{TX}}^{(l)}\}_{l \in S_X} \cup \{Z_{\mathcal{G}_{TA}}^{(l)}\}_{l \in S_A}$, and for any probabilistic distributions $\mathbb{Q}(Z_{\mathcal{G}_{TX}}^{(l)}), l \in S_{\mathcal{G}_{TX}}$, and $\mathbb{Q}(Z_{\mathcal{G}_{TA}}^{(l)}), l \in S_{\mathcal{G}_{TA}}$,

$$
I(\mathcal{G}; \mathcal{G}_T) \leq I(\mathcal{G}; \{Z_{\mathcal{G}_{TX}}^{(l)}\}_{l \in S_{\mathcal{G}_{TX}}} \cup \{Z_{\mathcal{G}_{TA}}^{(l)}\}_{l \in S_{\mathcal{G}_{TA}}}) \leq \sum_{l \in S_{\mathcal{G}_{TX}}} \mathrm{IB}_{\mathcal{G}_{TX}}^{(l)} + \sum_{l \in S_{\mathcal{G}_{TA}}} \mathrm{IB}_{\mathcal{G}_{TA}}^{(l)},
$$

where $\quad \mathrm{IB}_{\mathcal{G}_{TA}}^{(l)} = \mathbb{E}\left[ \log \frac{\mathbb{P}(Z_{\mathcal{G}_{TA}}^{(l)}|\mathbf{A}, Z_{\mathcal{G}_{TA}}^{(l-1)})}{\mathbb{Q}(Z_{\mathcal{G}_{TA}}^{(l)})} \right], \mathrm{IB}_{\mathcal{G}_{TX}}^{(l)} = \mathbb{E}\left[ \log \frac{\mathbb{P}(Z_{\mathcal{G}_{TX}}^{(l)}|Z_{\mathcal{G}_{TX}}^{(l-1)}, Z_{\mathcal{G}_{TA}}^{(l)})}{\mathbb{Q}(Z_{\mathcal{G}_{TX}}^{(l)})} \right].$

By assumption 1, $\mathcal{G} \perp Z_{\mathcal{G}_{TX}}^{(L)} | \{Z_{\mathcal{G}_{TX}}^{(l)}\}_{l \in S_X} \cup \{Z_{\mathcal{G}_{TA}}^{(l)}\}_{l \in S_A}$ ensures that the left side of the inequality holds. As the second inequality, under Assumption 1, we define an order $\prec$ on the set of random variables $\{Z_{\mathcal{G}_{TX}}^{(l)}\}_{l \in S_{\mathcal{G}_{TX}}} \cup \{Z_A^{(l)}\}_{l \in S_{\mathcal{G}_{TA}}}$ as follows:

1. For distinct timestamps $l$ and $l'$, $Z_{\mathcal{G}_{TX}}^{(l)}, Z_{\mathcal{G}_{TA}}^{(l)} \prec Z_{\mathcal{G}_{TX}}^{(l')}, Z_{\mathcal{G}_{TA}}^{(l')}$.

2. For a given timestamp $l$, $Z_{\mathcal{G}_{TA}}^{(l)} \prec Z_{\mathcal{G}_{TX}}^{(l)}$.

Subsequently, we define a sequence of sets based on this order.

$$
\begin{aligned}
H_{\mathcal{G}_{TA}}^{(l)} &= \{Z_{\mathcal{G}_{TX}}^{(l_1)}, Z_{\mathcal{G}_{TA}}^{(l_2)} | l_1 < l, l_2 < l, l_1 \in S_{\mathcal{G}_{TX}}, l_2 \in S_{\mathcal{G}_{TA}}\}, \\
H_{\mathcal{G}_{TX}}^{(l)} &= \{Z_{\mathcal{G}_{TX}}^{(l_1)}, Z_{\mathcal{G}_{TA}}^{(l_2)} | l_1 < l, l_2 \leq l, l_1 \in S_{\mathcal{G}_{TX}}, l_2 \in S_{\mathcal{G}_{TA}}\}.
\end{aligned}
$$

We may decompose $I(\mathcal{G}; \{Z_{\mathcal{G}_{TX}}^{(l)}\}_{l \in S_{\mathcal{G}_{TX}}} \cup \{Z_{\mathcal{G}_{TA}}^{(l)}\}_{l \in S_{\mathcal{G}_{TA}}})$ with respect to this order.

$$
I(\mathcal{G}; \{Z_{\mathcal{G}_{TX}}^{(l)}\}_{l \in S_{\mathcal{G}_{TX}}} \cup \{Z_{\mathcal{G}_{TA}}^{(l)}\}_{l \in S_{\mathcal{G}_{TA}}}) = \sum_{l \in S_{\mathcal{G}_{TA}}} I(\mathcal{G}; Z_{\mathcal{G}_{TA}}^{(l)}|H_{\mathcal{G}_{TA}}^{(l)}) + \sum_{l \in S_{\mathcal{G}_{TX}}} I(\mathcal{G}; Z_{\mathcal{G}_{TX}}^{(l)}|H_{\mathcal{G}_{TX}}^{(l)}).
\tag{41}
$$

Similar to Wu et al. (2020), we get upper bounds for $I(\mathcal{G}; Z^{(l)}_{\mathcal{G}_{TA}} \mid H^{(l)}_{\mathcal{G}_{TA}})$ and $I(\mathcal{G}; Z^{(l)}_{\mathcal{G}_{TX}} \mid H^{(l)}_{\mathcal{G}_{TX}})$.

$$
\begin{aligned}
I(\mathcal{G}; Z^{(l)}_{\mathcal{G}_{TA}} | H^{(l)}_{\mathcal{G}_{TA}}) &\leq I(\mathcal{G}, Z^{(l-1)}_{\mathcal{G}_{TX}}; Z^{(l)}_{\mathcal{G}_{TA}} | H^{(l)}_{\mathcal{G}_{TA}}) \\
&= I(Z^{(l-1)}_{\mathcal{G}_{TX}}, A; Z^{(l)}_{\mathcal{G}_{TA}} | H^{(l)}_{\mathcal{G}_{TA}}) \\
&\leq I(Z^{(l-1)}_{\mathcal{G}_{TX}}, A; Z^{(l)}_{\mathcal{G}_{TA}}) \\
&= \text{IB}^{(l)}_{\mathcal{G}_{TA}} - \text{KL}(\mathbb{P}(Z^{(l)}_{\mathcal{G}_{TA}}) || \mathbb{Q}(Z^{(l)}_{\mathcal{G}_{TA}})) \leq \text{IB}^{(l)}_{\mathcal{G}_{TA}}
\end{aligned}
\tag{42}
$$

$$
\begin{aligned}
I(\mathcal{G}; Z^{(l)}_{\mathcal{G}_{TX}} | H^{(l)}_{\mathcal{G}_{TX}}) &\leq I(\mathcal{G}, Z^{(l-1)}_{\mathcal{G}_{TX}}, Z^{(l)}_{\mathcal{G}_{TA}}; Z^{(l)}_{\mathcal{G}_{TX}} | H^{(l)}_{\mathcal{G}_{TX}}) \\
&= I(Z^{(l-1)}_{\mathcal{G}_{TX}}, Z^{(l)}_{\mathcal{G}_{TA}}; Z^{(l)}_{\mathcal{G}_{TX}} | H^{(l)}_{\mathcal{G}_{TX}}) \\
&\leq I(Z^{(l-1)}_{\mathcal{G}_{TX}}, Z^{(l)}_{\mathcal{G}_{TA}}; Z^{(l)}_{\mathcal{G}_{TX}}) \\
&= \text{IB}^{(l)}_{\mathcal{G}_{TX}} - \text{KL}(\mathbb{P}(Z^{(l)}_{\mathcal{G}_{TX}}) || \mathbb{Q}(Z^{(l)}_{\mathcal{G}_{TX}})) \leq \text{IB}^{(l)}_{\mathcal{G}_{TX}}
\end{aligned}
\tag{43}
$$

In summary, the proof of Eq. 9 is complete.

### A.3.4 LOWER BOUND OF $I(\mathbf{Y}; \mathcal{G}_T)$ (EQ. 10)

We apply the Nguyen, Wainright and Jordan's bound to prove the equation (Nguyen et al., 2010):

**Lemma 1** *Given any two $X_1$ and $X_2$ and any permutation invariant function g, we have the variational lower bound of $I(X_1; X_2)$:*

$$
I(X_1, X_2) \geq \mathbb{E}\left[g(X_1, X_2)\right] - \mathbb{E}_{\mathbb{P}(X_1)\mathbb{P}(X_2)}\left[\exp(g(X_1, X_2) - 1)\right].
$$

We substitute $g\left(Y, Z^{(l)}_{\mathcal{G}_{TX}}\right) = 1 + \log \frac{\prod_{v \in V_{\mathcal{N}(v)}} P\left(Y_v | Z^{(l)}_{\mathcal{G}_{TX}}, Z^{(l)}_{\mathcal{G}_{TA}}\right)}{Q(Y)}$ and obtain Eq. 10:

$$
I(\mathbf{Y}; \mathcal{G}_T) \geq 1 + \mathbb{E}_{P(\mathbf{Y}, \mathcal{G}_T)}[\log \frac{P(\mathbf{Y}|(Z^{(l)}_{\mathcal{G}_{TX}}, Z^{(l)}_{\mathcal{G}_{TA}}))}{Q(\mathbf{Y})}] - \mathbb{E}_{P(\mathbf{Y})}[\frac{\mathbb{E}_{P(\mathcal{G}_T)}P(\mathbf{Y}|(Z^{(l)}_{\mathcal{G}_{TX}}, Z^{(l)}_{\mathcal{G}_{TA}}))}{Q(\mathbf{Y})}].
$$

### A.4 DETAILS FOR DATASETS

In this section, we provide detailed information on the three datasets used in the experiment. When selecting datasets, we consider multiple dimensions including time span, graph density, and evolution frequency. Since the TGT framework focuses on global evolutionary features, we prioritize datasets with high dynamic evolution frequencies in temporal graphs during our selection process. Additionally, under similar evolution frequencies and time granularity, we favor datasets with fewer nodes per snapshot. Such datasets exhibit greater randomness under the same perturbation ratio, enabling a more effective validation of the model's robustness advantages in stochastic settings. Following these dataset selection criteria, we choose **Bitcoin**, **MathOverflow**, and **MOOC** for the main experiments and supplementary analytical ablation (see Table 1).

**Bitcoin**    Bitcoin is a cryptocurrency to trade anonymously over the web. Due to anonymity, there is counterpart risk, which has lead to the emergence of several exchanges where Bitcoin users rate the level of trust they have in others. We integrate the data from the two trading platforms Bitcoin-OTC and Bitcoin-Alpha into a time series dataset ***Bitcoin***. Both these exchanges allow users to rate others on a scale of -10 to +10 (excluding 0). According to guidelines, a rating of -10 should be given to fraudsters while at the other end of the spectrum, +10 means "you trust him as you trust yourself". The other rating values have intermediate meanings. We assign two-dimensional features to nodes, which represent the trust and trustworthiness level, respectively, and sum up the rating of users.

**MathOverflow**    ***MathOverflow*** is a temporal network of user interactions on the Stack Exchange website Math Overflow. Nodes represent users, and edges represent question-answer relationships. Three distinct types of interactions are modeled by directed edges $(u, v, t)$, where:

- user $u$ answers user $v$'s question at time $t$;

- user $u$ comments on user $v$'s question at time $t$;

- user $u$ comments on user $v$'s answer at time $t$.

While constructing adjacency relations based on triplets, we assign six-dimensional features to nodes, statistically capturing the counts of: questions posted, answers provided, comments on questions, questions commented on, comments on answers, and answers commented on. We use these features to build a temporal graph for experimental purposes.

**MOOC**   The *MOOC* dataset represents user behavior on a popular MOOC platform, modeled as a directed temporal network. Nodes represent users and course activities (targets), while edges represent user actions on these targets. Each action is annotated with attributes and a timestamp, where timestamps are normalized to start from timestamp 0. The dataset is directed, temporal, and attributed. Additionally, each action includes a binary label indicating whether the user withdrew from the course after this action—i.e., whether this was the user's final interaction with the platform.

For snapshot partitioning of a dataset, we follow these two principles.

• **Chronological order.** We divide the dynamic graph strictly by chronological order to avoid any missing or overlapping temporal intervals. This ensures that temporal dependencies are preserved as faithfully as possible. Because different intervals naturally contain varying numbers of nodes and edges, the node/edge counts reported in Table 1 reflect the **average** snapshot scale after segmentation. Presenting the average scale helps provide an intuitive sense of computational complexity.

• **Handling extreme variations in temporal activity.** In datasets with highly uneven temporal activity (e.g., MathOverflow), directly segmenting by time may produce snapshots that are either extremely sparse or extremely dense. To address this, we follow chronological segmentation but enforce reasonable lower and upper bounds on snapshot size by merging overly small adjacent snapshots or splitting overly large ones. This adjustment maintains strict temporal continuity (no overlap, no omission) while ensuring more stable computation and memory usage across snapshots.

## A.5   DETAILS FOR IMPLEMENTATION

The complete process of TGT in the link prediction task is shown in algorithm 1. We implemented TGT utilizing GAT (Veličković et al., 2018), incorporating a 2-layered, 2-headed attention mechanism for facilitating message passing and aggregation processes. The hidden dimension is 64.

In our experimental setup, we sampled neighbors with a size of $k = 20$ and constrained the spatiotemporal neighborhood to encompass $\Delta t = 5$ snapshots. Neighbor sampling was conducted according to the Bernoulli distribution, guided by attention weights. The Adam optimizer was employed, with an initial learning rate of 0.001, and the training process was iterated 1000 epochs with early stop. To ensure the robustness of our results, all experiments were conducted using three distinct random seeds: 0, 1, and 2. We reported the average performance and standard deviation derived from these repetitions. All the above experiments are conducted on two NVIDIA GeForce RTX 3090 GPUs with 24GB of memory each.

## A.6   DETAILS FOR ADVERSARIAL ATTACK EXPERIMENT

**Node Features**   To introduce perturbations to node features, we add Gaussian noise $r \cdot \epsilon$, where the interference amplitude $r$ is set as the average of the maximum values of each node feature, and $\epsilon \sim \mathcal{N}(0, 1)$. Specifically, we randomly select 10%, 20%, and 50% of nodes in each dataset to simulate increasing levels of interference, gradually escalating the perturbation intensity.

**Topology Structure**   To perturb the topological structure, we employ Nettack (Zügner et al., 2018), a proxy-model-based adversarial attack method that iteratively selects graph substructures with the highest impact on prediction loss as perturbation targets. This approach greedily identifies edges or node pairs whose modification most significantly affects the proxy model's gradient calculations, then applies perturbations such as edge deletion or negative sampling based on their impact on the optimization process. Specifically, Nettack simulates increasing perturbation severity by selecting the top 5%, 10%, and 20% of edges ranked by their influence on prediction loss, progressively intensifying structural disturbances to evaluate model robustness.

**Temporal Evolution**   To perturb the temporal evolution dynamics, we disrupt the modeling of underlying temporal dependencies by randomly permuting snapshots within the sequence. Specifically, we randomly select and replace a snapshot with another from a different time step within the same sequence, thereby introducing inconsistencies in the chronological order. Across all datasets, we apply such temporal perturbations 1, 2, and 5 times to simulate increasing levels of disruption, allowing us to evaluate how model performance degrades with progressively stronger interference in the temporal continuity.

In addition to the untargeted interference mentioned above, we also conducted a detailed investigation and selected five targeted attack methods to interfere with our data, and used this to verify the robustness of TGT.

• **Nettack**(Zügner et al., 2018): Nettack is a pioneering white-box adversarial attack for graph neural networks (GNNs), which perturbs graph structure and node features to misclassify target nodes while ensuring imperceptibility via preserving key graph properties; it adopts a simplified linearized GCN as the surrogate model for computational efficiency, leverages gradient-guided greedy selection to iteratively apply impactful modifications, and exhibits notable efficiency, generalizability across GNN architectures, and robustness to partial graph knowledge.

• **FGA**(Chen et al., 2019): Fast Gradient Attack (FGA) is a white-box adversarial attack for network embedding, which perturbs graph structure via link rewiring to degrade the performance of downstream tasks; it leverages gradient information of pairwise nodes to select the most impactful node pairs, and exhibits strong generalizability across multiple network embedding methods as well as high efficiency with minimal link modifications.

• **SGA**(Li et al., 2021): SGA is an adversarial attack for large-scale graphs, which perturbs structure (edge additions/deletions) to degrade downstream task performance; it adopts scalable strategies (e.g., sampling) for large-graph adaptability, leverages lightweight gradient-guided perturbation selection, and features strong scalability, high efficiency, and minimal imperceptible modifications.

• **PRBCD**(Geisler et al., 2021): Projected Randomized Block Coordinate Descent (PRBCD) is an adversarial attack for large-scale GNNs, which solely perturbs the adjacency matrix to degrade classification performance; it adopts randomized block coordinate descent with continuous relaxation of discrete 0,1 adjacency entries, leverages gradient optimization for efficient perturbation selection, and supports both evasion and poisoning attacks with strong scalability and low memory overhead.

• **GOttack**(Alom et al., 2025): Gottack is a universal adversarial attack for GNNs, which perturbs graph structure via topology-aware graph orbits learning to boost misclassification rates; it narrows the search space with orbit-based selection, and features high efficiency, strong cross-architecture generalizability, and subtle effective perturbations.

## A.7   ADDITIONAL ABLATION EXPERIMENT ON THUMBNAIL BASED ON VNGE

To validate the efficacy of the `thumbnail`'s evolutionary constraints, which is implemented through the von Neumann Graph Entropy (VNGE), we conducted supplementary experiments on noise-perturbed datasets, serving as a complement to the results presented in Fig. 2.

Table 5: Ablation study on holistic continuous evolutionary characteristics results (**AUC**) on datasets with data perturbation at different levels.

| Dataset | Model | Clean | Feature Interference | | | Structure Interference | | | Temporal Interference | | |
|---|---|---|---|---|---|---|---|---|---|---|---|
| | | | 10% | 20% | 50% | 5% | 10% | 20% | $n=1$ | $n=2$ | $n=5$ |
| **Bitcoin** | w/o thumbnail | 73.04±0.6 | 67.73±0.2 | 61.82±0.3 | 56.12±0.3 | 69.77±0.5 | 62.87±0.3 | 57.51±0.3 | 64.21±0.3 | 59.61±0.4 | 55.34±0.3 |
| | w/o T | 89.56±0.2 | 83.45±0.3 | 74.31±0.2 | 66.20±0.4 | 83.26±0.3 | 79.67±0.3 | 68.37±0.2 | 84.73±0.2 | 80.04±0.3 | 71.06±0.2 |
| | w/o VNGE | 86.91±0.5 | 80.17±0.3 | 75.46±0.4 | 69.73±0.2 | 82.55±0.3 | 78.04±0.3 | 72.42±0.3 | 83.79±0.2 | 81.54±0.5 | 74.37±0.3 |
| | **TGT** | **91.41±0.2** | **89.43±0.5** | **86.23±0.4** | **80.62±0.4** | **89.83±0.6** | **85.00±0.5** | **80.95±0.7** | **90.78±0.5** | **88.14±0.6** | **85.64±0.3** |
| **MathOverflow** | w/o thumbnail | 76.11±0.3 | 74.24±0.2 | 68.41±0.3 | 60.21±0.4 | 72.70±0.5 | 68.29±0.3 | 60.54±0.3 | 70.32±0.3 | 62.47±0.5 | 56.36±0.7 |
| | w/o T | 80.35±0.3 | 75.52±0.3 | 71.43±0.3 | 65.52±0.3 | 77.13±0.3 | 73.31±0.3 | 68.74±0.5 | 76.39±0.4 | 73.30±0.3 | 64.27±0.3 |
| | w/o VNGE | 80.29±0.3 | 77.34±0.3 | 71.46±0.3 | 67.98±0.3 | 77.43±0.4 | 74.67±0.3 | 71.49±0.2 | 78.61±0.3 | 76.07±0.3 | 69.83±0.2 |
| | **TGT** | **82.38±0.6** | **79.22±0.3** | **74.37±0.3** | **71.42±0.2** | **80.01±0.3** | **77.74±0.3** | **75.93±0.7** | **81.57±0.2** | **80.71±0.5** | **76.37±0.4** |
| **MOOC** | w/o thumbnail | 86.47±0.3 | 83.63±0.1 | 74.94±0.3 | 61.34±0.5 | 83.34±0.3 | 72.61±0.4 | 60.25±0.2 | 81.06±0.4 | 76.34±0.4 | 69.71±0.7 |
| | w/o T | 90.78±0.3 | 85.32±0.2 | 72.78±0.4 | 60.23±0.3 | 86.54±0.3 | 81.17±0.3 | 73.54±0.2 | 85.21±0.2 | 78.55±0.4 | 69.49±0.5 |
| | w/o VNGE | 92.03±0.2 | 86.73±0.2 | 77.35±0.4 | 66.27±0.3 | 89.58±0.4 | 84.90±0.5 | 77.31±0.3 | 84.19±0.2 | 79.41±0.3 | 70.59±0.4 |
| | **TGT** | **95.42±0.4** | **88.73±0.2** | **80.79±0.1** | **71.68±0.3** | **90.43±0.3** | **86.16±0.5** | **81.82±0.3** | **92.61±0.2** | **89.53±0.7** | **84.01±0.6** |

To construct the ablation baseline *w/o VNGE*, we selected normalized node degrees from Eq. 11 as a substitute for the von Neumann entropy. We also added an ablation control *w/o T*, where the term $I_s(\mathcal{G}_T; \mathcal{G})$ in Eq. 11 is set to zero entirely, and a further ablation setting *w/o thumbnail* that completely discards the construction of `thumbnail` from Eq. 11.

As shown in Table 5, the construction of the `thumbnail` plays a critical role in the model's representational capability and robustness. Even when structural information is excluded in the *w/o T* ablation (where the `thumbnail` is determined solely by node features from Eq. 6), the model still improves its representation of temporal graphs using node semantics compared with *w/o thumbnail*.

In contrast to naive alternatives for incorporating evolutionary constraints, such as normalized node degree (*w/o VNGE*), the `thumbnail` modeled via mutual information based on VNGE (our proposed method) demonstrates significant advantages in both representational capability and robustness. This confirms the effectiveness of the VNGE-based `thumbnail` in enhancing the model's ability to capture continuous evolutionary features while resisting perturbations.

## A.8 ADDITIONAL RESULTS ON ABLATION STUDY

We conducted individual ablation studies on the three purposed components to investigate the specific contribution of each to the experimental outcomes. When the data set has not been disturbed, the impact of each component on the experimental results is shown in Fig. 4:

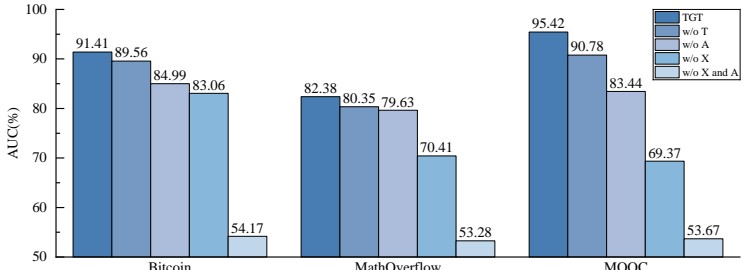

Figure 4: The figure shows the ablation experiment results on three datasets. Where $T$ represents the evolution constraint (Eq. 11), $A$ represents the structure mutual information (Eq. 13), and $X$ represents the feature mutual information (Eq. 12)

### A.8.1 FEATURE MUTUAL INFORMATION

After setting the feature mutual information (Eq. 12) to 0, we repeated the robustness test of feature interference on the three datasets. The results are shown in Fig. 5. Without the feature mutual information, the model's robustness to feature interference drops significantly.

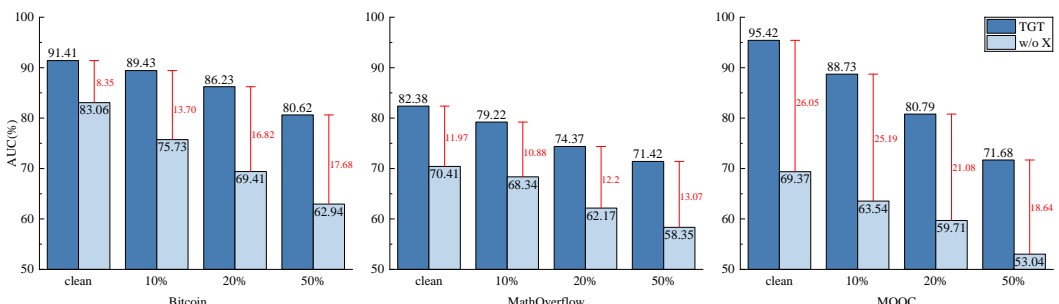

Figure 5: The figure shows the experimental results after the node features on the three data sets were perturbed to different degrees.

### A.8.2 STRUCTURE MUTUAL INFORMATION

With the structure mutual information (Eq. 13) set to zero, we conducted a repeated robustness assessment of structure interference across the three datasets, shown in Fig. 6. Compared with feature mutual information, structure mutual information has less impact on clean data, but its impact on the robustness of the model is more obvious.

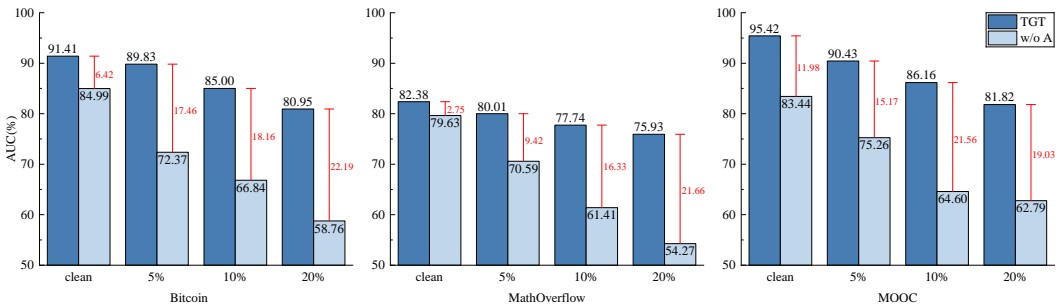

Figure 6: The figure shows the experimental results after the topology on the three data sets were perturbed to different degrees.

### A.8.3 ABLATION ON BACKBONE

In this section, we mainly design experiments and discuss the impact of the encoder's backbone network configuration on TGT performance. The backbone corresponds to Lines 6–8 of Algorithm 1 in Appendix A.1.4, where the encoder is applied. Although GAT is adopted in the main paper owing to its capacity to model node-specific contributions to the thumbnail through attention mechanisms, the TGT framework itself is backbone-agnostic.

Table 6: The results under clean data and three types of perturbations on Bitcoin dataset.

| Backbone | Clean | Feature Interference | | | Structure Interference | | | Temporal Interference | | |
|---|---|---|---|---|---|---|---|---|---|---|
| | | 10% | 20% | 50% | 5% | 10% | 20% | $n=1$ | $n=2$ | $n=5$ |
| GCN | 90.86 | 88.93 | 86.89 | 81.73 | 89.66 | 85.27 | 78.13 | 89.24 | 86.05 | 82.75 |
| GIN | 91.44 | 90.24 | 84.52 | 79.87 | 87.72 | 85.33 | 81.02 | 89.90 | 88.31 | 84.13 |
| GAT (Default) | 91.41 | 89.43 | 86.23 | 80.62 | 89.83 | 85.00 | 80.95 | 90.78 | 88.14 | 85.64 |

To elaborate on this flexibility, we have implemented two additional TGT variants with alternative encoders such as GCN (Kipf, 2016) and GIN (Xu et al., 2018). We repeat the experiments on the Bitcoin dataset by replacing the backbone and obtain the results shown in Table 6.

These results demonstrate two points: **(i)** TGT's performance is stable across different backbones. The changes in node encoder architecture have only minor influence on clean-data accuracy and robustness under perturbations. **(ii)** GAT performs slightly better, which is expected because the attention mechanism naturally aligns with the design of TGT. It allows different nodes in each snapshot to contribute unequally to thumbnail, reflecting our motivation to capture heterogeneous importance.

In summary, while GAT is a strong and intuitive choice for TGT, the framework does not rely on a specific backbone architecture and can be instantiated using standard GNNs such as GCN and GIN with comparable performance.

### A.9 HYPERPARAMETER SENSITIVITY EXPERIMENTS OF TGT

We performed hyperparameter sensitivity analyzes on the Lagrangian coefficients defined in Eq. 15 and Eq. 16, examining their impact on robustness behavior and optimization of temporal evolution. We conduct experiments on MOOC dataset, which has the most edges and the fastest evolution.

**The impact of Hyperparameter setting on thumbnail**  In this section, we primarily design experiments and discuss the impact of hyperparameter settings on thumbnail modeling, and further provide suggestions on the parameter selection methods for Lagrangian hyperparameters.

$\lambda$ serves to specify the constraint strength exerted by the thumbnail over the encoding process. To clarify this relationship, we have supplemented with experimental analyses on the number of thumbnail nodes as a function of $\lambda$. Different choices of $\lambda$ indirectly influence the thumbnail size, yet the thumbnail size remains close to the average number of nodes (and edges) across snapshots. This, in turn, ensures the two terms are of the same order of magnitude during gradient computation, thus validating the rationality of confining $\lambda$'s value range to 10.

Table 7: Thumbnail statistics under different $\lambda$ settings.

| Dataset | Thumbnail | $\lambda$ | | | | | Snapshot Avg. |
|---|---|---|---|---|---|---|---|
| | | 0.1 | 0.5 | 1 | 2 | 5 | |
| Bitcoin | nodes | 6,954 | 6,844 | 7,019 | 7,090 | 7,051 | 7,034 |
| | edges | 47,558 | 48,880 | 47,558 | 47,969 | 52,982 | 51,363 |
| MathOverflow | nodes | 20,474 | 19,965 | 23,376 | 22,371 | 24,105 | 21,683 |
| | edges | 160,491 | 262,295 | 315,105 | 417,491 | 478,325 | 207,581 |
| MOOC | nodes | 6,120 | 6,264 | 6,275 | 6,152 | 6,340 | 7,047 |
| | edges | 53,552 | 56,628 | 63,415 | 61,074 | 68,068 | 81,749 |

Regarding the two terms balanced by $\beta$ (Eq. 15): the $TB$ term is computed via snapshot node sampling, while the downstream task mutual information is calculated using thumbnail nodes. Table 7 confirms that the two terms are of the same order of magnitude, thus justifying the rationality of setting $\beta$'s value range within 10.

Unfortunately, tuning these two hyperparameters is an empirical process, as changes in data scenarios and specific requirements directly influence their optimal values. In practice, we recommend conducting a small number of warm-up training runs for hyperparameter search, thereby achieving more reliable hyperparameter selection.

Specifically, during the pre-training phase, we estimate the noise level of the input sequence via the variance of von Neumann graph entropy across consecutive snapshots, denoted as $\text{Var}(H(\mathcal{G}^t))$. This value is correlated with structural volatility, and we scale $\alpha$ proportionally to this variance:

$$\alpha = \alpha_0 \cdot \frac{\text{Var}(H(\mathcal{G}^t))}{\text{Var}(H(\mathcal{G}^t)) + \epsilon}. \tag{44}$$

This guarantees that $\alpha$ is increased solely when the graph sequence undergoes unstable evolution. During the process of $\alpha$'s gradual increase, we select the value that delivers acceptable performance under the clean setting as the appropriate hyperparameter value.

**Compression Balance Parameter of TGT**  The compression balance parameter $\beta$ of `thumbnail` in Eq. 15 controls the trade-off between preserving relevant information and compressing redundant details in the learned representation. We set $\beta$ to values including 0.001, 0.01, 0.1, 0.5, and 1, systematically increasing the compression strength to evaluate the robustness of the model under the compression constraints based on `thumbnail`.

The robustness results for node feature and structure perturbations are presented in Table 8. When $\beta$ is small, model performance degrades more significantly with increasing interference. As $\beta$ increases, while performance on clean data slightly declines, the model's interference resistance improves substantially. Practically, setting $\beta$ requires balancing robustness against the capability to represent clean data effectively, reflecting a critical trade-off in the framework's design.

We performed the same experiment under temporal interference and observed consistent results, as shown in Fig. 7, demonstrating that the optimization objectives $TB_{\mathcal{G}_{TX}}, TB_{\mathcal{G}_{TA}}$ in Eq. 15 leverage the `thumbnail` $\mathcal{G}_T$, which encodes rich continuous evolutionary features, as a compression constraint. This approach prevents the disconnection between the temporal graph's holistic evolutionary continuity and discrete snapshot features (e.g., nodes, topology), thereby enabling more coherent and robust model performance across perturbation. Finally we set $\beta = 0.1$ in other experiment.

Table 8: Robustness results (**AUC**) on MOOC with different $\beta$ in TGT.

| $\beta$ | Clean | Feature Interference | | | Structure Interference | | |
|---|---|---|---|---|---|---|---|
| | | 10% | 20% | 50% | 5% | 10% | 20% |
| 0.001 | 95.01 | 85.35 | 77.74 | 68.18 | 87.98 | 83.56 | 78.08 |
| 0.01 | 94.35 | 86.57 | 78.91 | 70.46 | 91.23 | 85.71 | 79.68 |
| 0.1 | 95.42 | 88.73 | 80.79 | 71.68 | 90.43 | 86.16 | 81.82 |
| 0.5 | 92.47 | 87.84 | 82.16 | 76.89 | 89.51 | 86.95 | 83.62 |
| 1 | 90.81 | 89.05 | 83.27 | 79.38 | 89.70 | 87.67 | 82.98 |

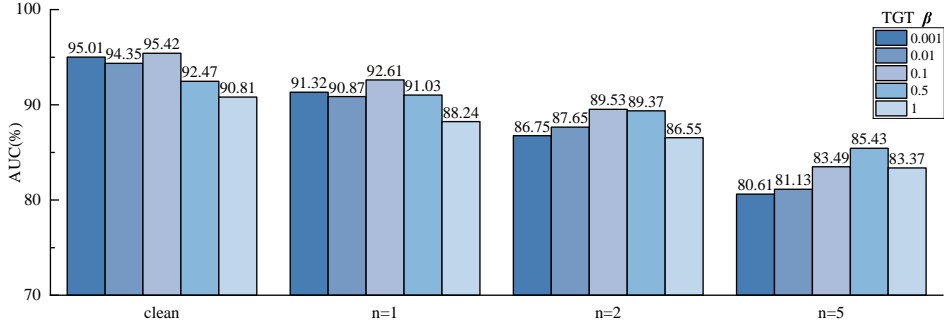

Figure 7: The results after the temporal perturbation on the MOOC with different $\beta$ settings.

**Evolutionary Constraint Parameter of TGT**   The evolution parameter $\alpha$ in Eq. 16 governs the strength of temporal continuity enforced by the `thumbnail`, guiding the model to preserve coherent evolution patterns across time while filtering out noise and irrelevant transitions. We set $\alpha$ to 0.1, 0.5, 1, 2, 5 to gradually enhance the strength of the continuity temporal constraint.

Table 9: Robustness results (**AUC**) on MOOC datasets with different $\alpha$ in TGT.

| $\alpha$ | Clean | Feature Interference | | | Structure Interference | | |
|---|---|---|---|---|---|---|---|
| | | 10% | 20% | 50% | 5% | 10% | 20% |
| 0.1 | 94.37 | 87.21 | 76.54 | 66.63 | 89.80 | 85.45 | 79.77 |
| 0.5 | 95.02 | 87.77 | 74.51 | 67.06 | 91.43 | 87.91 | 80.36 |
| 1 | 95.42 | 88.73 | 80.79 | 71.68 | 90.43 | 86.16 | 81.82 |
| 2 | 89.61 | 84.63 | 80.72 | 71.23 | 86.32 | 82.35 | 78.43 |
| 5 | 87.26 | 83.52 | 79.40 | 69.33 | 84.77 | 80.59 | 77.81 |

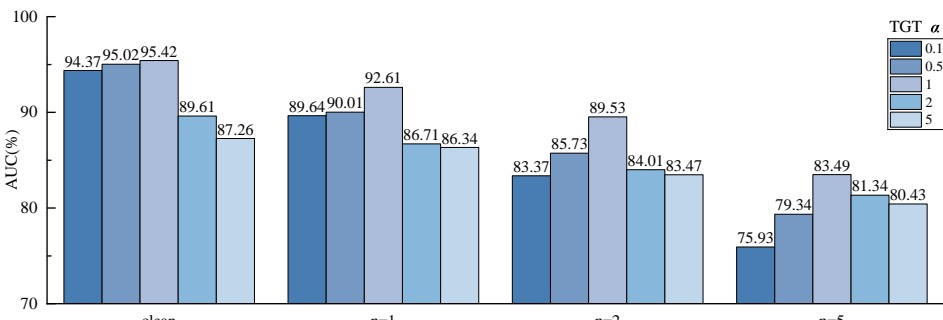

Figure 8: The results after the temporal perturbation on the MOOC with different $\beta$ settings.

Table 9 indicates that under untargeted feature perturbations, the robustness improvement from constraining the model with the von Neumann entropy, which captures topological evolution, is rela-

tively modest. By contrast, during structural attacks, increasing the parameter $\alpha$ reduces the magnitude of performance degradation caused by perturbations. This trend is particularly evident in the temporal perturbation experiment results visualized in Fig. 8. Notably, as $\alpha$ increases, the encoding of node features becomes sparser, leading to degraded performance on clean data. After synthesizing these experimental results, we ultimately set $\alpha = 1$ as the optimal value to balance robustness and clean-data representation.

**Window Size setting of TGT**   Regarding the time window size mentioned in Appendix A.5, we supplement the ablation experiment results on Bitcoin dataset. Interference settings are based on the most severe settings for each type of interference in the Table 3 (Feature 50%, structure 20%, temporal $n=5$).

Table 10: The ablation experiment results (**AUC**) with different window sizes.

| Window Size | Clean | Interference | | |
| --- | --- | --- | --- | --- |
| | | **Feature** | **Structure** | **Temporal** |
| 1 | 75.38 | 65.41 | 65.27 | 70.80 |
| 3 | 84.79 | 72.76 | 74.75 | 77.47 |
| 5 (*baseline*) | 91.41 | 80.62 | 80.95 | 85.64 |
| 7 | 92.52 | 79.03 | 80.70 | 86.20 |
| 9 | 90.55 | 81.37 | 84.70 | 85.93 |

Notably, unduly small window sizes result in inadequate thumbnail modeling, thereby degrading representation quality. For window sizes larger than 5, the model's performance exhibits negligible fluctuations. This confirms that the neighborhood relationships determined by window size exert a certain influence on the thumbnail. TGT exhibits strong adaptability to this hyperparameter.

## A.10   ADDITIONAL EXPERIMENT ON SIMILAR WORKS

Numerous approaches have leveraged the information bottleneck to compress raw data for robustness (Wu et al., 2020; Sun et al., 2022; Yu et al., 2021b;a). Some works have also attempted to adapt information bottleneck theory to temporal graphs for robustness (Yuan et al., 2024). In contrast to these related studies, which focus on achieving information compression, TGT prioritizes summarizing a reliable continuous evolutionary **skeleton** for temporal graph sequences as `thumbnail`. TGT is the first to integrate global evolution via the von Neumann entropy, characterizing the evolutionary backbone of temporal graphs. By imposing compression constraints through the `thumbnail`, it avoids excessive compression that causes information loss or insufficient compression that leaves noise redundancy, striking a balance between essential feature retention and disturbance resilience.

To further establish TGT's superiority, we compare it against methods which perform information bottleneck on temporal graphs and conduct supplementary experiments to validate its distinct advantages. We additionally selected two temporal datasets, **Collab** and **Yelp**. These datasets contain richer features but exhibit relatively lower evolution frequencies compared to our primary choices.

• Collab is an academic collaboration dataset containing papers published between 1990 and 2005. Nodes represent authors, while edges represent co-authorships. Each edge includes domain information such as "data mining," "database," "medical informatics," "theory," and "visualization," based on the field of the co-authored publication. Node features are processed using word2vec (Mikolov et al., 2013), and edges are encoded via one-hot encoding.

• Yelp is a forum dataset containing business information, including attributes such as reviews, photos, check-ins, business hours, parking availability, and ambiance. We define users and businesses as nodes and review actions as edges, processing node features with word2vec similarly to the Collab. We sample interactions from five major business categories, *Pizza*, *American (New) Food*, *Coffee & Tea*, *Sushi Bars*, and *Fast Food*, spanning from January 2019 to December 2020.

Our selected comparison baselines are **DGIB** (Yuan et al., 2024) and **GIB** (Wu et al., 2020) combined with an LSTM model, where GIB first generates discrete snapshot representations and the LSTM subsequently captures temporal dependencies. Experiments shows the superiority of TGT.

Table 11: Robustness results (**AUC**) on Collab and Yelp datasets with different data perturbation.

| Dataset | Model | Clean | Feature Interference | | | Structure Interference | | | Temporal Interference | | |
|---------|-------|-------|------|------|------|------|------|------|------|------|------|
| | | | 10% | 20% | 50% | 5% | 10% | 20% | $n=1$ | $n=2$ | $n=5$ |
| **Collab** | GIB+LSTM | 91.36±0.2 | 80.73±0.2 | 72.27±0.3 | 61.73±0.3 | 85.75±0.2 | 76.51±0.4 | 71.22±0.2 | 84.07±0.3 | 74.34±0.3 | 60.23±0.4 |
| | DGIB | 92.17±0.2 | 78.95±0.3 | 73.72±0.3 | 64.18±0.6 | 87.47±0.1 | 80.73±0.2 | 74.43±0.3 | 83.32±0.2 | 80.73±0.2 | 59.46±0.5 |
| | **TGT** | **93.41±0.3** | **90.26±0.3** | **85.06±0.3** | **74.92±0.3** | **91.19±0.4** | **85.07±0.2** | **78.72±0.4** | **84.42±0.3** | **82.31±0.2** | **73.11±0.3** |
| **Yelp** | GIB+LSTM | 77.52±0.4 | 71.71±0.2 | 62.63±0.2 | 54.76±0.5 | 75.41±0.2 | 72.56±0.4 | 65.31±0.2 | 74.02±0.3 | 68.34±0.3 | 57.26±0.7 |
| | DGIB | 76.88±0.2 | 71.54±0.4 | 67.34±0.5 | **62.98±0.4** | 75.27±0.4 | 74.51±0.2 | 73.43±0.3 | 75.39±0.3 | 72.11±0.3 | 65.22±0.6 |
| | **TGT** | **80.17±0.3** | **76.05±0.2** | **69.37±0.3** | 62.53±0.3 | **78.76±0.2** | **75.66±0.1** | **73.57±0.2** | **79.79±0.2** | **76.37±0.3** | **72.02±0.3** |

## A.11 SUPPLEMENTARY EXPERIMENTS UNDER DIFFERENT DOWNSTREAM TASKS

To verify the robustness of the temporal node representations learned by TGT, we conduct additional experiments on two representative downstream tasks. Specifically, we evaluate TGT on node classification and dynamic node property prediction. These experiments further demonstrate the model's robustness and its ability to generalize across diverse tasks.

### A.11.1 SUPPLEMENTARY EXPERIMENT ON NODE CLASSIFICATION

To demonstrate that TGT produces general-purpose temporal node representations that transfer to other downstream tasks, we expanded our evaluation to a node classification task on both the Reddit dataset (Kumar et al., 2019) and the MOOC dataset used in the main paper. The supplementary Reddit dataset is as follows:

• **Reddit**: A subreddit-focused sharing/discussion platform. We build the temporal dataset ***Reddit*** from public sources (SNAP's 2046 subreddit monthly networks, Pushshift.io timestamped comments). It is a directed temporal attributed network with subreddit hyperlinks (timestamps, sentiment polarity, post text attributes), featuring two core interactions: direct user replies and linear comment chains (less than 3 intermediates). Interactions have a valence score (upvotes – downvotes); node features (3D): activity (total comments), interaction quality (avg. valence), community participation (joined subreddits), derived from historical stats. One-month snapshot: 1k active subreddits, 10k users, 672k+ interactions; post texts encoded as LIWC features; labels from ban records (366 labeled nodes); task: binary node classification (banned/unbanned).

Table 12: Node classification performance (AUC) under clean and perturbed settings on Reddit and MOOC datasets.

| Dataset | Method | Clean | Interference | | |
|---------|--------|-------|---------|-----------|----------|
| | | | Feature | Structure | Temporal |
| **Reddit** | EvolveGCN | 56.4 | 48.6 | 51.7 | 51.8 |
| | JODIE | 59.9 | 51.6 | 51.4 | 50.1 |
| | DyREP | 63.2 | 54.5 | 57.6 | 52.4 |
| | TGN | 67.2 | 53.1 | 58.1 | 53.5 |
| | DIDA | 58.2 | 49.0 | 54.3 | 49.7 |
| | GIB+LSTM | 59.9 | 54.7 | 53.1 | 51.6 |
| | DGIB | 61.3 | 55.7 | 54.6 | 52.0 |
| | **TGT (ours)** | **64.6** | **58.3** | **59.0** | **57.3** |
| **MOOC** | EvolveGCN | 67.3 | 54.6 | 54.9 | 53.8 |
| | JODIE | 68.5 | 53.7 | 56.5 | 55.4 |
| | DyREP | 63.4 | 50.3 | 55.8 | 53.6 |
| | TGN | 64.4 | 53.1 | 54.6 | 51.7 |
| | DIDA | 56.7 | 53.7 | 54.9 | 53.0 |
| | GIB+LSTM | 59.1 | 51.2 | 54.7 | 52.3 |
| | DGIB | 61.3 | 54.7 | 55.2 | 53.1 |
| | **TGT (ours)** | **66.4** | **60.3** | **62.6** | **61.4** |

The results are reported in Table 12. Interference settings are based on the most severe settings for each type of interference in the Table 3 (Feature 50%, structure 20%, temporal $n=5$).

Under clean settings, TGT achieves competitive AUCs, comparable to or better than strong temporal baselines (e.g., TGN, DyREP). Under severe feature interference (50% Gaussian corruption), TGT retains substantially higher performance than most baselines (next best typically in the low-50s),

demonstrating superior denoising ability.Under structural interference and temporal interference, TGT consistently outperforms alternatives by meaningful margins.

**These results show that the thumbnail provides a robust evolution skeleton that helps not only link prediction but also node-level classification.** Intuitively, by encoding persistent evolutionary skeletons, TGT yields embeddings that **(i)** suppress transient/noisy signals that hurt classification under corruption, and **(ii)** preserve long-term structural cues that are predictive for node labels.

### A.11.2 SUPPLEMENTARY EXPERIMENT ON TARGETED ATTACKS WITH EXTRA BASELINES

To further measure the robustness of our TGT under more severe noise conditions, we conducted experiments on a wider range of noise data. We implemented **Nettack**, **FGA**, and **SGA** using DeepRobust[3], set the attack intensity n_perturbations to 20% of the average number of nodes, and implemented these five attack methods using the official code of **PRBCD**[4] and **GOttack**[5] (details of attack methods in Appendix A.6).

Table 13: Link prediction performance(AUC) comparison under untargeted and targeted attacks.

| Dataset | Method | Clean | Untargeted Interference | | | Targeted Attacks | | | | |
|---|---|---|---|---|---|---|---|---|---|---|
| | | | Feat.(20%) | Struct.(20%) | Temp.(n=5) | Nettack | FGA | SGA | PRBCD | GOttack |
| **Bitcoin** | EvolveGCN | 67.59±0.3 | 56.77±0.2 | 55.89±0.3 | 54.37±0.4 | 55.99±0.2 | 47.67±0.6 | 52.08±0.4 | 49.70±0.7 | 51.08±0.2 |
| | JODIE | 74.47±0.3 | 61.10±0.3 | 60.57±0.3 | 58.28±0.7 | 54.28±0.4 | 55.24±0.3 | 58.00±0.6 | 53.34±0.6 | 48.54±0.3 |
| | DyREP | 70.43±0.5 | 61.25±0.3 | 58.60±0.5 | 57.33±0.7 | 53.44±0.7 | 54.09±0.7 | 55.71±0.3 | 53.71±1.1 | 49.95±0.5 |
| | TGN | 69.36±1.1 | 62.31±0.4 | 58.73±0.5 | 61.74±0.8 | 52.17±1.0 | 54.71±0.3 | 54.92±0.8 | 52.18±0.2 | 50.70±0.7 |
| | DIDA | 73.57±0.3 | 68.43±0.3 | 65.69±0.4 | 64.29±0.4 | 62.29±0.5 | 64.15±0.9 | 63.43±0.6 | 59.85±1.0 | 60.50±0.4 |
| | GIB+LSTM | 70.79±0.3 | 63.73±0.5 | 62.93±0.2 | 63.41±0.8 | 60.19±0.5 | 59.43±0.4 | 61.27±0.3 | 61.89±0.4 | 57.89±0.7 |
| | DGIB | 72.99±1.3 | 63.63±0.5 | 59.13±0.4 | 62.53±0.6 | 62.87±0.4 | 63.04±0.3 | 62.53±0.7 | 60.65±0.4 | 58.81±0.3 |
| | Edgebank | 62.27±0.7 | 54.38±0.9 | 52.96±1.3 | 53.12±1.1 | 51.53±0.8 | 50.47±0.5 | 52.26±0.4 | 50.14±0.6 | 47.45±0.6 |
| | DYGFormer | 87.91±0.6 | 74.60±0.3 | 75.11±0.5 | 73.76±0.7 | 70.34±0.3 | 72.93±0.7 | 71.64±0.7 | 69.61±0.9 | 64.22±1.0 |
| | TPNet | 93.10±1.3 | 80.17±0.7 | 80.13±0.4 | 77.67±0.9 | 69.27±1.1 | 70.10±1.3 | 69.48±0.5 | 67.90±0.6 | 66.37±1.1 |
| | **TGT** | **91.41±0.2** | **86.23±0.4** | **80.95±0.7** | **85.64±0.3** | **82.11±0.5** | **81.24±0.3** | **80.32±0.9** | **78.24±0.4** | **75.78±0.7** |
| **MathOverflow** | EvolveGCN | 75.59±0.2 | 61.14±0.3 | 55.23±0.3 | 56.63±0.5 | 53.16±0.4 | 50.13±0.6 | 52.33±0.7 | 52.53±0.7 | 50.81±0.6 |
| | JODIE | 67.06±1.2 | 59.56±0.3 | 53.19±0.3 | 55.37±0.3 | 54.04±0.4 | 53.67±0.3 | 52.72±0.5 | 51.11±0.7 | 50.07±0.4 |
| | DyREP | 63.50±0.5 | 53.32±0.6 | 53.26±0.3 | 53.33±0.5 | 52.73±0.7 | 50.45±0.4 | 50.94±0.9 | 49.83±0.4 | 49.80±0.8 |
| | TGN | 64.50±0.6 | 58.96±0.3 | 53.97±0.3 | 55.23±0.7 | 53.22±0.5 | 53.26±0.3 | 53.34±0.4 | 50.31±0.7 | 50.89±0.5 |
| | DIDA | 74.37±0.4 | 68.63±0.1 | 67.03±0.4 | 65.44±0.5 | 60.93±0.7 | 61.26±0.5 | 60.64±0.8 | 58.01±0.3 | 58.54±1.0 |
| | GIB+LSTM | 77.52±0.3 | 69.38±0.6 | 63.21±0.8 | 62.33±0.8 | 58.43±0.3 | 57.12±0.6 | 60.44±0.3 | 56.64±0.4 | 55.64±0.5 |
| | DGIB | 80.29±0.3 | 73.63±0.5 | 70.43±0.3 | 70.24±0.5 | 63.22±0.7 | 65.41±1.2 | 62.04±0.3 | 63.71±1.0 | 60.33±0.4 |
| | Edgebank | 62.43±0.7 | 55.07±0.4 | 54.11±0.3 | 53.70±0.7 | 51.78±0.2 | 51.43±0.6 | 50.79±0.9 | 49.62±0.8 | 47.95±0.3 |
| | DYGFormer | 83.67±0.5 | 69.79±0.3 | 67.74±0.5 | 67.81±1.1 | 56.74±0.3 | 53.01±0.4 | 60.71±1.1 | 51.66±0.9 | 50.71±0.4 |
| | TPNet | 78.43±1.1 | 63.39±0.2 | 64.03±0.5 | 62.86±0.3 | 54.72±0.4 | 51.19±0.6 | 57.78±0.7 | 52.27±0.4 | 47.10±1.2 |
| | **TGT** | **82.38±0.6** | **74.37±0.3** | **75.93±0.7** | **76.37±0.4** | **70.46±0.5** | **69.33±0.8** | **65.59±0.4** | **66.50±0.5** | **61.17±1.4** |
| **MOOC** | EvolveGCN | 72.35±0.3 | 62.37±0.2 | 52.31±0.2 | 55.93±0.2 | 52.10±0.8 | 49.51±0.3 | 51.25±0.7 | 50.55±0.3 | 48.92±0.4 |
| | JODIE | 73.19±0.7 | 55.36±0.2 | 57.34±0.4 | 58.59±0.4 | 56.95±0.4 | 55.02±0.4 | 53.69±0.3 | 52.23±0.7 | 51.69±0.3 |
| | DyREP | 81.36±0.1 | 66.77±0.2 | 59.74±0.3 | 59.37±0.5 | 57.10±0.8 | 55.42±0.3 | 54.83±0.4 | 52.64±0.5 | 51.02±1.0 |
| | TGN | 79.36±1.0 | 68.41±0.3 | 61.33±0.3 | 63.53±1.1 | 59.60±0.7 | 61.65±0.3 | 58.93±0.7 | 54.77±0.3 | 53.87±0.5 |
| | DIDA | 89.84±0.5 | 73.62±0.3 | 61.03±0.4 | 73.66±0.2 | 63.26±0.4 | 64.76±0.5 | 64.13±0.3 | 63.24±0.7 | 61.38±0.4 |
| | GIB+LSTM | 92.34±0.3 | 65.25±0.3 | 74.37±0.3 | 69.68±0.7 | 66.94±0.3 | 68.58±0.4 | 65.79±0.4 | 63.13±0.5 | 64.33±1.1 |
| | DGIB | 93.06±0.1 | 75.24±0.2 | 79.69±0.3 | 70.52±0.6 | 74.31±0.9 | 71.14±0.5 | 70.51±0.7 | 68.69±0.4 | 66.03±0.7 |
| | Edgebank | 86.14±0.7 | 65.23±0.7 | 62.31±1.1 | 61.52±0.9 | 60.71±0.3 | 59.75±0.7 | 60.32±0.4 | 60.78±0.5 | 59.64±0.3 |
| | DYGFormer | 94.13±1.2 | 77.24±0.5 | 75.55±0.3 | 72.06±0.5 | 69.07±0.5 | 70.07±0.4 | 67.37±0.7 | 66.28±1.5 | 63.41±0.8 |
| | TPNet | 96.45±0.7 | 73.16±1.3 | 77.25±0.4 | 74.17±0.3 | 73.66±0.7 | 72.96±1.2 | 67.47±0.6 | 65.73±0.3 | 64.73±0.3 |
| | **TGT** | **95.42±0.4** | **80.79±0.1** | **81.82±0.3** | **84.01±0.6** | **80.33±0.3** | **78.11±0.4** | **81.38±0.7** | **77.65±0.6** | **78.60±0.4** |

In addition, we supplemented three more advanced baselines that outperform in clean data setting, thus more comprehensively validating the superiority of TGT. **Edgebank** (Poursafaei et al., 2022) is a memorization-based baseline for dynamic link prediction, which leverages temporal edge reoccurrence patterns to enhance evaluation rigor and exposes the flaw of easy negative sampling in existing protocols, featuring strong performance across settings. **DYGFormer** (Yu et al., 2023) is a dynamic graph learning framework integrating a novel architecture and a unified library, enabling efficient modeling of temporal dependencies and achieving strong generalization across diverse dynamic graph tasks. **TPNet** (Lu et al., 2024) is a temporal link prediction model that boosts performance via temporal walk matrix projection, enabling effective modeling of temporal dependencies.

To verify the reliability of our TGT, we generated the noisy datasets to conduct inductive link prediction experiments. The results in Table 13 consistently show that **TGT achieves the highest performance under every targeted attack across all datasets.** For instance, on Bitcoin, TGT reaches 82.11 under Nettack and 78.24 under PRBCD, whereas strong baselines such as DGIB and DYG-Former fall 5–15 points lower. Similar trends hold on MathOverflow, where TGT maintains 70.46

---

[3]DeepRobust: https://github.com/DSE-MSU/DeepRobust
[4]PRBCD: https://github.com/sigeisler/robustness_of_gnns_at_scale
[5]GOttack: https://github.com/cakcora/GOttack

under Nettack, substantially outperforming the next-best baselines. These results demonstrate that TGT does not merely resist mild perturbations but remains robust under attacks that specifically target influential nodes or edges associated with link prediction, indicating that **its thumbnail-guided global evolution modeling prevents overreliance on vulnerable local structures.**

Combined with its strong performance under untargeted feature, structural, and temporal interference, these findings provide direct and comprehensive evidence that TGT delivers outperforming robustness even against the most advanced targeted attack strategies.

A.11.3 SUPPLEMENTARY EXPERIMENT ON NODE PROPERTY PREDICTION

To further validate TGT's robustness and generalization ability across additional downstream tasks, we conducted supplementary experiments following thorough research: we selected the large-scale tgbn-genre and tgbn-reddit datasets(details in Table 14) from the TGB (Huang et al., 2023b) for node property prediction, and adopted TGB's standardized task metrics. This design effectively supplements the verification of TGT's cross-task adaptability while demonstrating its superiority.

Table 14: Details of TGB datasets for experiment on node property prediction.

| Dataset | nodes | edges | Steps | snap. nodes | snap. edge |
|---|---|---|---|---|---|
| tgbn-genre | 1,505 | 17,858,395 | 133,758 | 1307 | 78,345 |
| tgbn-reddit | 11,766 | 27,174,118 | 21,889,537 | 6843 | 440,603 |

• **Tgbn-genre**: Tgbn-genre dataset is a bipartite weighted user-music genre interaction network (based on listening behavior). Nodes are users and genres; edges denote time-stamped user-genre listens, with weights as song-genre percentage contributions. It is built by cross-referencing LastFM-song-listens (1k users, 1-month song listens) and Million-Song Dataset (song genre weights). Retained genres meet two criteria: more than 10% song weight and 1k dataset occurrences. Genre names are cross-checked to eliminate typos.

• **Tgbn-reddit**: Tgbn-reddit dataset is a user-subreddit interaction network (users and subreddits as nodes, edges indicating time-stamped user posts on subreddits) spanning 2005–2019, with the prediction task of ranking the subreddits a user will interact with most in the next week.

Table 15: Node property prediction performance (NDCG@10) under clean and perturbed settings on Tgbn-genre and Tgbn-reddit.

| Dataset | Method | Untarget Interference | | | | Target Attacks | | | | |
|---|---|---|---|---|---|---|---|---|---|---|
| | | Clean | Feat. | Struct. | Temp. | Nettack | FGA | SGA | PRBCD | GOttack |
| **Tgbn-genre** | EvolveGCN | 0.343 | 0.321 | 0.316 | 0.305 | 0.296 | 0.253 | 0.287 | 0.277 | 0.262 |
| | DyREP | 0.351 | 0.317 | 0.301 | 0.288 | 0.240 | 0.267 | 0.237 | 0.207 | 0.192 |
| | TGN | 0.367 | 0.323 | 0.311 | 0.325 | 0.259 | 0.274 | 0.236 | 0.246 | 0.219 |
| | DGIB | 0.332 | 0.311 | 0.307 | 0.303 | 0.277 | 0.278 | 0.263 | 0.303 | 0.293 |
| | DYGFormer | 0.365 | 0.306 | 0.291 | 0.280 | 0.247 | 0.231 | 0.275 | 0.252 | 0.209 |
| | TGT | 0.363 | 0.348 | 0.354 | 0.345 | 0.342 | 0.314 | 0.319 | 0.307 | 0.313 |
| **Tgbn-reddit** | EvolveGCN | 0.310 | 0.291 | 0.276 | 0.297 | 0.201 | 0.210 | 0.207 | 0.197 | 0.182 |
| | DyREP | 0.312 | 0.288 | 0.278 | 0.263 | 0.231 | 0.227 | 0.210 | 0.183 | 0.193 |
| | TGN | 0.315 | 0.285 | 0.277 | 0.271 | 0.206 | 0.217 | 0.224 | 0.201 | 0.196 |
| | DGIB | 0.304 | 0.269 | 0.265 | 0.261 | 0.223 | 0.227 | 0.216 | 0.211 | 0.187 |
| | DYGFormer | 0.316 | 0.279 | 0.271 | 0.234 | 0.200 | 0.204 | 0.198 | 0.161 | 0.183 |
| | TGT | 0.313 | 0.289 | 0.280 | 0.278 | 0.243 | 0.246 | 0.231 | 0.221 | 0.210 |

We processed the data using the same perturbation methods as employed in the experiments detailed in Appendix A.11.2, with NDCG@10 adopted as the evaluation metric. This setup aims to verify whether the methods' predictions of class importance align with the ordering of the ground truth.

The results are shown in Table 15. Specifically, while strong baselines such as TGN, DGIB, and DYGFormer suffer noticeable degradation under perturbations, TGT maintains **the highest NDCG@10 across all untargeted and targeted attacks** on TGBN-genre (0.354/0.345/0.342 under structural, temporal, and Nettack attacks respectively, far above TGN's 0.311/0.325/0.259). Re-

garding node property prediction robustness, **both TGBN results show that the thumbnail-guided global evolution modeling provides stable ranking consistency even under aggressive adversarial conditions.** For example, on TGBN-reddit, TGT achieves 0.243 under Nettack, outperforming all baselines by significant margins (the best baseline is approximately 0.231).

These observations confirm that TGT's core mechanism, using a VNGE-constrained thumbnail to encode long-term structural regularities, yields node representations that are not only scalable (confirmed in Table 4) but also substantially more resistant to perturbations.

### A.12 RELATED WORK

**Temporal graph representation learning**

In temporal graph representation learning, representations are derived from spatiotemporal neighbor info via message passing and aggregation based on topology. For instances, some approaches utilize recurrent neural networks (RNNs) to learn dynamic embeddings in temporal interactions (Kumar et al., 2019; Pareja et al., 2020; Han et al., 2020). Alternatively, other methods rely on attention mechanism to update neighbors (Rossi et al., 2020; Trivedi et al., 2019). Furthermore, numerous scholarly investigations have focused on model's robustness and generalizability. Zhang et al. (2022) integrates a disentangled spatiotemporal attention network with intervention and regularization to improve representation quality under distribution shifts. Zhu et al. (2019) boosts robustness by injecting random noise. There are also some attempts to fuzzy the adjacency matrix (Wu et al., 2019; Entezari et al., 2020). Other methods also consider the evolutionary trajectory of temporal graph in representation learning. Shamsi et al. (2024)combining topological data analysis with recurrent neural networks to effectively predict the evolution. Wang et al. (2025) models both temporal interaction trajectories and semantic edge types in a continuous-time dynamic heterogeneous GNN, enabling richer representations of complex evolving graphs. Yao et al. (2022) builds a hierarchical spatial graph via a quadtree and uses graph attention to model long-term dependencies in trajectories, enabling efficient and accurate similarity computation, especially for long trajectories. Li et al. (2024b) introduces a mutual evolution framework with a predictive module and an unlearning module co-training to improve unlearning efficacy and efficiency in graph neural networks while preserving prediction performance. There are also methods based on generative models that attempt to conduct more precise representation learning in various aspects, such as knowledge graphs (Zhang et al., 2024a;b) and retrieval augmented generation (Xiang et al., 2025; Zhang et al., 2025). The above works illustrate the importance of modeling the evolution and demonstrate the value of this research question. However, none of these methods mentioned above focus on robustness; when data is disturbed, the modeling of evolutionary trajectories may be inaccurate. To address the lack of theoretical support for generalizability and robustness in previous research, we derive an solution to enhance model interpretability.

**Von Neumann entropy in graph evolution**

For a quantum system with density matrix $\rho$, the von Neumann entropy (VNGE) is $S_{VN} = -\text{Tr}(\rho log \rho)$. $\rho$ describes a mixed state system consisting of pure quantum states $|\psi\rangle$, each with probability $p_i$, defined as $\rho = \sum_{i=1}^{T} p_i |\psi_i\rangle\langle\psi_i|$, where $T$ is the number of pure states. $S_{VN}$ plays a crucial role in the quantum measurement process, and can calculate the expectation of the measurable quantity. Braunstein interpreted the scaled regularized Laplacian as a density operator, enabling VNGE for graph characterization (Braunstein et al., 2006). Subsequent research revealed that its measurement is pivotal in comprehending network system structure and topological complexity (De Domenico et al., 2015). For instance, it aids in depicting quantum statistics in topological networks (Passerini & Severini, 2009) and assessing graph irregularity (Passerini & Severini, 2008). Liu et al. (2018) used VNGE to study dynamic genomes universal patterns, extending its use in dynamic cases. Then Huang et al. (2023a) designed an attention network considering VNGE for evolutionary feature extraction. We combine VGNE to set evolutionary constraints so that the latent Hinge graph can summarize more coherent temporal change characteristics.

**Information bottleneck in graph**

Alemi et al. (2016) applies variational estimation and integrates the information bottleneck theory (IB) (Tishby et al., 2000) to enhance robustness and generalizability in deep representation learning. Wu et al. (2020) formulated a hypothesis for transforming graphs into IID data, and subsequently

derives an IB optimization tailored to discrete structures. Methods such as Seo et al. (2024) and Sun et al. (2022) each offer distinct solutions for extracting the IB objectives from static graph data. Yu et al. (2021b; 2022; 2021a) have also demonstrated the remarkable performance of IB in discovering key information and inducing important substructures. Yuan et al. (2024) first attempts to extend IB to temporal graphs. Chen et al. (2024) investigated the application of the IB in the extrapolation of temporal knowledge graphs. However, these methods overlook graph evolution, as IB constraints are limited to middle latent variables. By leveraging the VNGE to characterize continuous evolutionary features, our TGT can access richer evolutionary information, which significantly enhances its ability to resist interference under perturbed conditions. Furthermore, we integrate temporal evolution via VGNE to capture entire sequence. The results are presented in Table 11.

### A.13   Limitations and Future Work

The main limitation of TGT is the scalable problem caused by the additional introduction of `thumbnail` characterization. Since the theoretical upper bound of TGT's computational complexity scales linearly with the number of nodes and edges across all graph snapshots (as shown in appendix A.1.4), scaling to larger datasets (e.g., datasets with millions of nodes) can be prohibitively time-consuming.While TGT demonstrates strong robustness under a wide range of feature, structure, and temporal perturbations, as well as various targeted attacks, we acknowledge that our reliance on a thumbnail-based architectural design may introduce new vulnerabilities under adaptive attacks specifically crafted to target the thumbnail. For example, an adversary could attack **(i)** the *thumbnail structure* via ontology-level perturbation, or **(ii)** the *bottleneck constraint* by designing gradient-aligned manipulations to weaken the VNGE-based regularization. These are meaningful and challenging scenarios that we agree deserve deeper investigation.

In our future work, we plan to explore pretraining-based approaches for thumbnail modeling, allowing the computational overhead to be offloaded to the pre-training stage and thereby improving the scalability. We will also explore the usage of `thumbnail` as a prompt, aiming to further improve the model's capability and enable the injection of external prior knowledge.

### A.14   Boarder Impact

This paper aims to advance the field of graph learning by proposing a robust representation method for temporal graphs.Our TGT framework inherently resists noise during representation learning, offering robustness against the ubiquitous real-world data noise and potential adversarial attacks such as data poisoning. This capability has positive social implications by enhancing the reliability and trustworthiness of machine learning applications in critical domains. Furthermore, we anticipate that our work will not pose direct social or ethical negative impacts.

### A.15   The Usage of LLM

LLM was employed exclusively for writing assistance, including grammar correction, spelling verification, and text polishing, without contributing in any other way to our work.

