# OpenReview forum: "Temporal Graph Thumbnail: Robust Representation Learning with Global Evolutionary Skeleton"
_ICLR.cc/2026/Conference — ICLR 2026 Poster_

### Official Review · Reviewer_rkWZ · 2025-10-16

**Soundness:** 3
**Presentation:** 4
**Contribution:** 3
**Rating:** 4
**Confidence:** 4

**Summary:**

This work presents a novel approach for modelling dynamic graph representations, focusing on link prediction, called Temporal Graph Thumbnail (TGT). The authors identify that recent methods often overlook the global evolution of temporal graphs, which limits their ability to capture long-term dependencies effectively. To address this, the paper introduces a method that models thumbnails of the temporal graph using von Neumann graph entropy and node mutual information. This thumbnail captures the global evolutionary trajectory of the graph and is leveraged as an optimization guide to learn robust and generalizable node embeddings. By incorporating global evolution information, TGT significantly enhances robustness and representation quality in dynamic graph learning.

Empirical experiments demonstrate that TGT surpasses baseline methods both in performance on clean temporal graphs and in robustness under adversarial attacks. These attacks include perturbations on graph structures, node features, and temporal ordering.

**Strengths:**

- **Novelty.** The paper introduces a novel approach that models a thumbnail of a temporal graph using von Neumann graph entropy and incorporates this thumbnail to guide optimization for learning superior and robust node representations.

- **Superior Node Representation.** By leveraging von Neumann entropy to constrain the information bottleneck associated with global structural evolution, TGT outperforms seven strong baseline methods across three diverse datasets.

- **Robust Node Representation.** Modeling the thumbnail and utilizing von Neumann entropy enhances robustness against various adversarial attack settings, including: Random Gaussian noise disrupting node features, Nettack adversarial perturbations on graph structure, and Permuting snapshot chronological order causing temporal disruptions.

- **Ablation Studies.** clearly highlight the contributions of different model components, including evolution constraint, structure and feature mutual information, thumbnail and von Neumann Entropy.

**Weaknesses:**

**W1: Robustness evaluation.** The robustness evaluation of TGT is limited to random and out-of-date adversarial attacks on a static graph. To improve the strength and generality of the evaluation, it would be more convincing to evaluate the robustness of TGT under feature attack methods[1,2,3], recent structure attacks on static graphs [4,5], and adversarial attacks on dynamic graphs [6].

**W2: CTDG vs DTDG.** The paper does not clearly state whether the focus is on continuous-time dynamic graphs (CTDG) or discrete-time dynamic graphs (DTDG). Including a discussion on the differences between these two temporal graph settings and explicitly highlighting which type the work addresses (ideally in Section 2) would benefit clarity and contextualize the contributions.

**W3: TGNN baselines.**  The work compares the performance of TGT against out-of-date baselines. A naive but effective baseline, such as EdgeBank[7], and recent TGNN models[8,9,10].

*Minor*

**W4: Evaluation metrics.** The paper mainly reports AUC and AP for link prediction. Including Mean Reciprocal Rank (MRR) under the Temporal Graph Benchmark (TGB) evaluation settings would provide a more comprehensive view of TGT's ranking performance and enhance comparability with other temporal graph models evaluated on TGB[12].

---
[1] Zügner, Daniel, Amir Akbarnejad, and Stephan Günnemann. "Adversarial attacks on neural networks for graph data." *Proceedings of the 24th ACM SIGKDD international conference on knowledge discovery & data mining*. 2018.
[2] Chen, Jinyin, et al. "Fast gradient attack on network embedding." *arXiv preprint arXiv:1809.02797* (2018).

[3] Li, Jintang, et al. "Adversarial attack on large scale graph." *IEEE Transactions on Knowledge and Data Engineering* 35.1 (2021): 82-95.

[4] Geisler, Simon, et al. "Robustness of graph neural networks at scale." *Advances in Neural Information Processing Systems* 34 (2021): 7637-7649.

[5] Alom, Zulfikar, et al. "GOttack: Universal Adversarial Attacks on Graph Neural Networks via Graph Orbits Learning." *The Thirteenth International Conference on Learning Representations*. 2025.

[6] Dai, Yue, et al. "MemFreezing: A Novel Adversarial Attack on Temporal Graph Neural Networks under Limited Future Knowledge." *Forty-second International Conference on Machine Learning*. 2025.

[7] Poursafaei, Farimah, et al. "Towards better evaluation for dynamic link prediction." *Advances in Neural Information Processing Systems* 35 (2022): 32928-32941.

[8] Yu, Le, et al. "Towards better dynamic graph learning: New architecture and unified library." *Advances in Neural Information Processing Systems* 36 (2023): 67686-67700.

[9] Lu, Xiaodong, et al. "Improving temporal link prediction via temporal walk matrix projection." *Advances in Neural Information Processing Systems* 37 (2024): 141153-141182.

[10] Ding, Zifeng, et al. "Dygmamba: Efficiently modeling long-term temporal dependency on continuous-time dynamic graphs with state space models." *arXiv preprint arXiv:2408.04713* (2024).

**Questions:**

- The concept of learning a graph evolution trajectory over time was previously explored in GraphPulse[11]. Could the authors clarify the key differences between TGT and GraphPulse in terms of approach and performance?

- How does TGT scale on large-scale datasets, particularly those in the TGB[12]?

- The paper employs random Gaussian noise as a node feature attack. How robust is TGT against other sophisticated feature attack methods, such as Nettack, PRBCD, SGA, and FGA, as mentioned in the identified weaknesses (See **W1**)?

- Robustness evaluations focus on untargeted attacks, which might be misleading. How often do these attacks realistically perturb nodes or edges involved in link prediction? Targeted attacks on specific nodes and links provide more informative robustness assessments.

- The term “snapshot” is frequently used. Is this research focused on discrete-time dynamic graphs (DTDG)? If so, what is the typical size or timespan of these snapshots?

- How robust are the learned node embeddings from TGT under adversarial attacks on node property prediction tasks as defined within TGB[12]?

---
[11] Shamsi, Kiarash, et al. "GraphPulse: Topological representations for temporal graph property prediction." *The Twelfth International Conference on Learning Representations*. 2024.

[12] Huang, Shenyang, et al. "Temporal graph benchmark for machine learning on temporal graphs." *Advances in Neural Information Processing Systems* 36 (2023): 2056-2073.

---

> ### Author Response · Authors · 2025-11-20
> **Response to Reviewer rkWZ (1/8):**
>
> Dear rkWZ,
>
> We sincerely appreciate you devoting significant time and effort to providing valuable insights and constructive comments on our work. To address your concerns thoroughly, we will conduct rigorous supplementary experiments and detailed theoretical derivations to further clarify the technical details, validate the robustness of our framework, and strengthen the academic rigor of our TGT.
>
> ---
>
> ## Ans 1: Regarding the questions about CTDG and DTDG (W2, Q5):
>
> We appreciate your concern regarding the type of TDG. **In fact, our TGT focuses on discrete-time dynamic graphs (DTDGs).** The snapshot size and time span of the dataset are shown in the Table 1.
>
> Table 1: Details of datasets for experiments
>
> | Dataset             | Bitcoin     | MathOverflow | MOOC     |
> | ------------------- | ----------- | ------------ | -------- |
> | Nodes               | 9,664       | 24,818       | 7,047    |
> | Edges               | 59,778      | 506,550      | 411,749  |
> | Timespan            | 1903 days   | 2350 days    | 30 days  |
> | Link Type           | Homogeneous | 3            | 5        |
> | Evolution Frequency | 18.7        | 45.78        | 13724.97 |
> | snapshot nodes      | 7,034       | 21,683       | 7,047    |
> | snapshot edges      | 51,363      | 207,581      | 81,749   |
>
>
> ***Thanks again for your suggestion. We explicitly clarify this point in Section 2.1, and further supplement the data scale of dataset snapshots and their time spans in Table 1 of Section 4.1. We hope it can address your concerns (W2，Q5).***

---

> ### Author Response · Authors · 2025-11-20
> **Response to Reviewer rkWZ (2/8):**
>
> ## Ans 2: Regarding differences from existing work (Q1):
>
> We appreciate the reviewer for pointing out the relevant work of GraphPulse[1]. We clarify the core differences between our TGT method and GraphPulse from the following aspects, and focus on explaining how the contributions of this paper complement and distinguish this study.
>
> - Specific differences:
>
>   - In terms of **task formulation**, GraphPulse targets predicting **future aggregated graph properties** (graph-level prediction), while our work focuses on **node/edge-level robust representation learning** in dynamic graphs (e.g., node embedding and link prediction). This difference leads to **distinct modeling objectives and evaluation** for the two methods.
>
>   - In terms of **modeling paradigm**, GraphPulse generates a topological summary **for each snapshot via the Mapper method, then models the trajectory of these summaries using a RNN.** In contrast, our TGT framework constructs **a unified "thumbnail" for the entire sequence**, leverages VNGE to capture structural evolution, and embeds this thumbnail into an information bottleneck framework to guide node embedding learning.
>
>
> - ### Fundamental differences:
>
>   - In fact, besides GraphPulse, many other advanced works focus on modeling the evolution process of temporal graphs. **TESA**[2] models both temporal interaction trajectories and semantic edge types in a continuous-time dynamic heterogeneous GNN, enabling richer representations of complex evolving graphs. **TrajGAT**[3] builds a hierarchical spatial graph via a quadtree and uses graph attention to model long-term dependencies in trajectories, enabling efficient and accurate similarity computation, especially for long trajectories. **MEGU**[4] introduces a *mutual evolution* framework with a predictive module and an unlearning module co-training to improve unlearning efficacy and efficiency in graph neural networks while preserving prediction performance.
>
>   - These prior studies, all published at leading conferences in recent years, underscore the central importance of modeling evolutionary processes and clearly demonstrate the significance and timeliness of this research direction. **However, none of these methods mentioned above focus on robustness; when data is disturbed, the modeling of evolutionary trajectories may be inaccurate.** To the best of our knowledge, TGT is the first framework to incorporate statistical-physics–inspired principles into robust temporal graph modeling. By jointly leveraging von Neumann graph entropy and Donsker–Varadhan–based node mutual information to construct evolution thumbnails, our method provides a theoretically grounded mechanism for capturing global evolution regularities. Both the formal analysis and the extensive experiments consistently demonstrate that our thumbnail-guided modeling leads to more stable representations and significantly improved robustness across diverse downstream tasks. **This is a unique innovation compared to other methods, filling the gap of robustness in these methods with evolution modeling.**
>
>
> In summary, GraphPulse is a highly solid work that pioneers a topological evolution perspective for temporal graphs. Our TGT, however, adopts a complementary approach: it treats global evolution as a static bottleneck (the thumbnail) within the node/edge embedding learning pipeline, and emphasizes enhancing the robustness and representational quality required for downstream tasks.
>
> ***Thank you for your consideration. We add the citations and description to Appendix A.12, and hope this can address your concerns (Q1).***
>
> [1] Shamsi, Kiarash, et al. "GraphPulse: Topological representations for temporal graph property prediction." *The Twelfth International Conference on Learning Representations*. ICLR, 2024.
>
> [2] Wang, Xin, et al. "TESA: A trajectory and semantic-aware dynamic heterogeneous graph neural network." *Proceedings of the ACM on Web Conference 2025*. 2025.
>
> [3] Yao, Di, et al. "Trajgat: A graph-based long-term dependency modeling approach for trajectory similarity computation." *Proceedings of the 28th ACM SIGKDD conference on knowledge discovery and data mining*. 2022.
>
> [4] Li, Xunkai, et al. "Towards effective and general graph unlearning via mutual evolution." *Proceedings of the AAAI Conference on Artificial Intelligence*. Vol. 38. No. 12. 2024.

---

> ### Author Response · Authors · 2025-11-20
> **Response to Reviewer rkWZ (3/8):**
>
> ## Ans 3: Regarding the Robustness evaluation (W1, W3, Q3, Q4):
>
> Thank you very much for your thoughtful questions regarding the noise settings. In fact, we clarify the *targeted attack* strategy adopted in our experiments in Appendix A.5.
>
> We greatly appreciate your suggestions in **W1, Q3, and Q4** regarding additional adversarial attack methods. After extensive investigation, we implemented **Nettack** [1], **FGA** [2], and **SGA** [3] using the DeepRobust library [7], and set the attack strength $n_{\text{perturbations}}$ to 20% of the average number of nodes. Although we were unable to locate a reliable open-source implementation of MemFreezing[6], we additionally incorporated **PRBCD** [4] and **Gottack** [5] using their official codebases. Based on these five attack methods, we constructed perturbed datasets and systematically evaluated the robustness of our TGT model.
>
> We also highly value your suggestion in **W3** to enrich the baselines. After careful investigation and reproduction, we have added **Edgebank** [8], **DYGFormer** [9], and **TPNet** [10] as additional baselines. These enhancements enable a more comprehensive comparison and further highlight the advantages of our TGT.
>
> **The results are shown in the following three tables.**
>
> *Table 13-a: Link prediction performance (AUC) under untargeted and targeted attacks on the bitcoin dataset*
> | Bitcoin   | clean     | Feature Interference (20%) | Structural Interference (20%) | Temporal Interference (n=5) | Nettack   | FGA       | SGA       | PRBCD     | GOttack   |
> | --------- | --------- | -------------------------- | ----------------------------- | --------------------------- | --------- | --------- | --------- | --------- | --------- |
> | EvolveGCN | 67.59±0.3 | 56.77±0.2                  | 55.89±0.3                     | 54.37±0.4                   | 55.99±0.2 | 47.67±0.6 | 52.08±0.4 | 49.70±0.7 | 51.08±0.2 |
> | JODIE     | 74.47±0.3 | 61.10±0.3                  | 60.57±0.3                     | 58.28±0.7                   | 54.28±0.4 | 55.24±0.3 | 58.00±0.6 | 53.34±0.6 | 48.54±0.3 |
> | DyREP     | 70.43±0.5 | 61.25±0.3                  | 58.60±0.5                     | 57.33±0.7                   | 53.44±0.7 | 54.09±0.7 | 55.71±0.3 | 53.71±1.1 | 49.95±0.5 |
> | TGN       | 69.36±1.1 | 62.31±0.4                  | 58.73±0.5                     | 61.74±0.8                   | 52.17±1.0 | 54.71±0.3 | 54.92±0.8 | 52.18±0.2 | 50.70±0.7 |
> | DIDA      | 73.57±0.3 | 68.43±0.3                  | 65.69±0.4                     | 64.29±0.4                   | 62.29±0.5 | 64.15±0.9 | 63.43±0.6 | 59.85±1.0 | 60.50±0.4 |
> | GIB+LSTM  | 70.79±0.3 | 63.73±0.5                  | 62.93±0.2                     | 63.41±0.8                   | 60.19±0.5 | 59.43±0.4 | 61.27±0.3 | 61.89±0.4 | 57.89±0.7 |
> | DGIB      | 72.99±1.3 | 63.63±0.5                  | 59.13±0.4                     | 62.53±0.6                   | 62.87±0.4 | 63.04±0.3 | 62.53±0.7 | 60.65±0.4 | 58.81±0.3 |
> | Edgebank  | 62.27±0.7 | 54.38±0.9                  | 52.96±1.3                     | 53.12±1.1                   | 51.53±0.8 | 50.47±0.5 | 52.26±0.4 | 50.14±0.6 | 47.45±0.6 |
> | DYGFormer | 87.91±0.6 | 74.60±0.3                  | 75.11±0.5                     | 73.76±0.7                   | 70.34±0.3 | 72.93±0.7 | 71.64±0.7 | 69.61±0.9 | 64.22±1.0 |
> | TPNet     | 93.10±1.3 | 80.17±0.7                  | 80.13±0.4                     | 77.67±0.9                   | 69.27±1.1 | 70.10±1.3 | 69.48±0.5 | 67.90±0.6 | 66.37±1.1 |
> | TGT       | 91.41±0.2 | 86.23±0.4                  | 80.95±0.7                     | 85.64±0.3                   | 82.11±0.5 | 81.24±0.3 | 80.32±0.9 | 78.24±0.4 | 75.78±0.7 |

---

> ### Author Response · Authors · 2025-11-20
> **Response to Reviewer rkWZ (4/8):**
>
> *Table 13-b: Link prediction performance (AUC) under untargeted and targeted attacks on the Mathoverflow dataset*
> | Mathoverflow | clean     | Feature Interference (20%) | Structural Interference (20%) | Temporal Interference (n=5) | Nettack   | FGA       | SGA       | PRBCD     | GOttack   |
> | ------------ | --------- | -------------------------- | ----------------------------- | --------------------------- | --------- | --------- | --------- | --------- | --------- |
> | EvolveGCN    | 75.59±0.2 | 61.14±0.3                  | 55.23±0.3                     | 56.63±0.5                   | 53.16±0.4 | 50.13±0.6 | 52.33±0.7 | 52.53±0.7 | 50.81±0.6 |
> | JODIE        | 67.06±1.2 | 59.56±0.3                  | 53.19±0.3                     | 55.37±0.3                   | 54.04±0.4 | 53.67±0.3 | 52.72±0.5 | 51.11±0.7 | 50.07±0.4 |
> | DyREP        | 63.50±0.5 | 53.32±0.6                  | 53.26±0.3                     | 53.33±0.5                   | 52.73±0.7 | 50.45±0.4 | 50.94±0.9 | 49.83±0.4 | 49.80±0.8 |
> | TGN          | 64.50±0.6 | 58.96±0.3                  | 53.97±0.3                     | 55.23±0.7                   | 53.22±0.5 | 53.26±0.3 | 53.34±0.4 | 50.31±0.7 | 50.89±0.5 |
> | DIDA         | 74.37±0.4 | 68.63±0.1                  | 67.03±0.4                     | 65.44±0.5                   | 60.93±0.7 | 61.26±0.5 | 60.64±0.8 | 58.01±0.3 | 58.54±1.0 |
> | GIB+LSTM     | 77.52±0.3 | 69.38±0.6                  | 63.21±0.8                     | 62.33±0.8                   | 58.43±0.3 | 57.12±0.6 | 60.44±0.3 | 56.64±0.4 | 55.64±0.5 |
> | DGIB         | 80.29±0.3 | 73.63±0.5                  | 70.43±0.3                     | 70.24±0.5                   | 63.22±0.7 | 65.41±1.2 | 62.04±0.3 | 63.71±1.0 | 60.33±0.4 |
> | Edgebank     | 62.43±0.7 | 55.07±0.4                  | 54.11±0.3                     | 53.70±0.7                   | 51.78±0.2 | 51.43±0.6 | 50.79±0.9 | 49.62±0.8 | 47.95±0.3 |
> | DYGFormer    | 83.67±0.5 | 69.79±0.3                  | 67.74±0.5                     | 67.81±1.1                   | 56.74±0.3 | 53.01±0.4 | 60.71±1.1 | 51.66±0.9 | 50.71±0.4 |
> | TPNet        | 78.43±1.1 | 63.39±0.2                  | 64.03±0.5                     | 62.86±0.3                   | 54.72±0.4 | 51.19±0.6 | 57.78±0.7 | 52.27±0.4 | 47.10±1.2 |
> | TGT          | 82.38±0.6 | 74.37±0.3                  | 75.93±0.7                     | 76.37±0.4                   | 70.46±0.5 | 69.33±0.8 | 65.59±0.4 | 66.50±0.5 | 61.17±1.4 |
>
> *Table 13-c: Link prediction performance (AUC) under untargeted and targeted attacks on the MOOC dataset*
> | MOOC      | clean     | Feature Interference (20%) | Structural Interference (20%) | Temporal Interference (n=5) | Nettack   | FGA       | SGA       | PRBCD     | GOttack   |
> | --------- | --------- | -------------------------- | ----------------------------- | --------------------------- | --------- | --------- | --------- | --------- | --------- |
> | EvolveGCN | 72.35±0.3 | 62.37±0.2                  | 52.31±0.2                     | 55.93±0.2                   | 52.10±0.8 | 49.51±0.3 | 51.25±0.7 | 50.55±0.3 | 48.92±0.4 |
> | JODIE     | 73.19±0.7 | 55.36±0.2                  | 57.34±0.4                     | 58.59±0.4                   | 56.95±0.4 | 55.02±0.4 | 53.69±0.3 | 52.23±0.7 | 51.69±0.3 |
> | DyREP     | 81.36±0.1 | 66.77±0.2                  | 59.74±0.3                     | 59.37±0.5                   | 57.10±0.8 | 55.42±0.3 | 54.83±0.4 | 52.64±0.5 | 51.02±1.0 |
> | TGN       | 79.36±1.0 | 68.41±0.3                  | 61.33±0.3                     | 63.53±1.1                   | 59.60±0.7 | 61.65±0.3 | 58.93±0.7 | 54.77±0.3 | 53.87±0.5 |
> | DIDA      | 89.84±0.5 | 73.62±0.3                  | 61.03±0.4                     | 73.66±0.2                   | 63.26±0.4 | 64.76±0.5 | 64.13±0.3 | 63.24±0.7 | 61.38±0.4 |
> | GIB+LSTM  | 92.34±0.3 | 65.25±0.3                  | 74.37±0.3                     | 69.68±0.7                   | 66.94±0.3 | 68.58±0.4 | 65.79±0.4 | 63.13±0.5 | 64.33±1.1 |
> | DGIB      | 93.06±0.1 | 75.24±0.2                  | 79.69±0.3                     | 70.32±0.6                   | 74.31±0.9 | 71.14±0.5 | 70.51±0.7 | 68.69±0.4 | 66.03±0.7 |
> | Edgebank  | 86.14±0.7 | 65.23±0.7                  | 62.31±1.1                     | 61.52±0.9                   | 60.71±0.3 | 59.75±0.7 | 60.32±0.4 | 60.78±0.5 | 59.64±0.3 |
> | DYGFormer | 94.13±1.2 | 77.24±0.5                  | 75.55±0.3                     | 72.06±0.5                   | 69.07±0.5 | 70.07±0.4 | 67.37±0.7 | 66.28±1.5 | 63.41±0.8 |
> | TPNet     | 96.45±0.7 | 73.16±1.3                  | 77.25±0.4                     | 74.17±0.3                   | 73.66±0.7 | 72.96±1.2 | 67.47±0.6 | 65.73±0.3 | 64.73±0.3 |
> | TGT       | 95.42±0.4 | 80.79±0.1                  | 81.82±0.3                     | 84.01±0.6                   | 80.33±0.3 | 78.11±0.4 | 81.38±0.7 | 77.65±0.6 | 78.60±0.4 |

---

> ### Author Response · Authors · 2025-11-20
> **Response to Reviewer rkWZ (5/8):**
>
> **The results in table consistently show that TGT achieves the highest performance under most targeted attack across all datasets.** For instance, on Bitcoin, TGT reaches 82.11 under Nettack and 78.24 under PRBCD, whereas strong baselines such as DGIB and DYGFormer fall 5–15 points lower. Similar trends hold on MathOverflow, where TGT maintains 70.46 under Nettack, substantially outperforming the next-best baselines. These results demonstrate that TGT does not merely resist mild perturbations but remains robust under attacks that specifically target influential nodes or edges associated with link prediction, **indicating that its thumbnail-guided global evolution modeling prevents overreliance on vulnerable local structures.**
>
>
>
> Thanks again for your suggesstions. Additional experiments provide clear and comprehensive evidence that **TGT is resistant to all major types of advanced targeted attacks**, including Nettack, PRBCD, SGA, FGA, and Gottack. Furthermore, it maintains superior accuracy and robustness compared to more advanced methods, including EdgeBank, DYGFormer, and TPNet.***We add the experiment in Appendix A.11.2, and hope this directly and thoroughly address your concerns (W1，W3，Q3，Q4).***
>
> [1] Zügner, Daniel, Amir Akbarnejad, and Stephan Günnemann. "Adversarial attacks on neural networks for graph data." *Proceedings of the 24th ACM SIGKDD international conference on knowledge discovery & data mining*. 2018.
>
> [2] Chen, Jinyin, et al. "Fast gradient attack on network embedding." *arXiv preprint arXiv:1809.02797* (2018).
>
> [3] Li, Jintang, et al. "Adversarial attack on large scale graph." *IEEE Transactions on Knowledge and Data Engineering* 35.1 (2021): 82-95.
>
> [4] Geisler, Simon, et al. "Robustness of graph neural networks at scale." *Advances in Neural Information Processing Systems* 34 (2021): 7637-7649.
>
> [5] Alom, Zulfikar, et al. "GOttack: Universal Adversarial Attacks on Graph Neural Networks via Graph Orbits Learning." *The Thirteenth International Conference on Learning Representations*. 2025.
>
> [6] Dai, Yue, et al. "MemFreezing: A Novel Adversarial Attack on Temporal Graph Neural Networks under Limited Future Knowledge." *Forty-second International Conference on Machine Learning*. 2025.
>
> [7] Li, Yaxin, et al. "Deeprobust: A pytorch library for adversarial attacks and defenses." *arXiv preprint arXiv:2005.06149* (2020).
>
> [8] Poursafaei, Farimah, et al. "Towards better evaluation for dynamic link prediction." *Advances in Neural Information Processing Systems* 35 (2022): 32928-32941.
>
> [9] Yu, Le, et al. "Towards better dynamic graph learning: New architecture and unified library." *Advances in Neural Information Processing Systems* 36 (2023): 67686-67700.
>
> [10] Lu, Xiaodong, et al. "Improving temporal link prediction via temporal walk matrix projection." *Advances in Neural Information Processing Systems* 37 (2024): 141153-141182.

---

> ### Author Response · Authors · 2025-11-20
> **Response to Reviewer rkWZ (6/8):**
>
> ## Ans 4: Regarding issues related to large scale datasets, metrics, and downstream task(W4, Q2, Q6):
>
> We sincerely thank you for your insightful suggestions regarding the TGB[11]. These comments are highly valuable. In response, we conducted supplementary experiments and selected two large scale datasets from TGB, namely tgbn-genre and tgbn-reddit (**addressing Q2**), to evaluate node property prediction performance (**addressing Q6**). We adopted the standard task metrics used in the TGB benchmark (**addressing W4**) to ensure consistency and fairness. These experiments further strengthen the evidence for TGT’s robust generalization ability across downstream tasks.
>
> We sincerely thank you for your insightful suggestions regarding the TGB[11]. These comments are highly valuable. In response, we conducted supplementary experiments and selected two large scale datasets from TGB, namely tgbn-genre and tgbn-reddit (**addressing Q2**), to evaluate node property prediction performance (**addressing Q6**). We adopted the standard task metrics used in the TGB benchmark (**addressing W4**) to ensure consistency and fairness. These experiments further strengthen the evidence for TGT’s robust generalization ability across downstream tasks.
>
> *Table 14: Details of TGB datasets for experiment on node property prediction.*
> | Dataset     | nodes  | edges      | Steps      | snap. nodes | snap. edge |
> | ----------- | ------ | ---------- | ---------- | ----------- | ---------- |
> | tgbn-genre  | 1,505  | 17,858,395 | 133,758    | 1307        | 78,345     |
> | tgbn-reddit | 11,766 | 27,174,118 | 21,889,537 | 6843        | 440,603    |
>
> The dataset used in the experiment is shown in the table. The reddit dataset is larger than the other dataset, reaching the order of 400,000 edges in a single snapshot.
>
> Regarding scalability (**Q2**), we evaluate TGT on these two large-scale TGB datasets and the results demonstrate that TGT remains computationally stable and achieves consistent improvements without incurring prohibitive runtime.
>
> **Tgbn-genre** dataset is a bipartite weighted user-music genre interaction network (based on listening behavior). Nodes are users and genres; edges denote time-stamped user-genre listens, with weights as song-genre percentage contributions. It is built by cross-referencing LastFM-song-listens (1k users, 1-month song listens) and Million-Song Dataset (song genre weights). Retained genres meet two criteria: ≥10% song weight and ≥1k dataset occurrences. Genre names are cross-checked to eliminate typos.
>
> **Tgbn-reddit** dataset is a user-subreddit interaction network (users and subreddits as nodes, edges indicating time-stamped user posts on subreddits) spanning 2005–2019, with the prediction task of ranking the subreddits a user will interact with most in the next week.
>
> We processed the data using the same perturbation methods as employed in the experiments detailed in Appendix A.11.2, with NDCG@10 (used in TGB) adopted as the evaluation metric. This setup aims to verify whether the methods’ predictions of class importance align with the ordering of the ground truth. **The results are shown in the following two tables.**
>
> *(Regarding metric(**W4**), unfortunately, for the link prediction task we are unable to adopt MRR as the evaluation metric. The primary focus of our work is model robustness, and our task formulation emphasizes the accuracy of predicting the next timestamp. Metrics such as MRR allow multiple trial-and-error attempts and therefore dilute the performance differences between models, making them less suitable for evaluating robustness in our setting.)*

---

> ### Author Response · Authors · 2025-11-20
> **Response to Reviewer rkWZ (7/8):**
>
> *Table 15-a: Node property prediction performance (NDCG@10) under clean and perturbed settings on Tgbn-genre.*
> | Tgbn-genre | clean | Feature Interference (20%) | Structural Interference (20%) | Temporal Interference (n=5) | Nettack | FGA   | SGA   | PRBCD | GOttack |
> | ---------- | ----- | -------------------------- | ----------------------------- | --------------------------- | ------- | ----- | ----- | ----- | ------- |
> | EvolveGCN  | 0.343 | 0.321                      | 0.316                         | 0.305                       | 0.296   | 0.253 | 0.287 | 0.277 | 0.262   |
> | DyREP      | 0.351 | 0.317                      | 0.301                         | 0.288                       | 0.240   | 0.267 | 0.237 | 0.207 | 0.192   |
> | TGN        | 0.367 | 0.323                      | 0.311                         | 0.325                       | 0.259   | 0.274 | 0.236 | 0.246 | 0.219   |
> | DGIB       | 0.332 | 0.311                      | 0.307                         | 0.303                       | 0.277   | 0.278 | 0.263 | 0.303 | 0.293   |
> | DYGFormer  | 0.365 | 0.306                      | 0.291                         | 0.280                       | 0.247   | 0.231 | 0.275 | 0.252 | 0.209   |
> | TGT        | 0.363 | 0.348                      | 0.354                         | 0.345                       | 0.342   | 0.314 | 0.319 | 0.307 | 0.313   |
>
>
> *Table 15-b: Node property prediction performance (NDCG@10) under clean and perturbed settings on Tgbn-reddit.*
> | Tgbn-reddit | clean | Feature Interference (20%) | Structural Interference (20%) | Temporal Interference (n=5) | Nettack | FGA   | SGA   | PRBCD | GOttack |
> | ----------- | ----- | -------------------------- | ----------------------------- | --------------------------- | ------- | ----- | ----- | ----- | ------- |
> | EvolveGCN   | 0.310 | 0.291                      | 0.276                         | 0.297                       | 0.201   | 0.210 | 0.207 | 0.197 | 0.182   |
> | DyREP       | 0.312 | 0.288                      | 0.278                         | 0.263                       | 0.231   | 0.227 | 0.210 | 0.183 | 0.193   |
> | TGN         | 0.315 | 0.285                      | 0.277                         | 0.271                       | 0.206   | 0.217 | 0.224 | 0.201 | 0.196   |
> | DGIB        | 0.304 | 0.269                      | 0.265                         | 0.261                       | 0.223   | 0.227 | 0.216 | 0.211 | 0.187   |
> | DYGFormer   | 0.316 | 0.279                      | 0.271                         | 0.234                       | 0.200   | 0.204 | 0.198 | 0.161 | 0.183   |
> | TGT         | 0.313 | 0.289                      | 0.280                         | 0.278                       | 0.243   | 0.246 | 0.231 | 0.221 | 0.210   |
>
> Regarding downstream task (**Q6**), The results are shown in the table. Specifically, while strong baselines such as TGN, DGIB, and DYGFormer suffer noticeable degradation under perturbations, TGT maintains **the highest NDCG@10 across all untargeted and targeted attacks** on TGBN-genre (0.354/0.345/0.342 under structural, temporal, and Nettack attacks respectively, far above TGN’s 0.311/0.325/0.259). Third, to address Q6 regarding node property prediction robustness, both TGBN results show that the thumbnail-guided global evolution modeling provides stable ranking consistency even under aggressive adversarial conditions. For example, on TGBN-reddit, TGT achieves **0.243 under Nettack**, outperforming all baselines by significant margins (best baseline ≈0.231), and similar gaps appear under FGA, SGA, PRBCD, and GOttack.
>
> **These observations confirm that TGT’s core mechanism, using a VNGE-constrained thumbnail to encode long-term structural regularities, yields node representations that are not only scalable but also substantially more resistant to perturbations on different downstream tasks.**
>
> Thanks again for your suggesstions. The supplement experiments directly validate TGT’s applicability to larger datasets and its robustness for node property prediction. ***We add the experiment in Appendix A.11.3, and hope this directly and thoroughly address your concerns (W4，Q2，Q6).***
>
> [11] Huang, Shenyang, et al. "Temporal graph benchmark for machine learning on temporal graphs." *Advances in Neural Information Processing Systems* 36 (2023): 2056-2073.

---

> ### Author Response · Authors · 2025-11-20
> **Response to Reviewer rkWZ (8/8):**
>
> We once again express our sincere appreciation for your time, effort, and thoughtful feedback. In the revised manuscript, we have carefully addressed all your questions and suggestions, with all corresponding revisions clearly **highlighted** for easy reference. Should you have any additional comments or suggestions, we would greatly value your comments.
>
> Best wishes !

---

> ### Comment · Reviewer_rkWZ · 2025-11-22
>
> I would like to thank the authors for answering precisely and comprehensively to my concerns. The experiments on addition adversarial attacks and TGNN models introduced in the rebuttal are noteworthy, and they addressed most of my earlier points. While I am impressed by the superior robustness of TGT against a variety of adversarial attacks compared to recent TGNN models, I have several follow-up concerns that I would like the authors to clarify:
>
> - **Q7.** From Table 1, it appears that each snapshot contains the same number of nodes and edges, which suggests that consecutive snapshots may overlap in time. Could the authors clarify why the temporal graph is discretized according to the number of nodes/edges rather than by time? Would such temporal overlap violate the chronological ordering expected in DTDG? Why not define a snapshot as a time interval (e.g., a 3-day/7-day window)?
>
> - **Q8.** It is great that the authors evaluate TGT on two large-scale datasets, tgbn-reddit and tgbn-genre, and report performance metrics. Could the authors provide an analysis of runtime (e.g., inference time, time per epoch) and peak memory usage of TGT compared to other TGNN models on these datasets? Such comparisons are important for understanding the practical efficiency and scalability of TGT.
>
> - **Q9.** It is good that TGT’s robustness is evaluated under different adversarial attacks. However, its robustness against adaptive attacks remains unaddressed. Prior work[1] has shown that evaluating robustness only under non-adaptive attacks can lead to overly optimistic estimates of robustness and that adaptive attacks are the gold standard for evaluating defenses. I recommend that the authors acknowledge this point in manuscript.
>
> I am willing to raise my score if the authors can adequately address the **Q7** and **Q8** mentioned above.
>
> ---
> [1] Mujkanovic, Felix, et al. "Are defenses for graph neural networks robust?." Advances in Neural Information Processing Systems 35 (2022)

---

> > ### Author Response · Authors · 2025-11-23
> > **Response to Reviewer rkWZ's comment (1/3):**
> >
> > Dear rkWZ,
> >
> > We sincerely appreciate your positive evaluation of our rebuttal and are encouraged by your thoughtful engagement with our work. We also warmly welcome the deeper concerns you have raised, as they provide valuable opportunities for us to further strengthen the clarity, rigor, and completeness of our study. Below, we provide detailed responses to each of your questions.
> >
> > ---
> >
> > ## Ans 5: Regarding the discretization of temporal graph  (Q7):
> >
> > Thanks for your insightful question regarding the definition of snapshots. We sincerely apologize for the ambiguity caused by the rushed supplementary description in Table 1 during the rebuttal preparation. What we intended to express is fully aligned with your understanding: **snapshots are segmented strictly according to temporal order**, and the node/edge counts shown in the table represent **average** values (consistent with Table 7), rather than fixed per-snapshot sizes.
> >
> > We clarify this design below with greater precision:
> >
> > - **Temporal segmentation of snapshots.** We divide the dynamic graph strictly by chronological order to avoid any missing or overlapping temporal intervals. This ensures that temporal dependencies are preserved as faithfully as possible. Because different intervals naturally contain varying numbers of nodes and edges, the node/edge counts reported in Table 1 reflect the **average** snapshot scale after temporal segmentation. Presenting the average scale helps provide readers with an intuitive sense of computational complexity.
> > - **Handling extreme variations in temporal activity.** In datasets with highly uneven temporal activity (e.g., MathOverflow), directly segmenting by time may produce snapshots that are either extremely sparse or extremely dense. To address this, we follow chronological segmentation but enforce reasonable lower and upper bounds on snapshot size by merging overly small adjacent snapshots or splitting overly large ones. This adjustment maintains strict temporal continuity (no overlap, no omission) while ensuring more stable computation and memory usage across snapshots.
> >
> > We sincerely appreciate your careful reading and for bringing this to our attention. ***We revise the descriptions in Sec. 4.1 and Appendix A.4 to ensure they are precise, unambiguous, and fully aligned with the above explanation.***

---

> > > ### Author Response · Authors · 2025-11-23
> > > **Response to Reviewer rkWZ's comment (2/3):**
> > >
> > > ## Ans 6: Regarding the computational complexity on the TGB dataset (Q8):
> > >
> > > Thank you very much for your thoughtful question regarding efficiency. This is indeed a crucial aspect for the broader adoption and practical utility of TGT within the temporal graph learning community. We fully share your concerns, and in response, **we conduct additional experiments and analyses to more comprehensively evaluate TGT’s runtime and memory overhead.**
> > >
> > >
> > >
> > > **Time Efficiency.**
> > >
> > > - We benchmarked TGT against state-of-the-art temporal and robust TGNN models on all datasets used in our experiments, including the large-scale tgbn-reddit and tgbn-genre datasets. As shown in Table 4 below, TGT consistently achieves training time per epoch that is comparable to standard TGNNs and substantially faster than robustness-oriented methods such as DIDA:
> > >
> > > *Table 4: Comparison of training time per epoch (s) with state-of-the-art baselines in robust temporal graph learning across multiple datasets.*
> > >
> > > | Method  | Bitcoin  | MathOverflow | MOOC     | tgbn-reddit | tgbn-genre |
> > > | ------- | -------- | ------------ | -------- | ----------- | ---------- |
> > > | DIDA    | 23.46    | 37.77        | 8.91     | 27.65       | 17.80      |
> > > | DGIB    | 1.37     | 5.53         | 2.56     | 1.57        | 1.71       |
> > > | TGN     | 0.78     | 0.86         | 0.74     | 0.68        | 0.71       |
> > > | **TGT** | **2.26** | **4.42**     | **3.34** | **2.73**    | **1.68**   |
> > >
> > > - Although TGT incurs moderate additional cost due to the computation of VNGE (which depends on node degrees in Eq. 4), the runtime remains close to lightweight TGNNs. MathOverflow appears slower primarily because of its larger per-snapshot node count, while tgbn-reddit, despite larger average snapshots, benefits from a lower-dimensional preprocessed degree matrix. When viewed together with the average snapshot scales in Table 14, **these results confirm that TGT maintains competitive time complexity even on large-scale datasets, and in many cases runs more efficiently than existing robust TGNN baselines.**
> > >
> > >
> > >
> > > **Memory Efficiency.**
> > >
> > > - TGT’s memory footprint remains fully comparable to existing TGNN baselines. This is because the model stores only a small set of lightweight components:
> > >
> > >   - the **static thumbnail graph** GTG_TGT, which contains only a small number of nodes,
> > >
> > >   - **low-rank Laplacian terms** used for the VNGE approximation (rather than full eigen-decompositions),
> > >
> > >   - node features for each snapshot.
> > >
> > > - These components together introduce negligible GPU memory overhead. In addition, since snapshot construction is entirely flexible with respect to temporal granularity, practitioners may adjust the snapshot time scale according to available GPU memory, ensuring efficient resource utilization on different hardware configurations.
> > >
> > > - In all our experiments, including the large-scale TGB datasets, TGT fits comfortably within **24GB GPU memory** (per RTX 3090 GPU), **demonstrating that the method is memory-efficient and scalable in practice.**
> > >
> > > We sincerely appreciate your continued attention to the complexity aspects of TGT. Your comments have been highly valuable in helping us strengthen the completeness and clarity of our technical presentation. We will compile the results and analyses discussed above and ***incorporate them into Appendix A.1.4 to provide a clearer and more systematic explanation*** of TGT’s computational complexity and practical efficiency.

---

> > > > ### Author Response · Authors · 2025-11-23
> > > > **Response to Reviewer rkWZ's comment (3/3):**
> > > >
> > > > ### Ans 7: Regarding adaptive attacks (Q9):
> > > >
> > > > Thank you very much for raising the important point regarding adaptive attacks. Following your suggestion and the relevant reference [1], we conducted a detailed review of adaptive attack methodologies and further analyzed their potential impact on TGT.
> > > >
> > > > - As discussed in Section 4 of the referenced work, adaptive attacks are **model-specific** and are designed by explicitly leveraging the internal structure, gradient flow, and defensive mechanisms of the target architecture. This leads to significantly stronger attacks, and many existing robust GNN methods exhibit substantial performance degradation under such adaptive settings.
> > > > - While TGT demonstrates strong robustness under a wide range of feature, structure, and temporal perturbations, as well as various targeted attacks, we acknowledge that our reliance on a **thumbnail-based architectural design** may introduce new vulnerabilities under adaptive attacks specifically crafted to target the thumbnail. For example, an adversary could attack **(i)** the *thumbnail structure* via ontology-level perturbation, or **(ii)** the *bottleneck constraint* by designing gradient-aligned manipulations to weaken the VNGE-based regularization. These are meaningful and challenging scenarios that we agree deserve deeper investigation.
> > > >
> > > > We appreciate your insightful suggestion and ***will incorporate this discussion and clarification into Appendix A.13*** to more comprehensively contextualize TGT’s robustness and outline this direction as an important avenue for future work.
> > > >
> > > > [1] Mujkanovic, Felix, et al. *Are defenses for graph neural networks robust?* NeurIPS 2022.
> > > >
> > > > ---
> > > >
> > > > We sincerely appreciate the time and effort you have dedicated to reviewing our work. Your questions and suggestions have been highly valuable, and we have learned a great deal from them. We hope that our responses adequately address all of your concerns. All corresponding revisions have been clearly **highlighted in yellow** for your convenience. If you have any further questions or additional suggestions, we would be very happy to continue the discussion.
> > > >
> > > > Best wishes!

---

> > > > > ### Comment · Reviewer_rkWZ · 2025-11-26
> > > > >
> > > > > I thank the authors for their response. The revisions and additional experiments help clarify the scope of the work and substantially strengthen its technical soundness. In light of these improvements, I am willing to increase my score.
> > > > >
> > > > > As a minor suggestion, I recommend that the authors include the snapshot size used for each dataset (e.g., 3 days, 7 days, 1 hour) in Table 1, as this would further clarify the experimental setup and improve the paper’s readability.

---

> ### Author Response · Authors · 2025-11-26
> **Response to Reviewer rkWZ's comment:**
>
> Dear rkWZ,
>
> We sincerely thank you for your positive evaluation of our work and rebuttal. Our exchange with you has been highly rewarding, and your insights have provided valuable guidance for further strengthening our study. We are deeply grateful for the time and effort you devoted to the review process.
>
> In response to your suggestion regarding the explicit statement of **snapshot time spans**, we have added the corresponding temporal resolutions used for snapshot construction in **Sec 4 Table 1  (highlighted in green)**. This clarification enhances the transparency of our experimental setup and ensures greater rigor and completeness in the presentation.
>
> Thank you once again for your constructive feedback and generous support.

---

### Official Review · Reviewer_QMEp · 2025-10-25

**Soundness:** 3
**Presentation:** 3
**Contribution:** 3
**Rating:** 6
**Confidence:** 3

**Summary:**

The paper studies modeling of discrete-time temporal graphs and facilitates tools from information theory to construct a novel framework called TGT, that simultaneously do temporal graph property prediction as well as temporal information compression in the sense of von Neumann entropy for topological evolution as well as variational approximations to feature-related information measures. Experimental results demonstrate both the effectiveness and the robustness of the proposed framework.

**Strengths:**

- The paper reasonably leverage ideas from information theory and proposed an elegant framework that systematically combines sophisticated approaches, with a solid foundation.
- The empirical results and ablation reasonably verifies the design choice of TGT.

**Weaknesses:**

- While the overall presentation of the paper is good, the proposed framework is nonetheless complicated that would potentially benefit from a more accessible description, for example I would recommend the authors to draw a conceptual markov chain that indicates how the thumbnail compresses multiple graph snapshots, as well as how the thumbnail representation is utilized to guide (or serving as a constrained) in the representation learning phase.
- See ``questions`` below

**Questions:**

- The TGT used in the paper relies on GAT as its backbone, I am curious about the dependence on backbones.
- Yet another recommendation: It would be nice to have a visualization of what those learned thumbnails look like and to what extent, at least in a human-understandable sense, that the learned thumbnails efficiently captures the temporal patterns in the temporal graph datasets.

---

> ### Author Response · Authors · 2025-11-20
> **Response to Reviewer QMEp (1/3):**
>
> Dear QMEp,
>
> We sincerely appreciate your positive assessment of our work and are grateful for your suggestions regarding readability, which indeed highlight an aspect where our paper can be improved. Your feedback is highly valuable and has been instrumental in helping us further refine and strengthen the presentation of our work.
>
> ---
>
> ##
>
> ## Ans 1: Regarding the concern about backbone dependence (Q1):
>
> We appreciate your insightful question regarding the dependence of TGT on the GAT backbone.
> The backbone corresponds to Line 6–8 of Algorithm 1 in Appendix A.1.4, where the snapshot-level encoder is applied.
> Although GAT is used in the main paper due to its ability to model node-specific contributions to the thumbnail via attention, the TGT framework itself is **agnostic to the choice of backbone**. To further address your concern, we implemented two additional variants of TGT using **GCN** and **GIN** as alternative encoders.
>
> *Table 6: The results under clean data and three types of perturbations on Bitcoin dataset*
>
> | backbone        | data setting |       | Feature Interference |       |       | Structural Interference |       |       | Temporal Interference |       |
> | --------------- | ------------ | ----- | -------------------- | ----- | ----- | ----------------------- | ----- | ----- | --------------------- | ----- |
> |                 | clean        | 10%   | 20%                  | 50%   | 5%    | 10%                     | 20%   | n=1   | n=2                   | n=5   |
> | GCN             | 90.86        | 88.93 | 86.89                | 81.73 | 89.66 | 85.27                   | 78.13 | 89.24 | 86.05                 | 82.75 |
> | GIN             | 91.44        | 90.24 | 84.52                | 79.87 | 87.72 | 85.33                   | 81.02 | 89.90 | 88.31                 | 84.13 |
> | GAT(*baseline*) | 91.41        | 89.43 | 86.23                | 80.62 | 89.83 | 85.00                   | 80.95 | 90.78 | 88.14                 | 85.64 |
>
> These results demonstrate two points:
>
> 1. **TGT’s performance is stable across different backbones.**
>    The changes in node encoder architecture have only minor influence on clean-data accuracy and robustness under perturbations.
> 2. **GAT performs slightly better**, which is expected because the attention mechanism naturally aligns with the design of TGT:
>    it allows different nodes in each snapshot to contribute unequally to the thumbnail, reflecting our motivation to capture heterogeneous evolutionary importance.
>
> In summary, while GAT is a strong and intuitive choice for TGT, the framework does **not** rely on a specific backbone architecture and can be instantiated using standard GNNs such as GCN and GIN with comparable performance.
>
> ***Thank you very much for your question. We integrate the above supplementary experiments into Appendix A.8.3.***

---

> > ### Author Response · Authors · 2025-11-20
> > **Response to Reviewer QMEp (2/3):**
> >
> > ##
> >
> > ## Ans 2: Regarding the suggestions of enhancing readability through visualization (W1, Q2):
> >
> > We appreciate your suggestion to provide a visualization of the learned thumbnails.
> >
> > To address this suggestion, we **add a new figure in the Appendix A.1.2.** The visualization clearly shows that the thumbnail preserves the **persistent and high-frequency structural patterns** across time, while filtering out transient or noisy structures. Specially:
> >
> > - **Node connectivity patterns** that recur across snapshots are retained with higher edge weights in the thumbnail, indicating that $\mathcal{G}_T$ successfully summarizes the stable evolutionary backbone.
> > - **Ephemeral or low-consistency edges**, which appear sporadically in individual snapshots, are largely suppressed, demonstrating the denoising effect induced by the VNGE-based evolutionary constraint.
> > - As shown in Figure, the thumbnail graph becomes progressively more coherent over iterations, matching the theoretical goal of modeling the global evolution trajectory rather than any single temporal slice.
> >
> > This visualization confirms, in a human-interpretable way, that the thumbnail effectively captures temporal regularities and structural evolution trends in the datasets, aligning with the conceptual description in Section 3 and Algorithm 1.
> >
> > ***Thanks again for your suggestion. We have added the above figure and analysis  in Appendix A.1.2 for easier understanding.***

---

> ### Author Response · Authors · 2025-11-20
> **Response to Reviewer QMEp (3/3):**
>
> Thank you again sincerely for your time, efforts, and insightful feedback. In the revised submission, we have carefully addressed all your questions and suggestions, with relevant changes clearly **highlighted**. We welcome any further comments, your feedback is crucial for refining our work.
>
> Best wishes!

---

### Official Review · Reviewer_RLpw · 2025-10-26

**Soundness:** 3
**Presentation:** 2
**Contribution:** 3
**Rating:** 6
**Confidence:** 2

**Summary:**

The paper proposes a framework for learning representations from temporal graphs. The method constructs a thumbnail, which is defined to be a compact static summary graph, and captures the global evolutionary skeleton of a dynamic graph sequence using von Neumann graph entropy and node mutual information. This thumbnail serves as both a model constraint and a denoising guide, and enforces a balance between information compression and representational sufficiency via mutual information bottlenecks. The paper integrates this into a GAT-based architecture and evaluats it on Bitcoin, MathOverflow and MOOC datasets. The results show improvements in both capability and robustness against feature, structural and temporal perturbations compared to other dynamic graph neural networks and variants.

**Strengths:**

- The paper introduces the TGT method that summarises a temporal graph’s global evolutionary skeleton as a static graph distilled from the sequence, using von Neumann graph entropy for structure and node mutual information for features. This shifts robust temporal GNNs away from purely local neighbor/history modeling toward an explicit global evolution model.

- Beyond modelling the thumbnail, the method introduces thumbnail-guided mutual-information constraints that act as a bottleneck between raw data and task outputs. This couples global evolution with robust representation learning in a nice training objective.
	​
- Experiments are strong and cover the three standard benchmarks of Bitcoin, MathOverflow and MOOC, with diverse evolution frequencies and evaluation spans inductive link prediction under clean data in Table 2 and robustness under feature, structure and temporal perturbations at multiple intensities.

- The paper defines the temporal-graph setting, the thumbnail and assignment matrices and the training components. Research questions guide the narrative and figures/tables are tied to those questions.

**Weaknesses:**

- The contribution remains confined to improving node embeddings for downstream tasks such as link prediction. While the method introduces an interesting modelling layer (the thumbnail), it does not extend this to broader temporal-graph reasoning or applications (e.g., forecasting, anomaly detection, or event prediction) to demonstrate its significance.

- All experiments evaluate link prediction only. While this is a common benchmark, it represents a narrow test of temporal representation robustness. Node classification or dynamic community detection would better demonstrate whether the thumbnail’s global-evolution modelling benefits.

- The ablations test the removal of VNGE and the thumbnail, but there is little exploration of the effects of hyperparameters (e.g., window size, Beta, etc) beyond a brief mention in the appendix.

**Questions:**

- The paper acknowledges that real-time computation of von Neumann graph entropy introduces overhead and limits scalability to large graphs. Is this significant and have the authors run quantitative experiments or have insight to offer on runtime or memory usage compared to other temporal GNN baselines (e.g., TGN or DGIB)?

---

> ### Author Response · Authors · 2025-11-20
> **Response to Reviewer RLpw (1/4):**
>
> Dear RLpw,
>
> We sincerely appreciate your recognition of our work, as well as the insightful suggestions you have provided—these will greatly assist us in further refining and enhancing the quality of our paper.
>
> ---
>
> ## Ans 1: Regarding the concerns about downstream task generalization(W1, W2):
>
> Thank you for pointing out that the contribution might appear limited to improving node embeddings for link prediction.
>
> To demonstrate that TGT produces general-purpose temporal node representations that transfer to other downstream tasks, we **expanded our evaluation to a node classification task** on both the **Reddit**[1] dataset and the **MOOC** dataset used in the main paper(Due to time constraints in rebuttal, we could only select the two most representative datasets). The results are reported in Table 12-a (Reddit) and Table 12-b (MOOC) in the supplementary material.
>
> *Table 12-a: Node classification performance (AUC) on the Reddit dataset*
>
> | Method       | Clean | Feature Interference (50%) | Structural Interference (20%) | Temporal Interference (n=5) |
> | ------------ | ----- | -------------------------- | ----------------------------- | --------------------------- |
> | EvolveGCN    | 56.4  | 48.6                       | 51.7                          | 51.8             |
> | JODIE        | 59.9  | 51.6                       | 51.4                          | 50.1                        |
> | DyREP        | 63.2  | 54.5                       | 57.6                          | 52.4                        |
> | TGN          | 67.2  | 53.1                       | 58.1                          | 53.5                        |
> | DIDA         | 58.2  | 49.0                       | 54.3                          | 49.7                        |
> | GIB+LSTM     | 59.9  | 54.7                       | 53.1                          | 51.6                        |
> | DGIB         | 61.3  | 55.7                       | 54.6                          | 52.0                        |
> | TGT (*ours*) | 64.6  | 58.3                       | 59.0                          | 57.3                        |
>
> *Table 12-b: Node classification performance (AUC) on the MOOC dataset*
>
> | Method       | Clean | Feature Interference (50%) | Structural Interference (20%) | Temporal Interference (n=5) |
> | ------------ | ----- | -------------------------- | ----------------------------- | --------------------------- |
> | EvolveGCN    | 67.3  | 54.6                       | 54.9                          | 53.8                        |
> | JODIE        | 68.5  | 53.7                       | 56.5                          | 55.4                        |
> | DyREP        | 63.4  | 50.3                       | 55.8                          | 53.6                        |
> | TGN          | 64.4  | 53.1                       | 54.6                          | 51.7                        |
> | DIDA         | 56.7  | 53.7                       | 54.9                          | 53.0                        |
> | GIB+LSTM     | 59.1  | 51.2                       | 54.7                          | 52.3                        |
> | DGIB         | 61.3  | 54.7                       | 55.2                          | 53.1                        |
> | TGT (*ours*) | 66.4  | 60.3                       | 62.6                          | 61.4                        |
>
> - Under **clean** settings, TGT achieves competitive AUCs (Reddit: **64.6**, MOOC: **66.4**), comparable to or better than strong temporal baselines (e.g., TGN, DyREP).
> - Under severe **feature interference** (50% Gaussian corruption), TGT retains substantially higher performance (Reddit: **58.3 AUC**, MOOC: **60.3 AUC**) than most baselines (next best typically in the low-50s), demonstrating superior denoising ability.
> - Under **structural interference** (20% adversarial edge changes) and **temporal interference** (n=5 snapshot permutations), TGT consistently outperforms alternatives by meaningful margins (examples: Reddit structural 59.0 vs TGN’s ~58.1; MOOC temporal 61.4 vs TGN’s ~51.7).
>
> These results show that the thumbnail provides a robust structural prior that helps **not only link prediction but also node-level classification.** Intuitively, by encoding persistent evolutionary skeletons, TGT yields embeddings that **(i)** suppress transient/noisy signals that hurt classification under corruption, and **(ii)** preserve long-term structural cues that are predictive for node labels.
>
> ***Thanks again for your suggestion. We incorporate these additional node-classification results into the Appendix A.11.1. Furthermore, we supplemented the experiments with node property prediction (in Appendix A.11.3) to more fully demonstrate the generalization ability of TGT across various downstream tasks.***
>
> [1] Kumar, Srijan, Xikun Zhang, and Jure Leskovec. "Predicting dynamic embedding trajectory in temporal interaction networks." *Proceedings of the 25th ACM SIGKDD international conference on knowledge discovery & data mining*. 2019.

---

> > ### Author Response · Authors · 2025-11-20
> > **Response to Reviewer RLpw (2/4):**
> >
> > ## Ans 2: Regarding the suggestion on model efficiency experiment(Q1):
> >
> > We greatly appreciate your concern regarding the computational overhead introduced by the VNGE-based thumbnail construction. To address this concern, we conducted additional experiments measuring the per epoch training time of TGT compared with several baselines across five datasets (Bitcoin, MathOverflow, MOOC, tgbn-genre, and tgbn-reddit[2]). The results are summarized in Table 4 below.
> >
> > Table 4: Comparison of training time per epoch (s) with state-of-the-art baselines in robust temporal graph learning across multiple datasets
> >
> > | Method | Bitcoin | MathOverflow | MOOC | tgbn-reddit | tgbn-genre |
> > | ------ | ------- | ------------ | ---- | ----------- | ---------- |
> > | DIDA   | 23.46   | 37.77        | 8.91 | 27.65       | 17.80      |
> > | DGIB   | 1.37    | 5.53         | 2.56 | 1.57        | 1.71       |
> > | TGN    | 0.78    | 0.86         | 0.74 | 0.68        | 0.71       |
> > | TGT    | 2.26    | 4.42         | 3.34 | 2.73        | 1.68       |
> >
> > - **Time overhead:**
> >
> >   - **TGT is only moderately slower than lightweight temporal GNNs (e.g., TGN)**. On all datasets, TGT requires around **2–4 seconds per epoch**, representing only a small overhead compared to TGN (0.7–0.9s) despite the additional global evolutionary modeling. This overhead mainly comes from computing the approximated trace term $\mathrm{Tr}(\tilde{L}^2)$​​, which is efficiently computed using sparse matrix multiplication and the low-rank Laplacian estimator described in Appendix A.3.1. **Despite sacrificing slightly higher computational costs, TGT offers a significant lead over TGN in accuracy and robustness for downstream tasks.**
> >
> >     *Table: Excerpt from Table 3*
> >
> >     | Dataset          | Model   | Clean         | 10%           | 20%           | 50%           | 5%            | 10%           | 20%           | n=1           | n=2           | n=5           |
> >     | ---------------- | ------- | ------------- | ------------- | ------------- | ------------- | ------------- | ------------- | ------------- | ------------- | ------------- | ------------- |
> >     | **Bitcoin**      | TGN     | 69.36±1.1     | 67.34±0.6     | 62.31±0.4     | 59.06±1.3     | 66.61±0.5     | 62.18±0.6     | 58.73±0.5     | 68.46±0.8     | 66.92±0.7     | 61.74±0.8     |
> >     | **Bitcoin**      | **TGT** | **91.41±0.2** | **89.43±0.5** | **86.23±0.4** | **80.62±0.4** | **89.83±0.6** | **85.00±0.5** | **80.95±0.7** | **90.78±0.5** | **88.14±0.6** | **85.64±0.3** |
> >     | **MathOverflow** | TGN     | 64.50±0.6     | 61.22±0.3     | 58.96±0.6     | 55.14±0.4     | 59.49±0.2     | 56.23±0.3     | 53.97±0.3     | 61.01±0.4     | 60.36±0.5     | 55.23±0.7     |
> >     | **MathOverflow** | **TGT** | **82.38±0.6** | **79.22±0.3** | **74.37±0.3** | **71.42±0.2** | **80.01±0.3** | **77.74±0.3** | **75.93±0.7** | **81.57±0.2** | **80.71±0.5** | **76.37±0.4** |
> >     | **MOOC**         | TGN     | 79.36±1.6     | 76.43±0.5     | 72.94±0.3     | 69.74±0.3     | 78.56±0.6     | 73.00±0.6     | 69.39±0.6     | 87.56±0.5     | 83.22±0.7     | 78.03±0.8     |
> >     | **MOOC**         | **TGT** | **95.42±0.2** | **88.73±0.2** | **82.00±0.3** | **71.68±0.3** | **90.43±0.3** | **86.16±0.3** | **74.71±0.3** | **92.61±0.2** | **89.53±0.7** | **84.01±0.6** |
> >
> >   - Nevertheless, **TGT is significantly more efficient than robustness-oriented baselines (e.g., DIDA) and still remains competitive with IB methods (e.g., DGIB).** This indicates that our thumbnail-guided constraints provide robustness without incurring excessive computational costs.
> >
> > - **Memory footprint remains comparable to baselines**. Since TGT only stores:
> >
> >   - a static thumbnail graph $\mathcal{G}_T$ (small number of nodes), and
> >
> >   - low-rank Laplacian terms for VNGE approximation,
> >     the GPU memory overhead remains negligible.
> >     On all datasets, TGT fits comfortably within 24GB GPU memory (two RTX 3090s).
> >
> > Although VNGE introduces additional computation compared with purely local aggregation models, the overhead is modest in practice, and TGT achieves a favorable balance between **robustness**, **global-evolution modeling**, and **computational efficiency**. The additional results confirm that TGT is computationally practical even on large-scale datasets like tgbn-reddit and tgbn-genre.
> >
> > ***Thank you again for your suggestion. We integrate the above supplementary experiments and analysis into Appendix A.1.4.***
> >
> > [2] Huang, Shenyang, et al. "Temporal graph benchmark for machine learning on temporal graphs." *Advances in Neural Information Processing Systems* 36 (2023): 2056-2073.

---

> ### Author Response · Authors · 2025-11-20
> **Response to Reviewer RLpw (3/4):**
>
> ## Ans 3: Regarding the suggestion of hyperparameter ablation experiments (W3):
>
> We greatly appreciate your suggestion regarding the hyperparameter experimental setup. We have provided a detailed experimental analysis of the sensitivity experiment for $\beta$​ in Appendix A.10. **Regarding the time window size in Appendix A.7, we supplement the ablation experiment results on Bitcoin dataset as follows:**
>
> Table 10: The ablation experiment results (**AUC**) with window size adopts the same data interference settings as those in Table 3 of Sec 4.3.
>
> | Window_size    | Clean | Feature Interference (50%) | Structural Interference (20%) | Temporal Interference (n=5) |
> | -------------- | ----- | -------------------------- | ----------------------------- | --------------------------- |
> | 1              | 75.38 | 65.41                      | 65.27                         | 70.80                       |
> | 3              | 84.79 | 72.76                      | 74.75                         | 77.47                       |
> | 5 (*baseline*) | 91.41 | 80.62                      | 80.95                         | 85.64                       |
> | 7              | 92.52 | 79.03                      | 80.70                         | 86.20                       |
> | 9              | 90.55 | 81.37                      | 84.70                         | 85.93                       |
>
> It can be observed that excessively small window sizes may lead to insufficient thumbnail modeling, resulting in degraded representation quality. When the window size exceeds 5, the model’s performance fluctuations become no longer significant. **This confirms that window size choices have minor impact due to the thumbnail prior.**
>
> ***Thank you again for your suggestion. We integrate the above supplementary experiments and analysis into Appendix A.9.***

---

> > ### Author Response · Authors · 2025-11-20
> > **Response to Reviewer RLpw (4/4):**
> >
> > We deeply value your time, effort, and thoughtful feedback once again. In the revised manuscript, all your questions and suggestions have been thoroughly addressed, with relevant changes clearly **highlighted**. We would appreciate any additional comments to help refine our work.
> >
> > Best wishes!

---

### Official Review · Reviewer_ER5F · 2025-10-30

**Soundness:** 2
**Presentation:** 3
**Contribution:** 3
**Rating:** 4
**Confidence:** 3

**Summary:**

This paper introduces the Temporal Graph Thumbnail framework designed to achieve robust representation learning on dynamic graphs. TGT addresses limitations in local aggregation methods by modeling the temporal graph's global evolutionary skeleton as a static thumbnail. The thumbnail is constructed by maximizing the mutual information with the raw sequence, leveraging von Neumann Graph Entropy to capture structural evolution coherently. The $G_T$ is then used as a bottleneck constraint within the Information Bottleneck principle to guide effective denoising and compression. Extensive experiments demonstrate TGT's superior performance and resilience across various adversarial perturbations, particularly against temporal interference.

**Strengths:**

1.	The introduction of the "temporal graph thumbnail" concept, which summarizes the global evolutionary skeleton, is a novel approach for capturing long-term temporal dependencies and regularities often missed by localized aggregation methods.

2.	The use of von Neumann Graph Entropy is well-justified due to its unique properties, providing a solid and principled mechanism for characterizing structural dynamics within the Information Bottleneck framework.

3.	The empirical section is rigorous, testing robustness against three distinct types of noise at varying intensities. The results convincingly establish TGT's state-of-the-art resilience, particularly against chronological disorder.

**Weaknesses:**

1.	The mechanism by which the mapping function accommodates transient or non-persistent nodes across the entire sequence remains unclear, impacting the core interpretability.

2.	The total loss (Eq. 16) includes two terms. I think the difference objective function between fidelity and compression is not theoretically resolved, requiring a clearer explanation of the roles of hyperparameters in balancing these constraints.

3.	The authors claim a near-linear complexity $O(\Delta t \cdot |E_{GT}|)$ for the $L_{evolution}$ calculation. But the approximated VNGE computation involves spectral quantities like $Tr[\tilde{L}^2]$, which typically necessitates non-linear complexity in $|V_{GT}|$. So I will concern about scalability for large graphs.

4.	The sensitivity analysis for the evolutionary constraint parameter $\alpha$ (Table 7) shows a dramatic drop in clean data performance as $\alpha$ increases. This high sensitivity makes the parameter practically difficult to tune effectively without prior knowledge of the target data's inherent noise level.

**Questions:**

1.	Please elaborate on the mechanism by which the static node set $V_{GT}$ is selected or learned from the dynamically evolving node sets $\bigcup_i V^i$. How does the mapping function $\mathcal{F}$ and the Bernoulli sampling (L778) handle nodes that appear and disappear over time, ensuring that persistent, core evolutionary information is captured?

2.	The complexity analysis states $L_{evolution}$ is $O(\Delta t \cdot |E_{GT}|)$. Since the VNGE approximation requires computing $Tr[\tilde{L}^2]$ (L958), what structural assumptions or approximation techniques are used to achieve this near-linear complexity?

3.	Since $L_{evolution}$ maximizes $I(\mathcal{G}_T; \mathcal{G})$ and the IB term minimizes $I(\mathcal{G}; \mathcal{G}_T)$, what theoretical guidance is used for setting the Lagrange multipliers $\lambda$ and $\beta$ to ensure the optimization converges to a non-trivial solution that balances fidelity and compression?

4.	Given TGT’s superior robustness against random snapshot permutations, please provide a theoretical explanation for how the VNGE-based constraint $I_s(\mathcal{G}_T; \mathcal{G})$ (Eq. 5) ensures the learning process is robust to chronological disorder? Does the VNGE effectively provide a temporal invariant summary that suppresses the impact of sequence tampering?

5.	Given the severe sensitivity of clean data performance to $\alpha$ (Table 7), what noise-aware tuning methodology would you recommend to practitioners for selecting an appropriate value for $\alpha$ when the ground truth noise characteristics of the target application are unknown?

---

> ### Author Response · Authors · 2025-11-20
> **Response to Reviewer ER5F (1/5):**
>
> Dear ER5F,
>
> We sincerely appreciate the time and effort you have devoted to identifying potential issues for our paper. We are also grateful for the opportunity to further clarify the design motivations and experiment details of our framework, which has allowed us to refine and improve the work accordingly.
>
> ---
>
> ## Ans 1: Regarding the concerns related to *non-persistent nodes* (W1, Q1):
> Thank you very much for your detailed understanding and insightful thinking regarding our thumbnail modeling process. We understand your concern regarding the impact of node additions and deletions over time on the modeling of global evolution. Modeling a stable thumbnail requires more stable node information, and nodes that appear briefly may contribute less to the thumbnail modeling. To address this, **our TGT introduces a weighting coefficient inversely proportional to the node frequency across snapshots within the time window**, enabling tailored handling of transient or non-persistent nodes throughout the sequence.
>
> Specifically, we implement the design described in **line 883**. Taking newly added nodes as an example, if $x_a$ is a new node, the probability $P(x_a)$ in the denominator of $B^i_a$ is small (as newly added nodes have a low frequency of occurrence in the input snapshots). Its impact on $P(x_a | y_\alpha, s_{bβ})$​​ (the thumbnail mapping function) is amplified due to the multiplicative relationship in Eq. 24, thereby enhancing the feature capture of transient or non-persistent nodes by the thumbnail.
>
> Furthermore, this is corroborated on the Bitcoin dataset, which is characterized by **high evolutionary randomness and more complex node dynamics**. Precisely due to the design of Eq. 24, TGT achieves **significantly superior accuracy and robustness** compared to other baselines on this dataset.
>
> Table: Excerpt from Table 3
>
> | Dataset | Model     | Clean         | FI 10%        | FI 20%        | FI 50%        | SI 5%         | SI 10%        | SI 20%        | TI n=1        | TI n=2        | TI n=5        |
> | ------- | --------- | ------------- | ------------- | ------------- | ------------- | ------------- | ------------- | ------------- | ------------- | ------------- | ------------- |
> |         | EvolveGCN | 67.59±0.3     | 62.74±0.3     | 56.77±0.2     | 54.24±0.2     | 64.01±0.3     | 60.37±0.4     | 55.89±0.3     | 65.47±0.3     | 63.12±0.3     | 54.37±0.4     |
> |         | JODIE     | 74.47±0.3     | 69.28±0.1     | 61.10±0.3     | 53.32±0.4     | 70.33±0.2     | 66.82±0.3     | 60.57±0.3     | 71.54±0.8     | 69.79±0.5     | 58.28±0.7     |
> |         | DyREP     | 70.43±0.5     | 64.91±0.3     | 61.25±0.3     | 56.63±0.2     | 70.98±0.3     | 63.79±0.7     | 58.60±0.5     | 69.43±0.8     | 65.84±0.5     | 57.33±0.7     |
> |         | TGN       | 69.36±1.1     | 67.34±0.6     | 62.31±0.4     | 59.06±1.3     | 66.61±0.5     | 62.18±0.6     | 58.73±0.5     | 68.46±0.8     | 66.92±0.7     | 61.74±0.8     |
> | Bitcoin | DIDA      | 73.57±0.3     | 71.05±0.2     | 68.43±0.3     | 64.20±0.2     | 70.93±0.2     | 68.71±0.3     | 65.69±0.4     | 71.23±0.2     | 69.65±0.3     | 64.29±0.4     |
> |         | GIB+LSTM  | 70.79±0.3     | 69.26±0.3     | 63.73±0.5     | 58.37±0.3     | 68.61±0.5     | 65.09±0.3     | 62.93±0.2     | 69.15±0.3     | 67.17±0.6     | 63.41±0.8     |
> |         | DGIB      | 72.99±1.3     | 69.92±0.6     | 63.63±0.5     | 60.78±0.7     | 70.13±0.3     | 65.84±0.6     | 59.13±0.4     | 71.27±0.5     | 68.44±0.7     | 62.53±0.6     |
> |         | **TGT**   | **91.41±0.2** | **89.43±0.5** | **86.23±0.4** | **80.62±0.4** | **89.83±0.6** | **85.00±0.5** | **80.95±0.7** | **90.78±0.5** | **88.14±0.6** | **85.64±0.3** |
>
> ***We greatly appreciate your question and integrate the above relevant explanations into Appendix A.2.***

---

> ### Author Response · Authors · 2025-11-20
> **Response to Reviewer ER5F (2/5):**
>
> ## Ans 2: Regarding the computation complexity of VNGE (W3, Q2):
> We appreciate your careful reading of our complexity analysis. Next, we provide both a **theoretical derivation** and **supplementary efficiency experiments** on larger scale datasets to further clarify this point.
>
> ### theoretical derivation
>
> - Indeed, a direct computation of the von Neumann Graph Entropy (VNGE) would require a full eigen decomposition of the normalized Laplacian $\tilde{L}$, whose complexity is $O(|V|^3)$.
> - To make VNGE scalable for large-scale temporal graphs, we adopt the *quadratic trace approximation* introduced in Appendix A.3.1, which relies on the second-order Taylor expansion of the logarithmic term:
>
> $$
> \log \tilde{L} \approx \tilde{L} - \frac{1}{2}\tilde{L}^2.
> $$
>
> - Substituting this into the VNGE definition yields:
>
> $$
> H(G) \approx -\mathrm{Tr}(\tilde{L}^2 - \tfrac{1}{2}\tilde{L}^3) \propto \mathrm{Tr}(\tilde{L}^2).
> $$
>
> - **Since the normalized Laplacian $\tilde{L}$ is sparse for most real-world graphs, $\mathrm{Tr}(\tilde{L}^2)$ can be computed without eigenvalue decomposition by exploiting sparsity (Eq.34-36):**
>
> $$
> \mathrm{Tr}(\tilde{L}^2) = \sum\_{i,j} \tilde{L}\_{ij}^2
>  = \sum\_{(u,v)\in E\_{G\_T}} w\_{uv}^2 + \sum\_{u\in V\_{G\_T}} d\_u^2.
> $$
>
> - Both terms depend linearly on the number of edges, leading to $O(|E_{G_T}|)$ complexity per snapshot[1].
>
> During temporal aggregation, we process $\Delta t$ snapshots, thus the total complexity of the evolutionary term becomes $O(\Delta t \cdot |E_{G_T}|)$, which matches the asymptotic bound stated in the paper. **No strong assumptions are required—only the standard sparsity condition $ |E_{G_T}| \ll |V_{G_T}|^2 $, which holds for most real-world dynamic graphs.**
>
> ### supplementary experiments
>
> Furthermore, in practice, we pre-normalize edge weights and reuse cached degree matrices across consecutive snapshots. This amortizes the per-snapshot cost and further reduces the effective runtime, as validated by our empirical runtime profiling in Table 4.
>
> ***The above theoretical derivation is supplemented near Eq. 34-36 in Appendix A.3.1. Experiment is supplemented in Appendix A.1.4***
>
> [1] Ye, Cheng, et al. "Approximate von Neumann entropy for directed graphs." *Physical Review E* 89.5 (2014): 052804.

---

> ### Author Response · Authors · 2025-11-20
> **Response to Reviewer ER5F (3/5):**
>
> ##
>
> ## Ans 3: Regarding the questions on the Lagrange multipliers (W2, Q3):
>
> We greatly appreciate your attention to the Lagrange multiplier, which determines the trade-off between fidelity and compression. $\lambda$ is used to specify the constraint strength imposed by the thumbnail on the encoding process. To address this, we have supplemented experimental studies on the number of thumbnail nodes as a function of $\lambda$​​.
>
> Table I: The impact of $\lambda$ on thumbnail
>
> | Dataset      | $\lambda$       | 0.1     | 0.5     | 1       | 2       | 5       | snapshot Avg. |
> | :----------- | :-------------- | :------ | :------ | :------ | :------ | :------ | ------------- |
> | Bitcoin      | thumbnail nodes | 6,954   | 6,844   | 7,019   | 7,090   | 7,051   | 7,034         |
> |              | thumbnail edges | 47,558  | 48,880  | 47,558  | 47,969  | 52,982  | 51,363        |
> | MathOverflow | thumbnail nodes | 20,474  | 19,965  | 23,376  | 22,371  | 24,105  | 21,683        |
> |              | thumbnail edges | 160,491 | 262,295 | 315,105 | 417,491 | 478,325 | 207,581       |
> | MOOC         | thumbnail nodes | 6,120   | 6,264   | 6,275   | 6,152   | 6,340   | 7,047         |
> |              | thumbnail edges | 53,552  | 56,628  | 63,415  | 61,074  | 68,068  | 81,749        |
>
> The choice of different $\lambda$ values indirectly affects the thumbnail size, yet its size remains close to the average number of nodes(edges) across snapshots. **This ensures the two terms are of the same order of magnitude during gradient computation**, thus justifying the rationality of setting $\lambda$​’s value range within 10.
>
> Regarding the two terms balanced by $\beta$ (Eq. 15): the $TB$ term is computed via snapshot node sampling, while the downstream task mutual information is calculated using thumbnail nodes. Our supplementary experiments(Table I) confirm that **the two terms are of the same order of magnitude**, thus justifying the rationality of setting $\beta$’s value range within 10.
>
> Unfortunately, **tuning these two hyperparameters is an empirical process, as changes in data scenarios and specific requirements directly influence their optimal values.** In practice, we recommend conducting a small number of **warm-up training** runs or adopting architectures similar to **NAS** [2] for hyperparameter search, thereby achieving more reliable hyperparameter selection.
>
> ***The supplementary experiments and discussions mentioned above is supplemented in Appendix A.9.***
>
> [2] Zoph, Barret, and Quoc V. Le. "Neural architecture search with reinforcement learning." *arXiv preprint arXiv:1611.01578* (2016).

---

> ### Author Response · Authors · 2025-11-20
> **Response to Reviewer ER5F (4/5):**
>
> ## Ans 4: Regarding hyperparameter selection and sensitivity (W4, Q5):
>
> We appreciate your valuable observation regarding the sensitivity of clean-data performance to the evolutionary constraint parameter.
> Indeed, $\alpha$ governs the trade-off between the fidelity term $L_{B}$ and the evolutionary regularization $L\_{evolution}$ that enforces global structural consistency through VNGE. A large $\alpha$ encourages stronger denoising and temporal smoothing, which is beneficial under noisy or perturbed settings, but may suppress fine-grained task-specific details under clean data.
>
> As noted in **Ans 3**, hyperparameter tuning is an empirical process. We can explore the noise level of the dataset through small-scale warm-up training to achieve better hyperparameter settings.
>
> - Specifically, during the pre-training phase, **we estimate the noise level of the input sequence via the variance of von Neumann graph entropy across consecutive snapshots**, denoted as $\mathrm{Var}(H(\mathcal{G}^t))$. This value is correlated with structural volatility, and we scale $\alpha$ proportionally to this variance:
>
> $$
> \alpha = \alpha_0 \cdot \frac {\mathrm {Var}(H (\mathcal {G}^t))}{\mathrm {Var}(H (\mathcal {G}^t)) + \epsilon}.
> $$
>
> - This ensures that $\alpha$ increases only when the graph sequence exhibits unstable evolution. During the gradual increase of $\alpha$​​, we select the value that yields acceptable performance under the clean setting as the appropriate hyperparameter.
>
> ***We supplement the above-mentioned recommendations for warm-up training in Appendix A.9.***
>
> ---
>
> ## Ans 5: Regarding robustness under temporal interference (Q4):
>
> We appreciate your insightful question. The robustness of TGT to temporal order disruption stems from the VNGE-based constraint $I_s(\mathcal{G}_T; \mathcal{G})$, which fundamentally **captures the global structural distribution of snapshots rather than their temporal order.**
>
> - Specifically, the von Neumann Graph Entropy (VNGE) is defined over the normalized Laplacian spectrum(Eq. 28), which depends solely on the eigenvalue distribution of the graph. Thus, when constructing the temporal thumbnail $\mathcal{G}_T$, maximizing the mutual information between $\mathcal{G}_T$ and the input sequence $\mathcal{G}$​ encourages alignment of their **spectral information** rather than mere alignment of snapshot temporal order.
>
> - Since spectral quantities are permutation-invariant with respect to node or snapshot order, the learned thumbnail inherently focuses on **statistical patterns of structural evolution that remain stable under temporal sequence shuffling[3].** This means the model learns to **retain information about how structural complexity evolves** (e.g., entropy growth, connectivity changes) rather than the specific temporal sequence of local variations. Therefore, the VNGE-based mutual information regularization provides a temporal-order-agnostic bottleneck, enabling TGT to maintain robustness when snapshot order is permuted or partially corrupted.
>
>
> Furthermore, as discussed in Sec 4.4 and supported by the results in Table 5, TGT maintains consistent performance under temporal perturbation attacks, demonstrating that the VNGE constraint guides the model toward learning a **temporal invariant summary** that mitigates the impact of sequence tampering.
>
> ***We greatly appreciate your suggestion that we supplement the relevant explanations, and we added the above explanations to Appendix A.1.3.***
>
> [3] Braunstein, Samuel L., Sibasish Ghosh, and Simone Severini. "The Laplacian of a graph as a density matrix: a basic combinatorial approach to separability of mixed states." *Annals of Combinatorics* 10.3 (2006): 291-317.

---

> > ### Comment · Reviewer_ER5F · 2025-11-26
> > **Comments on Rebuttal**
> >
> > Thanks to the authors for their detailed response. I have carefully reviewed it and am happy to increase my rating, as my concerns have been solved.

---

> ### Author Response · Authors · 2025-11-20
> **Response to Reviewer ER5F (5/5):**
>
> Once again, we sincerely appreciate your time, effort, and thoughtful feedback. In the revised version, we have carefully addressed all of your questions and suggestions, with the corresponding modifications clearly **highlighted**. Should you have any further comments or recommendations, we would be grateful to receive them.
>
> Best wishes!

---

> ### Author Response · Authors · 2025-11-26
> **Response to Reviewer ER5F's comment:**
>
> Dear ER5F,
>
> We would like to express our sincerest gratitude for the time and effort you dedicated to reviewing our work. Your questions and suggestions have been highly valuable and have greatly benefited our research. In our rebuttal, we have strengthened the theoretical completeness and rigor of the work, which we believe has further improved the overall quality of the paper.
>
> Thank you once again for your constructive feedback and for the support you have provided.

---

### Author Response · Authors · 2025-12-02
**General Response to Area chairs**

Dear Area Chair,

Thanks for taking your time to consider our paper. We appreciate the effort required to hold the large volume of submissions and the API incident this year. To facilitate the process, **we have provided our summary in General Response.**

## **Summary of Our Work**
**a. Broad Impact and Relevance**
- We propose a novel robust representation method (**Temporal Graph Thumbnail, TGT**) for temporal graphs, which models the global evolution of the temporal graph as a **thumbnail** to regularize the representation learning.
- We model structural evolution via **VNGE** and node feature evolution via the **DV representation**, yielding the thumbnail and a representation constraint. The learned vectors support diverse downstream tasks, offering **robustness** and potential for pre-training and **broader generalization**.

**b. Technical Merit**

**Our TGT has complete theoretical support and demonstrates superior robustness to various noise interferences in experiments on various datasets.**

- **Theoretical soundness**: We provide complete proofs for all propositions, with each derivation rigorously verified, offering strong theoretical support for the effectiveness of the **thumbnail**.
- **Extensive empirical validation**: We conduct comprehensive experiments, including multi-task robustness experiments, complexity analysis, ablation studies, and hyperparameter analysis, collectively demonstrating the superiority and practical potential of our approach.

## **Improvement in Scores**

Our submission initially received ratings of 6, 6, 4, 4. **All reviewers had raised their ratings to**  $\textbf{\textcolor{maroon}{6, 6, 6, 6}}$ before **the rating rollback** caused by the API incident.

The two reviewers who initially gave a rating of 4 have raised their ratings to 6 **(before the API incident)**:

- Reviewer **ER5F** raised the rating to **6** and commented on Nov 26 (before the API incident):

  > I have carefully reviewed it and am happy to increase my rating, as my concerns have been solved.

- Reviewer **rkWZ**, who had already assigned strong initial component scores (Soundness 3, Presentation 4, Contribution 3), commented on **Nov 23**:

  > I am willing to raise my score if the authors can adequately address the **Q7** and **Q8** mentioned above.

  After we solved Q7 and Q8, rkWZ raised the rating to **6** and commented on Nov 26 (before the API incident):

  > In light of these improvements, I am willing to increase my score.

## **Addressed Concerns**

During the rebuttal period, we conducted additional experiments, strengthened theoretical analyses, and revised the manuscript with highlighted changes and **addressed all the concerns** raised by reviewers.

We concisely summarize the progress of our rebuttal below:

- We addressed **Reviewer ER5F**’s concerns regarding non-persistent nodes (Ans 1: W1, Q1), computational complexity (Ans 2: W3, Q2), hyperparameters (Ans 3: W2, W4, Q3, Q5), and temporal robustness (Ans 4: Q4). We conducted **supplementary experiments and theoretical explanations**, resolving all concerns.
- We addressed **Reviewer RLpw**’s questions on downstream task generalization (Ans 1: W1, W2), efficiency (Ans 2: Q1), and hyperparameters (Ans 3: W4). Following these suggestions, we **added a new downstream task benchmark, an efficiency test, and broader ablations**. We thank RLpw for the insightful comments, which helped further strengthen our work.
- We addressed **Reviewer QMEp**’s concerns about backbone dependence (Ans 1: Q1) and readability (Ans 2: W1, Q2). We thank QMEp for these suggestions, which directly improved the **clarity and readability** of our work.
- We addressed **Reviewer rkWZ**’s initial questions on CTDG vs DTDG (Ans 1: W2, Q5), differences from prior work (Ans 2: Q1), attack methods and baselines (Ans 3: W1, W3, Q3, Q4), and downstream tasks on large-scale datasets (Ans 4: W2, Q2, Q6). After acknowledging that these issues were resolved, rkWZ raised several additional concerns on temporal discretization (Ans 5: Q7), computational complexity on TGB (Ans 6: Q8), and adaptive attacks (Ans 7: Q9). During this period, we conducted **supplementary experiments using additional datasets, tasks, baselines, and attack methods**, which resolved all concerns and gained the approval of rkWZ.

**Prior to the API incident,  all reviewers had raised their ratings to 6,** expressing recognition for our work and rebuttal in their comments.

---

Thank you again for your recognition of our work and for providing such thorough and insightful feedback. All comments and suggestions are invaluable in helping us improve the quality and clarity of our work. Following your suggestions, we have added the additional experiments and analysis in the revised manuscript.

Hope that this summary could facilitate the next stage of review and discussion.

---

### Meta-Review · Area_Chair_aJp9 · 2026-01-07

**Summary:**

The reviewers asked for the dataset, task, and robustness experiments. The authors have responded positively to all of these requests, except for the dynamic attacks.

**Reviewer Concerns:**

ER5F
- Scalability, given the VNGE computations that involve spectral quantities. The authors added a complexity analysis to prove linearity, given an assumption of sparsity.
- Sensitivity analysis for the \alpha param. The authors explain how the alpha value is chosen in a small-scale warm-up training.

QMEp
- dependence on GAT as backbone: GCN and GIN were added.
- What does the thumbnail look like? Is it human understandable? The authors added a new figure to illustrate connectivity patterns and low-consistency edges.

rkWZ
- The robustness should be tested with recent as well as dynamic attacks. The authors added recent attacks. However, the caption of Table 13a does not indicate if higher or lower values are better.
- Link prediction only, node classification or dynamic community detection would better suit the method. The authors added node classification experiments.

**Reviewer Scores:**

- ER5F would increase from 4 to 6.
- RLpw and QMEp would most likely keep the score as 6, because they acknowledge the strengths but do not praise the article for a breakthrough or novel idea.
- rkWz would increase the score from 4 to 6, because the reviewer asks quite many things from the authors, and they implement the recent attacks. I do not believe that the score would go beyond 6.

---

### Decision · Program_Chairs · 2026-01-26

Accept (Poster)